# Distinct basal ganglia contributions to learning from implicit and explicit value signals in perceptual decision-making

Tarryn Balsdon [1,2] ✉, M. Andrea Pisauro [1,3] & Marios G. Philiastides [1] ✉

Metacognitive evaluations of confidence provide an estimate of decision accuracy that could guide learning in the absence of explicit feedback. We examine how humans might learn from this implicit feedback in direct comparison with that of explicit feedback, using simultaneous EEG-fMRI. Participants performed a motion direction discrimination task where stimulus difficulty was increased to maintain performance, with intermixed explicit- and no-feedback trials. We isolate single-trial estimates of post-decision confidence using EEG decoding, and find these neural signatures re-emerge at the time of feedback together with separable signatures of explicit feedback. We identified these signatures of implicit versus explicit feedback along a dorsal-ventral gradient in the striatum, a finding uniquely enabled by an EEG-fMRI fusion. These two signals appear to integrate into an aggregate representation in the external globus pallidus, which could broadcast updates to improve cortical decision processing via the thalamus and insular cortex, irrespective of the source of feedback.

Practising a task helps to improve performance[1]. In tasks involving a behavioural response to an external stimulus, performance can benefit from learning at the level of action selection, decision processes, or indeed, sensitivity in perceiving the task-relevant features of the stimulus[2]. Explicit feedback about whether responses were correct has been shown to increase the learning rate at each of these levels[3,4]. These performance improvements, however, have also been shown to occur in the absence of explicit feedback[1,4,5]. Though several variables could contribute to improved performance in the absence of explicit feedback (including mere exposure to a stimulus[6]), one variable of particular interest is the metacognitive evaluation of decision confidence. Confidence provides an estimate of the likelihood that a decision is correct[7,8], and so could be used as a proxy for explicit feedback, we refer to this proxy as 'implicit feedback'.

Our understanding of the role of confidence in learning lags that of explicit feedback. The influence of explicit feedback on learning can be understood from a reinforcement learning framework, which traditionally described how one learns to select actions that maximise

future external rewards in value-based learning environments[9]. More recently this framework has also been successfully applied to perceptual learning[3]. In this context, reinforcement-like learning can occur based on explicit feedback information without external rewards[10] or purely based on the observer's internal estimate of confidence as a proxy for feedback[11,12]. This suggests that the computational description of learning could be generalised to incorporate different feedback signals[13]. However, it is unclear whether the same neural processes used for explicit feedback are flexibly appropriated to implement learning from implicit signals, such as confidence.

To date, there has been no direct comparison of the neural mechanisms for learning from confidence to those of learning from explicit feedback within the same experimental task. Ideally, this comparison would be made across intermixed trials where explicit feedback is periodically available, as in ecological contexts[14]. However, in these contexts it could be beneficial to wait for infrequent but reliable feedback rather than implement learning based on confidence estimates that are an imprecise estimate of decision accuracy[15,16] or

[1]Centre for Cognitive Neuroimaging, School of Psychology and Neuroscience, University of Glasgow, Glasgow, UK. [2]Laboratory of Perceptual Systems, DEC, ENS, PSL University, CNRS UMR 8248, Paris, France. [3]School of Psychology, University of Plymouth, Plymouth, UK. ✉e-mail: tarryn.balsdon@glasgow.ac.uk; Marios.Philiastides@glasgow.ac.uk

that might be vulnerable to biases[17–19]. Alternatively, confidence could provide a more fine-grained estimate of performance, reflecting the graded precision of underlying decision processes[20], as opposed to the binary outcome of explicit feedback (which is also blind to the processing that led to the outcome). In this way, confidence could be valuable in providing more nuanced information of how to adjust decision-making processes, even in the presence of explicit feedback. For example, confidence could be linearly combined (integrated) with the predicted outcome (expected value) to form the reward prediction error from explicit feedback, as an aggregate reinforcement signal[21]. This might necessitate not only separate neural signatures for the different (confidence vs explicit value) signals but also the presence of a downstream process in which the two signals form an aggregate reinforcement-like representation to jointly influence learning.

The dopaminergic system is well known for its role in organising reward-seeking behaviour and motor responses[22,23]. The striatum, in particular, has been repeatedly linked to the representation of expected reward[24], action values[25,26], reward prediction errors[27,28], and more intrinsic motivational (hedonistic) signals[29], including confidence[12,30,31]. Similarly, the presence of extensive cortico-striatal circuits[32] along with the cortico-basal ganglia-thalamocortical loop[33], which are amenable to plastic changes via phasic dopaminergic firing[34], make the basal ganglia well positioned for implementing learning updates and guiding future actions. However, it is of ongoing debate how distinct subregions within the basal ganglia might distinctly or jointly contribute to learning[35,36], and whether there is a distinction in the computation of explicit versus implicit feedback signals.

Here, we provide observers with intermittent valid explicit feedback during a perceptual decision-making task to test whether observers learn by exploiting their internal confidence estimates, even when explicit feedback is frequently available. Patterns of behavioural perseveration suggest that this was indeed the case. We use this design of intermixed trials with and without explicit feedback to directly compare the neural signatures of learning from explicit (outcome value) and implicit feedback signals (such as confidence). We first isolate endogenous electrophysiological trial-wise estimates of confidence and outcome value using a decoding analysis of EEG. We then harness these trial-wise estimates to inform the analysis of simultaneously acquired fMRI data. We traced the BOLD signatures of implicit and explicit feedback to a spatial gradient in the striatum, where implicit feedback is represented more dorsally while explicit feedback is represented more ventrally. Moreover, these two signals of implicit and explicit feedback appear to integrate into an aggregate representation in the external globus pallidus (GPe). A psychophysiological interactions analysis suggests GPe broadcasts updates to improve cortical decision processing via the thalamus, irrespective of whether the source of feedback included explicit feedback or not.

## Results

Participants ($N = 23$) performed a variant of the classic random dot motion direction discrimination task[37] (see 'Methods') while simultaneous EEG-fMRI was recorded. Each trial was composed of three time-windows (Fig. 1a): The decision-window (in which the stimulus was presented and perceptual decision reported); the bet-window (in which participants were given the opportunity to bet that their responses were correct, 'bet trial'; or not, 'no-bet trial'); and the feedback-window (in which participants were cued whether they would receive explicit feedback or not, and if so, shown the points awarded for the trial). On each trial, 1 point was gained for a correct response, or 1 point lost for an incorrect response. These values were doubled if the participant bet that they made the correct response. The total accumulated points corresponded to a monetary bonus at the end of the experiment. On explicit-feedback trials, participants were shown the true points gained/lost on that trial; on no-feedback trials participants were instructed to infer how

many points were awarded, cued with two question marks. Explicit- and no-feedback trials were intermixed throughout the experiment. Participants were given instructions about the feedback prior to beginning the experiment.

### Behavioural signatures of learning from confidence as implicit feedback

Task difficulty (proportion of coherent dots) was controlled across blocks (6 in total) to maintain performance between 55 and 75% correct (see 'Methods'). There was an overall increase in task difficulty (Fig. 1b, mean difference in coherence between first and last block = $-10.43\% \pm 4.67$, 95% within-subject confidence interval, $t(22) = 4.64$, $p < 0.001$) with no substantial change in sensitivity (Fig. 1c, mean difference in $d' = -0.37 \pm 0.45$ $t(22) = 1.67$, $p = 0.109$, with Bayes factor, calculated based on the savage-dickey ratio with a unit-information prior, $BF_{10} = 0.75$, representing insubstantial evidence in favour of the null hypothesis for no difference in sensitivity). This is a typical signature of perceptual learning, suggesting participants learnt to improve at the task during the course of the experiment. As an additional check, we simulated observers performing the same trials as our human participants, but who do not learn, and so experience decreased sensitivity to the decreased stimulus coherence. This decreased sensitivity is plotted in Fig. 1c (blue dashed line). We designed this experiment such that participants were frequently given explicit feedback (50% of randomly intermixed trials), and so did not have to rely on decision confidence as the sole source of information to improve their performance. Evidence for learning on no-feedback trials could therefore be considered robust evidence for the involvement of confidence in learning.

We confirmed that participants use their confidence to inform their bets with two stereotypical signatures of confidence. First, participants were more likely to be correct when they bet (difference in $d'$ = $0.98 \pm 0.28$; $t(22) = 7.35$, $p < 0.001$; Fig. 1d). Second, participants showed faster reaction times when they bet (mean difference in median reaction time = $0.07\,s \pm 0.02$, $t(22) = 6.63$, $p < 0.001$; Fig. 1e). Although the overall tendency to bet may be biased by other factors such as risk aversion, trials on which participants bet still reflect higher confidence, and so bet responses can be used to decode neural representations of post-decision confidence in the decision-window.

We confirmed that participants were relying on the explicit feedback using an analysis of response perseveration, a typical signature of incorporating feedback for learning. Participants should not repeat the same response to the same stimulus type after receiving negative feedback (and should, after receiving positive feedback). This was indeed the case, a $2 \times 2$ repeated measures ANOVA showed a significant main effect of feedback sign (positive vs negative) on the tendency to repeat a response given a stimulus repetition (normalised by the repetition tendency irrespective of feedback; Fig. 1f, left; $F(1,22) = 60.04$, $p < 0.001$ after Bonferroni correction for three comparisons). This effect was somewhat moderated by feedback value (|1| vs |2|; but the interaction would not survive correction for multiple comparisons; $F(1,22) = 5.18$, $p = 0.033$, uncorrected). As an additional check, we compared the differences in response perseveration with the behaviour of simulated observers who do not learn (as in Fig. 1c dashed blue line). We found that the probability of obtaining as large a difference in perseveration due to feedback sign as observed in our data was $p < 0.001$; and the probability of obtaining an interaction as large, $p = 0.006$. This suggests that feedback did reinforce behaviour, that is, participants relied on explicit feedback to improve their performance (learn).

Given the design of intermixed explicit-feedback and no-feedback trials, participants could have relied solely on explicit feedback to improve their performance. However, we found the same pattern of response perseveration on no-feedback trials, depending on whether the participant bet on their previous response (Fig. 1f, right; within-

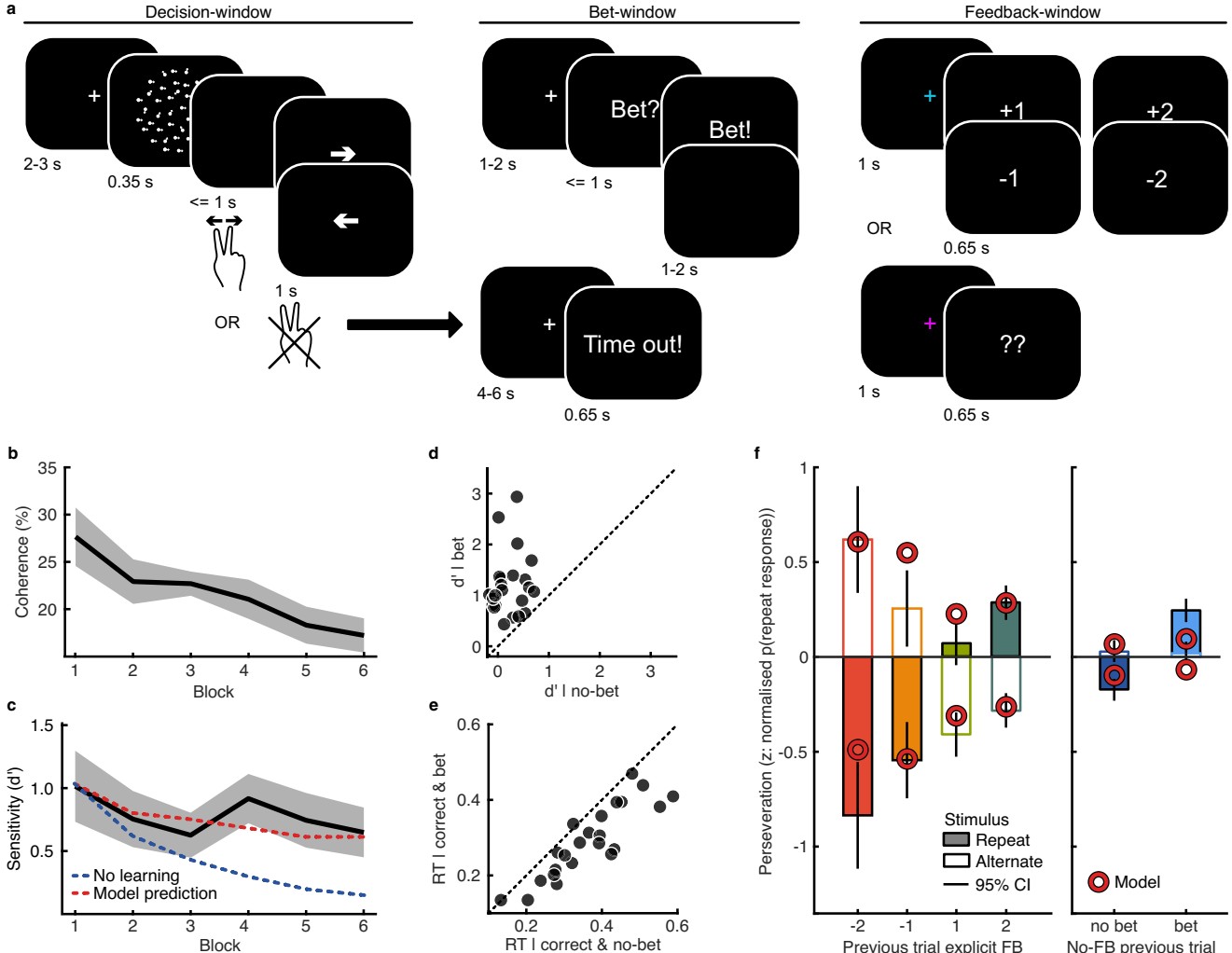

**Fig. 1 | Methods and behaviour. a** Trials consisted of three time-windows: the decision-window; bet-window; and feedback-window. In the decision-widow, a fixation cross was presented for a variable duration, followed by 350 ms of random dot motion, and up to 1 s to enter the left/right response, with visual feedback about the button pressed. In the bet-window, participants were given up to a second to bet that their response was correct, to double the value of the points gained/lost on that trial. In the feedback-window, participants were cued (coloured fixation) about whether they would receive explicit feedback, or be shown two question marks indicating they should think about how many points they think they won/lost on that trial. If participants failed to enter their left/right decision within one second after the stimulus, they were informed with the words 'time out', they lost a point, and the trial was excluded from the analysis. **b** Coherence (percent of dots moving in the correct direction) was adjusted each block (50 trials) to maintain accuracy around 75% correct. Line shows the average, shaded region shows 95% within-subject confidence intervals. **c** Average sensitivity ($d'$) across blocks, with 95% within-subject confidence intervals shaded. The red dashed line shows the fit of the best fitting model, which uses confidence to learn on every trial. The blue dashed line shows how sensitivity would decrease with decreased stimulus coherence simulating observers who do not learn. **d** Sensitivity ($d'$) on bet (ordinate) and no-bet (abscissa) trials. **e** Median reaction time (s) on correct trials for bet and no-bet trials. **f** Perseveration, an average of the normalised probability of repeating a response, for repeat (filled) and alternate (open) stimuli, by feedback on the previous trial, or the previous bet response on no-feedback trials (error bars show 95% CI within-subjects difference between repeat and alternate stimuli). The markers show the predictions of the computational model using confidence to learn. All panels include $N = 23$ participants, source data are provided as a Source data file.

subjects t-test, t(22) = 4.69, $p < 0.001$. This is in line with the hypothesis that high confidence is treated as a positive outcome, to be used as reinforcement.

We used a simple computational model comparison to support the behavioural evidence for learning from confidence. The models, based on previous perceptual learning models[2,3], assume the basic framework of Signal Detection Theory[38], where responses are made by placing a criterion to discriminate the perceptual evidence from leftward vs rightward stimuli (two overlapping Gaussian distributions). Learning improves sensitivity to the stimuli[2], the distance between the means of the perceptual evidence, in units of standard deviation (though sensitivity is decreased for stimuli with decreased coherence). We model learning by shifting the mean perceptual evidence, $\mu$, away

from the response criterion (see 'Methods' for details). The shift is implemented in accordance with reinforcement learning, where the size of the shift is proportional to the reward prediction error (the difference between the explicit feedback value, $r_t$, and the expected value, $E[V_t]$), moderated by a learning rate, $\alpha$, such that: $\mu_{t+1} = \mu_t + \alpha(r_t - E[V_t])$. Three models were implemented to compare which trials participants use confidence to learn on: (1) a model that does not use confidence, learning only occurs on explicit feedback trials; (2) a model that learns in the same way on feedback trials, but additionally uses confidence to learn on no-feedback trials (confidence substitutes the reward prediction error); (3) a model that uses confidence to learn on all trials, where confidence moderates the expected value on explicit feedback trials (similar to ref. 13). The model that used

confidence on all trials provided the best description of behaviour (Fig. 1c,f; $\sum BIC_3 - BIC_1 = -82.21$; $\sum BIC_3 - BIC_2 = -32.11$; protected exceedance probability = 0.94; see 'Methods', Supplementary Fig. S1). This suggests confidence is used for learning even on trials where explicit feedback is provided. Indeed, we found that simulated behaviour learning from explicit feedback alone did not show the difference in response perseveration following bet/no-bet responses (as in Fig. 1f, $p = 0.046$). Note that this modelling exercise merely supports the use of confidence for learning, we do not seek to examine its exact implementation. Indeed, these models do not sufficiently capture all aspects of the behavioural data, such as the interaction between feedback value and sign, which may require more complex model formalisations.

### Neural signatures of implicit and explicit feedback

We used an asymmetric (EEG to fMRI) fusion analysis[39,40] of the simultaneous EEG-fMRI data. To examine the EEG signatures of confidence as implicit feedback, a Linear Discriminant Analysis (LDA, see 'Methods') was used to isolate weights on the EEG channel activity (spatial filter) that best discriminated bet from no-bet trials in the decision-window, which was separated from the bet-response by 1–2 s. The summed product of the spatial filter and channel activity gives the bet-prediction, a continuous variable where, for each trial, the higher the bet-prediction the more likely the trial was a bet trial. Spatial filters were first generated for each time-point within the decision-window, and then a robust individual filter was selected for each participant as the average of five consecutive filters that best discriminated bet from no-bet trials within the group-level significant time-window (see 'Methods'). Applying these individual spatial filters over time within a trial shows how the EEG activity relevant for discriminating bet from no-bet trials emerges over time (Fig. 2a). The average topography of the discriminating components of this spatial filter (insert of Fig. 2a) is in line with the literature on post-decision confidence[41–44]. There is no evidence that the finger movements in the bet window (1–2 s later) had any effect on the LDA, nor potential confounds from the perceptual decision task (see Supplementary Fig. S3).

As validation of the behavioural relevance of the EEG bet-prediction, we show that the bet-prediction not only discriminated bet from no-bet trials (from the time of the response to 0.25 s after, mean $F(1,22) = 15.36$, cluster corrected $p < 0.001$; Fig. 2a, left), but also showed a significant main effect of response accuracy (from 0.1 to 0.24 s after the response, mean $F(1,22) = 6.66$, cluster corrected $p = 0.002$). That the decoded bet-prediction in the decision-window is related to post-decision confidence is supported by the significant prediction of correct responses (mean GLM $\beta$-weight = 0.22 ± 0.16, $t(22) = 6.62$, $p < 0.001$) and reaction times on correct trials (mean GLM $\beta$-weight = 0.15 ± 0.06, $t(22) = 5.31$, $p < 0.001$). The decoder is not driven by an error detection signal, as the decoder predicted bet decisions even on correct trials only (mean GLM $\beta$-weight = 1.53 ± 0.42, $t(22) = 17.34$, $p < 0.001$). In this way, the EEG bet-prediction could reflect a more graded representation of the underlying confidence used to arbitrate whether or not to bet on individual trials.

Applying the same spatial filters estimated within the decision-window onto EEG activity during the feedback-window (i.e. projecting the data through the same "spatial generators" discriminating bet from no-bet responses earlier in the trial), we found the confidence-relevant EEG activity re-emerged. On no-feedback trials, the bet-prediction dissociated bet from no-bet trials (from 0.35 to 0.45 s following feedback, mean $t(22) = 2.58$, cluster corrected $p = 0.003$, Fig. 2b, top right), suggesting that in the absence of explicit feedback, participants did follow the instruction to use their confidence to infer the points they may have gained/lost. On feedback trials, a small window of significant difference between explicit feedback sign was driven primarily by the trials with feedback of −2 (betting on an incorrect response; main effect of feedback sign, mean $F(1,22) = 4.96$, cluster corrected

$p = 0.035$; Fig. 2b, top left). This could be due to the salience of explicit feedback contradicting confidence, or perhaps the integration of explicit feedback to revise confidence.

The signals underlying this bet-prediction that re-emerged in the feedback-window had evolved from the decision-window confidence representation (median correlation coefficient $r = 0.03$, range [−0.07, 0.15]), and were significantly less related to post-decision confidence per se (significantly less predictive of choice accuracy; within subject difference in beta-weights, $t(22) = 3.46$, $p = 0.002$; and response time, $t(22) = -4.27$, $p < 0.001$). We later present some evidence that the bet-prediction in the feedback-window is more related to an implicit value signal.

We next identified the EEG signatures of explicit feedback, using the same LDA analysis to discriminate positive (+1 or +2 points) from negative (−1 or −2 points) explicit feedback trials, training within the feedback-window. Spatial filters were first generated for each time-point within the feedback-window, and then a robust individual filter was selected for each participant as the average of five consecutive filters that best discriminated positive from negative feedback trials within the group-level significant time-window (see 'Methods', Supplementary Fig. S2).

Similar to the bet-prediction, we applied this filter over time in the feedback window (Fig. 2b, bottom left). Splitting trials by feedback, there was a main effect of feedback sign (positive vs negative feedback) from 0.35 to 0.53 s (2 (feedback sign) × 2 (feedback value) ANOVA, mean $F(1,22) = 9.62$, cluster corrected $p < 0.001$). Despite being trained only to discriminate feedback sign, the feedback-prediction showed an interaction between feedback sign and absolute (1 vs. 2) value (same ANOVA, from 0.43 to 0.52 s following feedback, mean $F(1,22) = 6.81$, cluster corrected $p = 0.006$). This reflects a representation of overall outcome value as opposed to just outcome valence. Moreover, the feedback-prediction in the feedback-window predicted response perseveration on the following trial (mean GLM $\beta$-weight for repeating stimulus/response interaction = 0.08 ± 0.02, $t(22) = 8.77$, $p < 0.001$), suggesting this analysis was sensitive to the activity relevant for learning from explicit feedback.

In addition, the feedback-prediction also dissociated bet from no-bet trials on no-feedback trials (from 0.17 to 0.26 s following feedback, mean $F(1,22) = 2.84$, cluster corrected $p = 0.002$; Fig. 2b, bottom right), which could reflect signatures of learning from implicit feedback. Applying the spatial filter back in time onto EEG activity during the decision-window, the feedback prediction also dissociated bet from no-bet trials (from 0.17 to 0.30 s following the response, mean $t(22) = 2.96$, cluster corrected $p < 0.001$; Fig. 2a, bottom). This suggests that some feedback-relevant EEG activity may be present following decisions, either in relation to expected feedback or a direct implementation of early learning updates prior to explicit feedback.

We then examined the relevance of the bet- and feedback- predictions for learning. While both the feedback-prediction and the bet-prediction showed behaviourally relevant signals in the decision- and feedback-windows, we examined whether the signals in the feedback-window were more relevant for learning. We used the feedback-window EEG-predictions (feedback-prediction on feedback trials and bet-prediction on no-feedback trials) as the reward prediction errors in the computational model, and found this resulted in no substantial difference in the fit to behaviour (compared to the behaviour-only model relying on the explicit feedback and bet responses; $\sum BIC_{beha} - BIC_{EEG} = 1.004$, protected exceedance probability in favour of the EEG-informed model = 0.45). But, the EEG-predictions from the feedback-window provided a better description of behaviour than those from the bet-window ($\sum BIC_{EEG\_fb} - BIC_{EEG_d} = -141.08$; protected exceedance probability >0.99). This indicates that the feedback-window predictions are more related to outcome value and its use for learning, than the earlier signals from these same spatial filters in the decision-window.

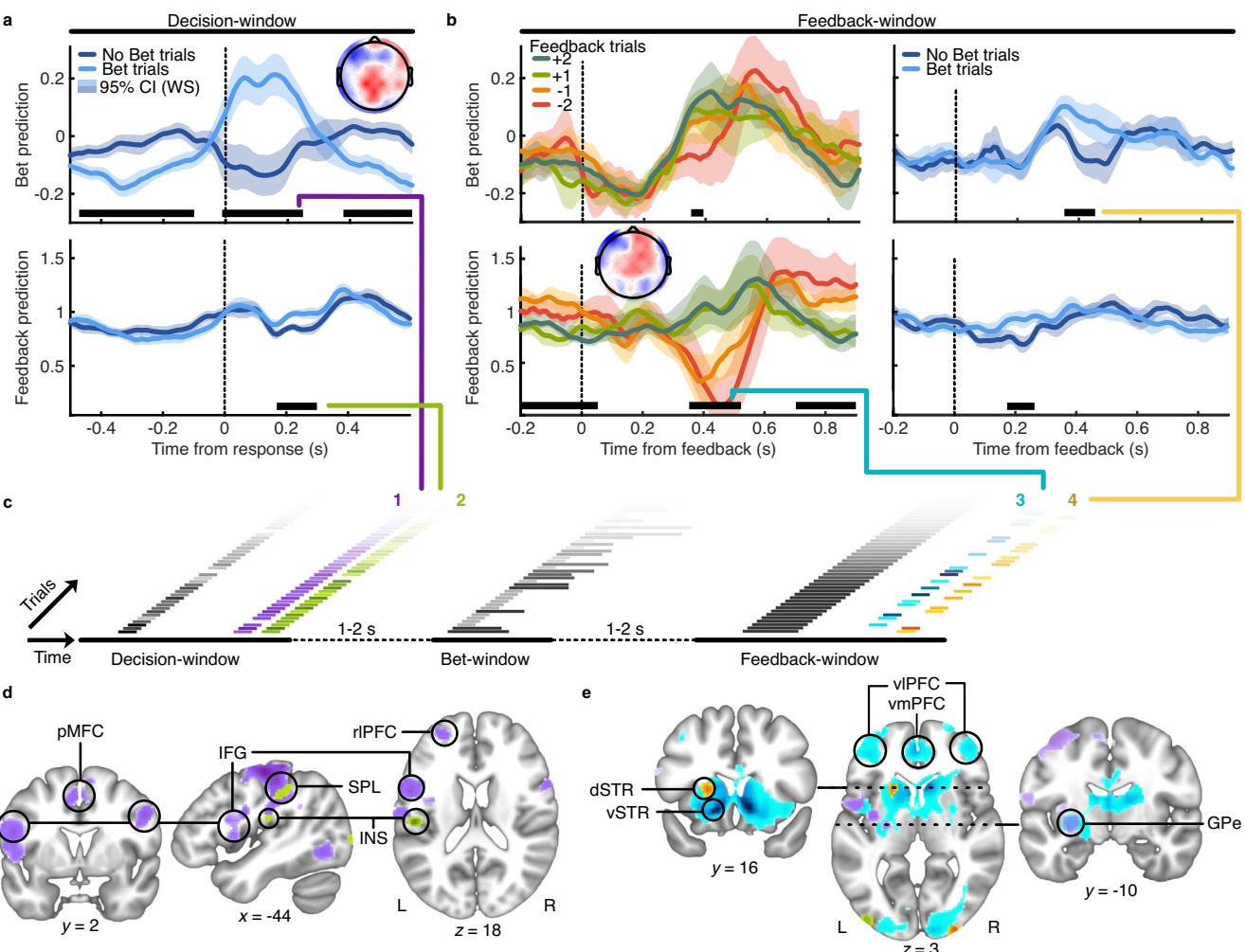

**Fig. 2 | EEG signatures of implicit and explicit feedback and their BOLD correlates. a** EEG bet-prediction in the decision-window (top) generated by selecting the spatial filter that best discriminated bet from no-bet responses in the decision-window (insert shows the group average scalp projection). Below, EEG feedback-prediction in the decision-window, from the spatial filter generated in the feedback window (insert in bottom left of panel **b**). The lines show the average grouping trials by whether the participant bet (light blue) or not (dark blue). Shaded regions indicate 95% within-subjects confidence intervals, and black horizontal lines show windows of cluster-corrected significant differences. **b** EEG predictions in the feedback-window for explicit feedback trials (left) and no-feedback trials (right). The top row shows the bet-prediction, using the spatial filter from the decision-window. The bottom row shows the feedback-prediction, based on the spatial filter (scalp projection in insert) from the decoding analysis on the feedback trials in this feedback-window. Same format as (**a**). **c** Construction of the EEG-informed fMRI GLM analysis with four EEG regressors (coloured) and traditional regressors (grey)

which corresponded to stimulus onset (with amplitude modulated by decision time), bet-cue (with duration modulated by bet response time), and feedback cue. Time within a trial progresses left to right, with example trials layered. **d** Clusters of voxels with a significant positive relation to the EEG bet-prediction (1, purple) and feedback-prediction (2, green) in the decision-window (darker colour indicates greater voxel z-statistic; All results are reported at Z ≥ 2.57, and cluster-corrected using a resampling procedure; see 'Methods'). **e** Clusters of voxels with a significant positive relation to the EEG feedback-prediction (3, blue) and bet-prediction (4, yellow) in the feedback-window, with the clusters from the decision-window for comparison. dSTR dorsal striatum, GPe external globus pallidus, IFG inferior frontal gyrus, INS insular cortex, pMFC posterior medial frontal cortex, rlPFC rostrolateral prefrontal cortex, SPL superior parietal lobe, vlPFC ventrolateral prefrontal cortex, vmPFC ventromedial prefrontal cortex, vSTR ventral striatum. All panels include N = 23 participants, source data are provided as a Source data file.

Together these analyses give us four EEG-predictions to inform the GLM analysis of simultaneously acquired fMRI BOLD signal: the bet-prediction that best discriminates bet from no-bet trials in the decision-window (related to post-decision confidence); the bet-prediction that re-emerges at the time of feedback, that discriminates bet from no-bet trials in the absence of explicit feedback (related to implicit outcome value); the feedback-prediction that best discriminates positive from negative feedback in the feedback-window (related to explicit outcome value); and the feedback-prediction in the decision-window, that shows the pattern of EEG activity relevant for discriminating the sign of explicit feedback is present even before explicit feedback is given (related to expected outcome value). In addition, regressors on stimulus onset (modulated by response time),

the bet cue, and the feedback cue were used to capture BOLD related to these externally driven events (Fig. 2c, see 'Methods'). Full details of the results of this GLM can be found in Supplementary Figs. S4–S11 and Supplementary Table S1. An analysis of the variance inflation factor indicated correlations in these variables were not substantial enough to be problematic for the fMRI analysis (the maximum ranged between 1.67 and 4.25 across subjects). The most correlated EEG-predictors were the bet-prediction in the decision-window and the feedback-prediction in the decision-window, the median correlation across subjects was $r = 0.115$ (ranging from −0.039 to 0.53).

These EEG-predictions give a fine-grained estimation of the subjective variables used to implement behavioural responses, as well as capturing trial-by-trial variability in the neural activity underlying these

internal variables. In this way they afford greater explanatory power in capturing meaningful differences between trials with otherwise the same behavioural outcomes. In addition, these predictions disentangle effects in close temporal proximity (within the decision- or feedback-windows), due to the temporal resolution of electrophysiological signals, which would otherwise be difficult to dissociate in sluggish BOLD responses. Here, we focus on clusters of voxels in which BOLD related to the EEG-predictions, leveraging the trial-by-trial variability in the internal representations captured by the LDA analysis of the electrophysiological signals. We expect these EEG-predictions to be related to (respectively): (1) post-decision confidence; (2) expected outcome value; (3) explicit outcome value; and (4) implicit outcome value.

In the decision-window, the EEG representation of post-decision confidence (1) was associated with significant clusters in bilateral parietal lobe, posterior medial frontal cortex, inferior frontal gyri, and left rostrolateral prefrontal cortex, reflecting both regions involved in decision-making and the computation of decision confidence, consistent with previous literature[20,41–43,45,46] (Fig. 2D, Supplementary Fig. S12). In addition, the external globus pallidus was significantly related to post-decision confidence, and these significant voxels largely overlapped with those corresponding to the later representation of explicit outcome value (3). The representation of expected outcome value (2) was, in particular, associated with the parietal operculum, extending into insular cortex, which has previously been associated with the anticipation of reward[47]. In the feedback-window, the representation of explicit outcome value (3) was associated with regions consistent with the valuation network, including the striatum and frontal lobes[48–50], while the representation of implicit outcome value (4) was associated with a cluster in left dorsal striatum (Fig. 2e). A key finding emerging from these results is the presence of a dorsal-ventral spatial gradient within the striatum, where implicit outcome value (based on confidence) is represented more dorsally, while explicit outcome value is represented more ventrally.

Of note, this EEG-informed analysis produces some key differences from the binary behavioural variables that would have been used in a stand-alone fMRI analysis (see Supplementary Figs. S4–S11). In particular, the binary bet vs no-bet comparison in the decision-window does not capture the relationship between post-decision confidence and the external globus pallidus. Similarly, the dorsal striatum cluster corresponding to the re-emergence of confidence as an implicit outcome value estimate in the feedback-window is absent for a comparison of binary bet vs no-bet trials in the feedback window. The stand-alone fMRI analysis does resolve a cluster in the caudate (which has previously been associated with learning under uncertainty[51,52]), however, this does not capture the gradient with explicit value in the striatum, which deserves further investigation. Had the endogenous trial-wise variability captured in the EEG-predictions resulted from unrelated noise, these variables would have been less powerful predictors in the fMRI GLM. The EEG-informed analysis also provides more specific results, for instance, the stand-alone feedback regressor results in a large cluster over occipital cortex that likely captures the difference in luminance of the feedback visual cues. This suggests that the EEG-predictions give a closer approximation of the graded subjective variables underlying behaviour, where this can reveal a richer picture of the BOLD correlates, especially in the basal ganglia.

### Integration of implicit and explicit feedback in external Globus Pallidus

Our computational modelling analysis suggested learning is supported by both confidence and explicit feedback when explicit feedback is given, implying some integration of these signals at the neural level. A cluster of voxels centred on the external globus pallidus showed significant relation to both the EEG representation of post-decision confidence and later, in the feedback-window, the EEG representation of explicit outcome value (Fig. 2e). The external globus pallidus (GPe)

receives both cortical and striatal projections as well as sending inhibitory projections to the subthalamic nucleus to control motor (dis) inhibition[53,54], thereby playing an important role in the mediation of motivated behaviour[55]. For this reason, we investigated the post hoc hypothesis that the external globus pallidus acts as a main subcortical hub for integrating implicit and explicit feedback to drive learning.

First, we examined the relationship between GPe BOLD and BOLD related to implicit and explicit feedback. We isolated the subcortical voxels that were jointly significant for post-decision confidence and explicit outcome value as the GPe region of interest (ROI) and compared the BOLD response in this ROI with the dorsal striatal cluster related to implicit outcome value, and a ventral striatal cluster most strongly related to explicit outcome value (Fig. 3a). The connectivity between the striatal ROIs and the GPe was assessed by taking the single-trial Pearson correlation between the voxel-average normalised BOLD over the 10 s following feedback. The average (Fisher transformed) correlation is presented in Fig. 3b for trials in each feedback condition separately (the BOLD time-courses within the feedback window for each ROI are shown in Fig. 3c, averaged within each feedback condition, the reader may appreciate the qualitative similarity with the EEG predictions in the feedback-window). Both dorsal and ventral striatum showed strong positive correlation with GPe (dorsal mean = 0.498 ± 0.096; ventral mean = 0.275 ± 0.066; which was significantly less than the dorsal correlation within subjects $t(22) = 8.16$, $p < 0.001$). For both dorsal and ventral striatum, the correlation with GPe was on average greater the more explicit feedback disagreed with bet-choices (when the participant bet but received negative feedback, compared to when they bet but received positive feedback), although the effect of feedback condition was not significant (comparing positive and negative feedback on bet trials, $t(22) = 0.48$, $p = 0.63$ for ventral striatum, and $t(22) = 1.94$, $p = 0.07$ for dorsal stratum). Overall, GPe BOLD covaried with both dorsal and ventral striatum, without substantial difference depending on whether explicit feedback was provided, nor on the outcome value.

Next, we assessed the post hoc hypothesis that the GPe BOLD in the feedback-window could be driven by earlier BOLD related to post-decision confidence, by taking the correlation in BOLD from up to 10 s following the decision with the later GPe BOLD from up to 10 s following feedback. Figure 3d shows the average (Fisher transformed) correlation of the GPe BOLD in the feedback-window with the BOLD in the decision-window for the GPe (itself), dorsal striatum, ventral striatum, three regions related to post-decision confidence (inferior frontal gyrus, left rlPFC, and posterior medial frontal cortex), as well as the insula (related to the expected outcome value). GPe BOLD in the feedback window was significantly correlated with the earlier BOLD in the IFG, and this correlation was on average greater than (though not significantly different from) the earlier BOLD in the GPe itself, suggesting there could be a Granger causal connection. We found no evidence for a difference in this relationship depending on whether explicit feedback was provided ($t(22) = 0.912$, $p = 0.37$). We formally assessed this relationship within subjects using vector autoregressive models of the BOLD timeseries from the GPe and IFG clusters (see 'Methods'). Leave-one-out Granger causal tests showed significant evidence against excluding the lagged IFG BOLD in predicting GPe BOLD in 18/23 participants (median $X^2 = 38.3$, median $p = 2.62e^{-6}$, median lag = 12 s, including non-significant participants). That is, earlier BOLD related to confidence in the IFG continues to drive GPe BOLD responses following feedback. Taken together, these results suggest that GPe integrates post-decision cortical confidence and later subcortical outcome value signals, with no substantial difference in this integration depending on whether explicit feedback is provided. However, we emphasise that this effect is small (Fig. 3D) and encourage this integration of confidence and explicit feedback to be more closely assessed in future work.

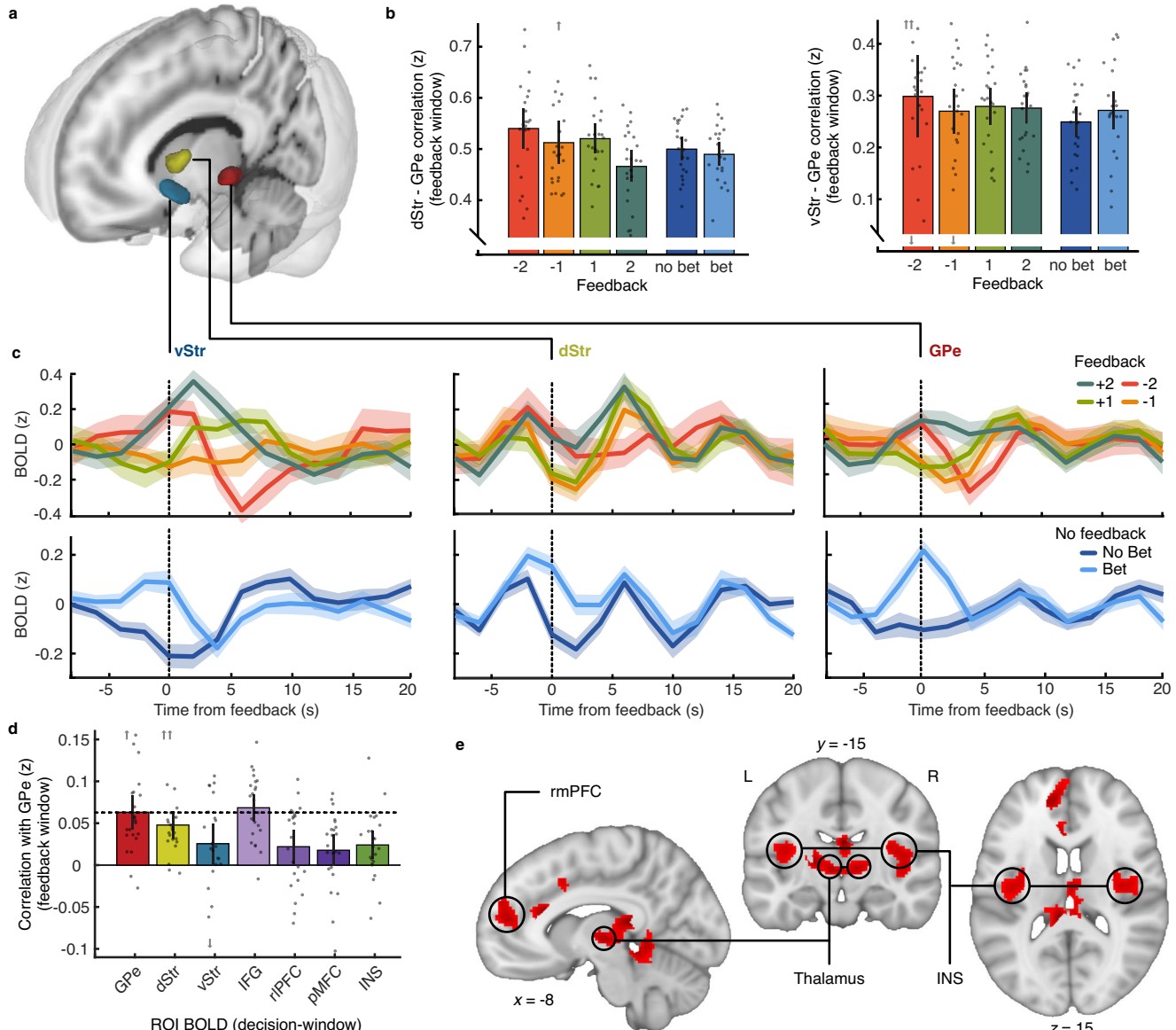

**Fig. 3 | Implicit and explicit feedback signals in the external globus pallidus.**
**a** Subcortical regions of interest from the feedback-widow: the dorsal striatum (dStr, yellow, related to the bet-prediction in the feedback-window); the ventral striatum (vStr, blue, related to the feedback-prediction in the feedback-window) and the external globus pallidus (GPe, red, related to both the feedback-prediction in the feedback-window, and the bet-prediction in the decision-window).
**b** Correlation between GPe and dStr (left) and vStr (right) by feedback condition (or bet/no-bet trials on no-feedback trials), taken as the average Fisher transformed Pearson correlation coefficient over the 10 s following feedback. Error bars show 95% within-subject confidence intervals. Points show individual participants centred to the group mean (arrows indicate out-of-range points). **c** Time-course of the normalised BOLD response (averaged over voxels) following feedback by explicit feedback condition (top) and no-feedback condition (bet vs. no-bet, bottom) for vStr (left), dStr (middle), and GPe (right). Lines show averages across

participants. Shaded regions show 95% within-subject confidence intervals.
**d** Correlation between ROI BOLD in the 10 s following the decision with GPe BOLD in the 10 s following feedback, 6–8 s later (average Fisher transformed Pearson correlation coefficient). Horizontal dashed line marks the within-region correlation, which can be taken as a benchmark, error bars show 95% within-subject confidence intervals. Colours correspond to the clusters in Fig. 2d, e. Points show individual participants centred to the group mean (arrows indicate out-of-range points).
**e** Significant clusters from the PPI analysis showing the interaction between GPe BOLD following feedback and response perseveration on the following trial. dSTR dorsal striatum, GPe external globus pallidus, IFG inferior frontal gyrus, INS insular cortex, pMFC posterior medial frontal cortex, rlPFC rostrolateral prefrontal cortex, rmPFC rostromedial prefrontal cortex; vSTR ventral striatum. All panels include $N = 23$ participants, source data are provided as a Source data file.

Finally, we explored how an integrated feedback signal encoded in GPe could be used to implement learning, using a Psychophysiological interaction analysis (see 'Methods', Supplementary Fig. S13). As the psychological variable we took the interaction between stimulus and response repetition on the following trial, that is, response perseveration. As suggested by behaviour (Fig. 1f), we expect positive feedback to be used to increase the likelihood of repeating a response given a stimulus repetition, and negative feedback used to decrease

the likelihood of response repetition given stimulus repetition. The psychological variable modelled this interaction, with positive values for switching responses to repeated stimuli following negative feedback (or no-bet responses in the absence of explicit feedback), as well as repeating responses for repeating stimuli following positive feedback (or bet responses in the absence of explicit feedback). The physiological variable was the GPe BOLD from the time of feedback to the start of the next trial. The analysis therefore resolves where increased

coupling with GPe (following feedback on trial $n$) results in a stronger influence of (implicit and explicit) feedback on the decision processes on the following trial (pattern of perseveration on trial $n+1$). Figure 3e shows the results of this analysis over all trials (both explicit feedback and no-feedback trials), highlighting significant clusters in the thalamus, insula, and the rostromedial prefrontal cortex. The GPe could therefore modulate action value (in medial PFC) and action (dis)inhibition in the thalamus to update decision processes based on both implicit and explicit feedback[56,57]. Running the PPI analysis separately on explicit feedback trials and no-feedback trials resulted in clusters of significant voxels overlapping with those presented in Fig. 3e, with no substantial (>2.57) differences in the z-statistics (Supplementary Fig. S14). This is consistent with the suggestion that the GPe integrates implicit and explicit sources of feedback to drive learning irrespective of the feedback source.

## Discussion

We tested the use of decision confidence for learning in a context where participants could have relied solely on frequent explicit feedback to improve their performance. At the behavioural level we observed no-feedback trials to be modulated by confidence in a similar manner to explicit feedback trials, and a computational modelling analysis suggested learning integrated confidence even on explicit feedback trials. This was supported by our analysis of the neural data. Distinct implicit/explicit sources of value information modulated striatal responses along a dorsal-ventral spatial gradient. We saw evidence that implicit and explicit striatal value signals were integrated in the external globus pallidus, which was significantly modulated by confidence in the decision-window, and by explicit outcome value in the feedback-window. Stronger connectivity between the external globus pallidus and the thalamus, insular and frontal cortex predicted the interaction between response perseveration and implicit/explicit feedback sign, supporting the role of GPe in modulating learning via information flow in the basal ganglia.

Our results point to additional processes by which confidence can be used for learning. The neural signatures of confidence were not only present following decisions, but re-emerged at the time of feedback. In the decision-window, the confidence prediction estimated from the linear discriminant analysis of the EEG signals corresponded both to brain regions associated with decision-making and the computation of confidence (in line with previous literature[20,41–43,45,46]). Confidence evolved after the decision-window (as has been shown for outcome value signals[48]), such that while the confidence predictions from the feedback-window were estimated using the same spatial generators as the decision-window, the predictions were found to systematically covary with a distinct cluster of BOLD in the dorsal striatum. The distinction between post-decision confidence and the use of confidence as implicit outcome value was made prominent in this experiment by intermixing explicit- and no-feedback trials with a separate outcome stage, whereas previous experiments examining the neural signatures of confidence in learning have not included an outcome stage[11,12]. We were thus able to delineate separable neural processes associated with the computation of post-decision confidence and the use of confidence as implicit feedback to inform future behaviour.

While the confidence derived implicit feedback signals were isolated to the dorsal striatum, the strongest relation with the EEG explicit outcome value prediction was found in the ventral striatum. This is suggestive of a striatal spatial gradient along the dorsal-ventral axis for implicit vs explicit outcome value, respectively. There are various accounts of the functional division of dorsal and ventral striatum, including associative aspects[35], the type of learning[58], and the learning phase[59,60]. These results add nuances to this discussion: here we see a graded reliance on implicit and explicit feedback representations along the dorsal-ventral axis of the striatum, distinguished across intermixed explicit- and no-feedback trials within the same task. This emphasises the broader distinction of the roles of dorsal and ventral striatum within the context of their cortical inputs[61,62]. Previous studies have shown an integration of post-decision confidence with subjective external value in ventromedial prefrontal cortex[63,64] yet here we suggest separable encoding in the striatum. In this way, the value of an external motivator (whether inferred or explicitly signalled reward) and the value of internal motivations (confidence in performing well at the task) could be encoded separately for the sake of flexible weighting according to the learning context.

Our analysis further suggests these implicit and explicit representations of outcome value could be integrated to form an aggregate representation in the external globus pallidus. BOLD in the GPe was related to EEG-predicted confidence in the decision-window and explicit outcome value in the feedback-window, while a Granger causal analysis suggested feedback-window BOLD was also influenced by earlier BOLD in the IFG related to confidence (although this was a small effect). The external globus pallidus is a central part of the indirect pathway through the basal ganglia[53]. Activation of the indirect pathway was originally thought to increase cortical inhibition via the subthalamic nucleus and thalamus[65]. The functioning of the indirect pathway has since been shown to be more complex[55,66,67], including the modulation of decision-making processes[68,69]. In particular, Lilascharoen and colleagues[70] have shown GPe neurons connecting directly to the parafascicular thalamic nucleus modulating behavioural flexibility. This is in line with our connectivity analysis, where the strength of connectivity between GPe and the thalamus and insular cortex predicted the interaction between response perseveration and implicit/explicit outcome. These exploratory findings therefore build on mounting evidence supporting the cardinal role of GPe in modulating behaviour[54], where in addition, strong projections back from GPe to GABA interneurons in the striatum[66] put the GPe in the position to control information flow throughout the basal ganglia[71], which deserves further investigation in future work.

In summary, these results are consistent with the ubiquitous use of confidence as implicit feedback for improving future behaviour. Our analysis suggests the value of implicit feedback is encoded in a distinct manner to that of explicit feedback, where we show a dorsal-ventral spatial gradient within the striatum corresponding to these distinct motivational sources. This illustrates a distinction in striatal coding of value within a single task and learning context. The signals from dorsal and ventral striatum appeared to be combined in the external globus pallidus, facilitating the updating of choice behaviour for learning via the thalamus in similar ways following both explicit-feedback and no-feedback trials. This highlights that even when external feedback is available, metacognitive estimates of confidence could provide us with additional nuanced information to update our internal processes for improving behaviour.

## Methods

### Participants

Participants ($N = 30$; 17 male/13 female; age range: 22–33 years) were recruited from the local mailing list and asked to provide written informed consent prior to beginning the experiment. All were right-handed, had normal or corrected to normal vision, and reported no history of neurological problems. The study was approved by the College of Science and Engineering Ethics Committee at the University of Glasgow (CSE01355). Participants were remunerated for their participation in the experiment based on their overall performance (up to a maximum of £10) and an additional fixed payment of £10 for their participation. Due to problems in data collection, two participants were excluded, an additional four participants were excluded for performance below 55% correct, and one participant for too few 'bet' responses (betting on fewer than 15% of trials).

## Materials

MRI data was collected using a 3-Tesla Siemens TIM Trio MRI scanner (Siemens, Erlangen, Germany) with a 12-channel head coil. Functional volumes (235 per block) were captured with a T2*-weighted gradient echo, echo-planar imaging sequence (32 interleaved slices, gap: 0.3 mm, voxel size: $3 \times 3 \times 3$ mm, matrix size: $70 \times 70$, FOV: 210 mm, TE: 30 ms, TR: 2000 ms, flip angle: 80˚). An anatomical reference image was acquired using a T1-weighted sequence (192 slices, gap: 0.5 mm, voxel size: $1 \times 1 \times 1$ mm, matrix size: $256 \times 256$, FOV: 256 mm, TE: 2300 ms, TR: 2.96 ms, flip angle: 9˚). For distortion correction, phase and magnitude field maps were acquired ($3 \times 3 \times 3$ mm voxels, 32 axial slices, TR: 488 ms, short TE: 4.92 ms, long TE: 7.38 ms).

EEG was collected using a 64-channel (10–20) MR-compatible EEG amplifier system (Brain Products, Germany; with Ag/AgCl electrodes, with in-line 10 kOhm surface-mount resistors, EasyCap GmbH, Germany), recorded with Brain Vision software (Brain Vision, USA) at a sampling rate of 5000 Hz. Reference and ground electrodes were built-in between electrodes Fpz and Fz and between electrodes Pz and Oz, respectively. EEG was synchronised with the MRI data acquisition (Syncbox, Brain Products, Germany) with MR triggers stretched to 50 $\mu$s using an in-house pulse stretcher. EEG cables were bundled and secured to a cantilever beam running out the back of the bore.

Stimuli were presented on an LCD projector (running at 60 Hz) viewed from the rear of the MR scanner bore via a mirror at a total distance of 95 cm. Stimulus presentation was controlled using Presentation software (Neurobehavioral Systems). Behavioural responses were collected using an MR-compatible button box.

## Task

Each trial was composed of three time-windows, separated by a variable interval of 1–2 s: the decision window; the bet window; and the feedback window. Trials were separated by a variable interval of 2–3 s, with a mid-block break of 30 s. Participants performed 6 blocks of 50 trials (300 trials total).

In the decision window, participants performed a variant of the classic random dot kinematogram (RDK) motion direction discrimination task[37]. Participants were presented with an array of 100 white dots placed randomly within a circular aperture (4.8 degrees of visual angle), on a black background. Each frame, dot positions were updated according to angular coordinates such that a proportion of dots moved horizontally left or right (coherent direction), while the other dots moved in random directions. The stimulus was presented for approximately 350 ms. The participant was asked to decide whether the dots were moving left or right, and were given up to 1 s to enter their response with a button press. The proportion of dots moving in the coherent direction was first chosen based on a 2-down 1-up staircase procedure prior to the main experiment, and then decreased after each block proportionally to the improvement in performance in the previous block (relative increase in proportion correct from the first to the second half of the block), with two exceptions: (1) if proportion correct averaged over the entire block was greater than 0.7, coherence was reduced by ¼ of its current value; or (2) if proportion correct did not increase above 0.5, and was on average less than 0.55, coherence was increased by ½ of its current value).

In the bet window, participants were cued with the text 'Bet?' and were given up to 1 s to press a button to bet that their response was correct, doubling the points gained for a correct decision (but also doubling the points lost for an incorrect decision). The absence of a button press within this time was taken as a 'no-bet' response.

In the feedback window participants were cued about what type of feedback they would receive. A cyan cue signalled they would receive explicit feedback (text showing +1 point for correct, −1 for incorrect, or +/−2 in case the participant bet). A magenta cue meant no explicit feedback would be provided, instead participants were shown a question mark cueing them to assess how many points they thought they earnt (where points were awarded in the same manner as explicit feedback trials). Participants were given instructions about this prior to the experiment, and all explicit feedback reflected the true points gained/lost on that trial.

If participants failed to respond within 1 s of stimulus offset in the decision window, they were shown the text 'Time out!' after a 4–6 s interval (the duration of the bet and feedback windows) and lost 1 point before starting the next trial.

## Pre-processing

**EEG.** EEG pre-processing was performed using EEGLAB[72] and custom scripts[40,73–75] implemented in MATLAB (MathWorks Inc). MR gradient artefacts were removed by subtracting a drifting template (average over 80 TRs centred on each TR). Then, a 12 ms median filter was applied, data were downsampled to 1000 Hz, and bandpass filtered between 0.5 and 40 Hz. Blink artefacts were removed by extracting the first principal component from an eye-calibration routine where the participant was instructed to blink, before starting the scanner. Ballistocardiogram (BCG) artefacts were removed by projecting out the principal component closest to a BCG template (created using previous data with the same materials[75]). BCG principal components were extracted from the data low-pass filtered at 4 Hz (the frequency range where BCG artefacts are mainly observed). Since the BCG shares frequency content with the EEG, we adopted this conservative approach to minimise loss of signal power in the underlying EEG signal, where our multivariate discriminant analysis (see below) likely relies on components orthogonal to the BCG artefact. Data were finally re-referenced to the average. No baseline was applied (due to the long duration between time-windows).

**fMRI.** fMRI analyses were conducted using FSL software[76] (FMRIB, fmrib.ox.ac.uk/fsl). The Brain Extraction Tool[77] (BET) was used for brain extraction of the structural images and local field maps. The first 5 volumes of each functional run were discarded. Functional images were slice-time corrected, high-pass filtered at 50 Hz, spatially smoothed (8 mm full-width half maximum Gaussian kernel), and unwarped using the field maps (using FEAT[78,79]). Motion correction (using MCFLIRT[80]) was performed with parameters saved for later use as nuisance regressors in the GLM analysis. A two-stage registration aligned functional to structural with boundary-based registration, and structural to standard Montreal Neurological Institute (MNI) space with a 12 DOF, nonlinear search.

## Analysis

**Behaviour.** Proportion correct was calculated as the proportion of trials where the observer responded with the true stimulus direction, excluding timed out trials. Statistics were computed on sensitivity ($d'$; the difference in normalised hit and false-alarm rates). Reaction times were calculated from stimulus offset. Response perseveration was calculated as the probability of repeating a response given a stimulus repeat or stimulus alternation in each feedback condition. To account for different base rates of repetition, this probability was normalised and divided by the overall normalised probability of repeating a response to a repeating/alternating stimulus of each participant. Statistics ($2 \times 2$ repeated measures ANOVA for the explicit feedback conditions, t-test for bet/no-bet on no-feedback trials) were computed on the difference between stimulus repeat and stimulus alternate scores.

**Computational model.** To support the behavioural results indicating participants use confidence to learn, we used a simple computational model comparison. The models assume the basic framework of Signal Detection Theory[38] (Supplementary Fig. S1). On each trial the observer has a sample of noisy perceptual evidence and must decide if this evidence resulted from the presentation of a leftward or rightward

stimulus. We assume the noise affecting the perceptual evidence is Gaussian distributed with unit variance ($\sigma = 1$), added to the different mean perceptual evidence from leftward and rightward stimuli ($\mu_L$, $\mu_R$). To decide, the observer compares the perceptual evidence to a criterion, $c$, above which the observer responds that the perceptual evidence resulted from a rightward stimulus. Incorrect responses result from the overlap in the distributions, where the noise affecting the perceptual evidence can push the evidence to the wrong side of the criterion. Sensitivity, $d'$, is defined as:

$$d' = \frac{|\mu_R - \mu_L|}{\sigma} \quad (1)$$

Learning improves sensitivity. Previous work has addressed how adjusting the weighted integration of neural activity can improve the signal-to-noise ratio of the perceptual evidence[2,3], thus improving sensitivity. We implement this in our model by moving the means ($\mu_L$, $\mu_R$) further from the criterion. For simplicity, we fix the criterion at 0 and use a parameter, $\mu_0$, to define the starting values of the mean perceptual evidence, such that $\mu_L = -\mu_0$, $\mu_R = \mu_0$, and $d' = 2\mu_0$. A second free parameter, $b$, determines bet decisions by operating as a secondary criterion; the observer bets when the absolute value of the perceptual evidence exceeds this criterion.

A change in the stimulus coherence ($coh_{old} \rightarrow coh_{new}$) has a systematic effect on the mean evidence:

$$\mu_{new} = \mu_{old} \bullet \frac{coh_{new}}{\sqrt{2\sigma^2 coh_{old}}} \quad (2)$$

Without learning, $d'$ would decrease with decreased coherence. Instead, learning shifts the means of the distributions of perceptual evidence further from the criterion (balancing the shift back toward the criterion with decreased coherence). We implement these shifts with learning in accordance with reinforcement learning frameworks (as in ref. [13]): A third free parameter, the learning rate, $\alpha$, moderates the influence of a trial-wise learning signal, $L_t$, to update the means on each trial:

$$|\mu_{t+1}| = |\mu_t| + \alpha L_t \quad (3)$$

If $L_t$ is positive, the mean corresponding to the chosen stimulus is updated, otherwise both means are updated (moved away from the criterion, 0; see Supplementary Fig. S1). We compared how this learning signal incorporated confidence across three models. The basis of the learning signal is the reward prediction error, the difference between the explicit value of the feedback on that trial, $r_t$, and the expected value on that trial, $E[V_t]$. The expected value started at 1 (expecting to be correct) and was updated according to a Rescorla-Wagner rule[81] using the same learning rate as Eq. [3]. Model 1 did not incorporate confidence, the reward prediction error was computed on explicit feedback trials, and set to 0 on no-feedback trials. Model 2 used this same reward prediction error on explicit feedback trials, but used confidence as the learning signal on no-feedback trials. Model 3 used confidence on all trials: on explicit feedback trials, the expected value was moderated by confidence; on no-feedback trials confidence was used as the learning signal as in Model 2. In all cases, if the participant bet on that trial, the expected value was increased by 1 (in line with the points earnt).

Trial-wise confidence estimates were extracted based on the expected value of the perceptual evidence on each trial. Confidence was computed according to an ideal observer model[82] as the probability of a correct response given the perceptual evidence (based on the cumulative density of the joint distribution of evidence).

All models had three free parameters: The initial mean perceptual evidence, $\mu_0$, the criterion for betting, $b$, and the learning rate, $\alpha$. The models were fit to minimise the negative log likelihood of the

participant's perceptual decisions and bet responses on each trial. The log likelihoods were calculated from the cumulative Gaussian probability density corresponding to the choice and confidence (demarked by the criteria), and the model was fit using a constrained nonlinear interior point optimisation algorithm implemented with MATLABs fmincon function. Model and parameter recovery analyses are presented in Supplementary Fig. S1.

We note that these simple models do not capture all behavioural patterns in the data, for example, all three models failed to produce as large an interaction between feedback value and feedback sign as in the data. But the purpose of this modelling exercise was merely to support the behavioural results suggesting participants were using their confidence to learn: The largest difference between the models is on which trials confidence influences learning. These models were not designed to maximally capture behaviour. Instead, we focus on the neural mechanisms of learning from confidence, leaving computational model development to future work.

**EEG.** We used a linear discriminant analysis (LDA) to select spatial filters (linear channel weights) that maximally discriminate between bet and no-bet trials, and positive and negative feedback trials (two separate analyses). In each analysis, the LDA was trained on data from a sliding 60 ms window (10 ms intervals) across epochs locked to the time of the response (decision-window) or the time of feedback (feedback-window) using an iterative recursive least squares algorithm for linear logistic regression[83,84]. The sum of the product of the trained weights ($w$, spatial filter) by the multichannel activity ($x$) at trial $t$ gives a continuous prediction ($y$) of the binary variable to be discriminated:

$$y(t) = w^T x(t) = \sum_{i=1}^{D} w_i x_i(t) \quad (4)$$

In this way, $y(t)$ collapses the multichannel data into an aggregate representation that preserves single-trial information while providing a greater signal-to-noise ratio than individual channel activity[85]. Discrimination sensitivity was evaluated using the area under the receiver operating curve (Az) based on predictions ($y(t)$) from a leave-one-out cross validation. We generated a robust single spatial filter for each participant, in each of the decision- and feedback-windows, by taking the time-point with the greatest discrimination sensitivity (Az) over a 9-point moving average within the group-level significant time-window (evaluated using a one-sided t-test), and averaging the 5 spatial filters centred on that selected time-point. This procedure is visual depicted in Supplementary Fig. S2. The spatial filter can be visualised by taking the scalp projections of the discriminating components (the forward model; insets of Fig. [2]a, b):

$$a = \frac{Xy}{y^T y} \quad (5)$$

Where $X$ is the matrix of channel activity and $y$ the vector of predictions. This single spatial filter (for each participant) was then used to generate predictions over time, in both the decision- and feedback-windows. Differences between conditions were tested at each time-point using within-subject t-tests (bet vs no-bet) or ANOVAs (feedback 2 sign × 2 value) at an alpha level of 0.05, with cluster correction applied across time[86].

**fMRI.** We used an EEG-informed GLM analysis with eight explanatory variables and seven confound variables (the six motion correction parameters, and a custom variable coding for excluded trials and mid-block breaks). Explanatory variables were modelled as boxcar functions convolved with a normalised double-gamma probability density function. The eight explanatory variables were temporally positioned: (1) at stimulus onset, (2) the participant-specific time-point selected for the bet-prediction spatial filter relative to response time, (3) the participant

specific time-point where the feedback-prediction best generalised relative to response time, (4 and 5) the time of the bet-cue (for bet and no-bet trials separately), (6) locked to the feedback cue, (7) the participant-specific time-point selected for the feedback-prediction spatial filter relative to feedback time, and (8) the participant specific time-point where the bet-prediction best generalised relative to feedback. All variables were a boxcar function with a duration of 0.1 s, with the exception of the bet cue on bet trials (4; which was extended to the time of the bet-response), and the feedback cue (6; 1 s, the duration of the cue). The boxcar amplitude of the bet-cue on bet trials, the bet-cue on no-bet trials and the feedback-cue variables (4, 5, and 6) was set to 1. The boxcar amplitude of the stimulus onset variable (1) was modulated by the choice reaction time on each trial. The boxcar amplitudes for the remaining four variables were parametrically modulated by the EEG predictions (the bet-prediction in the decision-window, (2); the feedback-prediction in the decision-window, (3); the feedback-prediction in the feedback-window, (7); and the bet-prediction in the feedback-window (8)). These variables correspond to EEG-predictors 1–4 in Fig. 2c. The EEG feedback-prediction in the feedback-window was only placed on explicit feedback trials, and the bet-prediction in the feedback-window was only placed on no-feedback trials. The design of the eight explanatory variables is visually represented in Fig. 2c. The subject-wise maximum variance inflation factor ranged between 1.67 and 4.25. Note that though the EEG-predictors rely mainly on cortical activity (in close proximity to the sensors) to generate the trial-wise representations, the EEG-informed fMRI analysis can also reveal the contribution of deeper structures that covary systematically with these trial-wise representations. The full results of this analysis are presented in Supplementary Figs. S4–S11, alongside a control analysis using similar variables unmodulated by the EEG-predictors. This control analysis was conducted to be equivalent to the EEG-informed GLM but without the EEG: the bet-prediction in the decision-window was replaced with the binary bet/no-bet response; the feedback-prediction in the decision-window was removed as there is no behavioural equivalent for this; the feedback-prediction in the feedback-window was replaced by the explicit feedback itself ($-2/-1/1/2$); and the bet-prediction in the feedback-window was replaced again by the binary bet-response. In this way, this analysis reflects what we could have achieved without the EEG, for the interested reader.

Significant clusters were selected using a minimum z-statistic of 2.57 at the group level, and a minimum cluster size of 110 voxels, obtained using a permutation test (described previously[41,75]; the 95th percentile of the empirical null cluster size, calculated over 200 permutations of the EEG feedback-prediction variable (7) with otherwise the same GLM described above). ROI timeseries, epoched to the closest TR to the time of the decision or feedback, were generated by taking the average z-scored BOLD across ROI voxels after removing values greater than 3.1 standard deviations from the mean over three iterations. Correlations across ROI timeseries were calculated as the average Fisher transformed Pearson correlation across a time-window from 0 to 10 s in the decision-window and the feedback-window (or 0 to 10 s within the feedback-window). The decision- and feedback-windows were separated by 4 to 8 s.

We used vector autoregressive models of the full timeseries of GPe and IFG BOLD to test for Granger causality. For each participant, we first selected the appropriate lag in the model based on the Akaike Information Criterion (AIC). The median lag was 6 (12 s) ranging from 2 to 17 (4–34 s). All models were found to be stable. Leave-one-out Granger causal tests were conducted within subjects, with 18/23 participants showing significant evidence against the null hypothesis to exclude the history of IFG BOLD in predicting GPe BOLD (over and above the history of GPe BOLD itself). For participants failing to reject the null, the $p$-values ranged from 0.0599 to 0.461, for all other participants, $p < 0.027$ and median $p = 4.11e^{-7}$.

We also conducted a Psychophysiological Interaction (PPI) analysis[87] to examine how the GPe could be used to implement learning. As the physiological variable we took the time-course of the voxel average BOLD in the GPe ROI from the time of feedback to the start of the next trial. The psychological variable described the interaction between stimulus and response repetition that we took as a behavioural signature of learning: trials were coded as +1 for alternating responses to repeating stimuli following negative feedback (or no-bet responses on no-feedback trials), as well as trials with repeating responses to repeating stimuli following positive feedback (or bet responses on no-feedback trials); trials were coded as −1 for repeating a response to a repeated stimulus following negative feedback (or no-bet trials), as well as alternating responses to repeating stimuli following positive feedback (or bet trials). This can be summarised as a positive coding for trials contributing to the positive bars in Fig. 1f and a negative coding for trials contributing to the negative bars in Fig. 1f. For further demonstration, an example of the psychological, physiological, and interaction regressors are visually displayed with some annotation in Supplementary Fig. S13. Clusters of significant voxels were again selected with a statistical threshold of z >= 2.57 and minimum size of 110 voxels. Figure 3e shows the results of the analysis across both explicit feedback and no-feedback trials. We also compared the results of the analysis conducted separately on explicit feedback compared to no-feedback trials, taking a difference in z-statistic >2.57 as evidence for a difference in connectivity. We found no evidence for a difference within the significant regions of the analysis conducted across all trials. These results are presented in full in Supplementary Fig. S14.

## Reporting summary

Further information on research design is available in the Nature Portfolio Reporting Summary linked to this article.

## Data availability

Raw behavioural data, pre-processed EEG, and fMRI Z-statistic maps are available on the Open Science Framework: https://osf.io/q29uf/. Source data are provided with this paper.

## Code availability

Analysis code is available on the Open Science Framework: https://osf.io/q29uf/.

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

## Acknowledgements

This work was supported by the European Research Council (865003; M.G.P.) and in part by the Economic and Social Research Council (ES/L012995/1; M.G.P.). We thank Ralitsa Kostova for assistance with data collection.

## Author contributions

Conceptualisation: M.A.P. and M.G.P.; methodology: M.A.P. and M.G.P.; software: T.B., M.A.P. and M.G.P.; formal analysis: T.B.; investigation: M.A.P.; resources: M.G.P.; data curation: T.B.; writing—original draft: T.B.; writing—review & editing: T.B., M.A.P. and M.G.P.; visualisation: T.B. and M.G.P.; supervision: M.G.P.; project administration: M.G.P.; funding acquisition: M.G.P.

## Competing interests

The authors declare no competing interests.
