## [Peer Review File · Nature Communications]

Distinct basal ganglia contributions to learning from implicit and explicit value signals in perceptual decision-makingEditorial Note: Parts of this Peer Review File have been redacted as indicated to maintain the confidentiality of personal communication.

REVIEWER COMMENTS

Reviewer #1 (Remarks to the Author):

This manuscript by Balsdon and colleagues quantitatively examines how confidence is used for learning both under the presence and absence of explicit feedback. The authors first demonstrate through behavioral results that when asked to discriminate visual motion and before any explicit feedback was given, participants' bet behaviors of whether their discrimination choice was correct or not are indicative of decision confidence. The authors continue to extract neural signatures pertaining to four distinct processes in bet- and feedback-prediction: post-decision confidence, implicit outcome value, explicit outcome value, and expected outcome. Using simultaneous EEG-fMRI, the authors show that the neural processes, especially the post-decision confidence and implicit outcome value related confidence, can be delineated with both high temporal and spatial resolution.

Overall I find this manuscript interesting and persuasively written. The paradigm was neatly designed and well suited to identify decision-making processes associated with implicit/explicit outcome. The study is also a great demonstration of the complementary power of simultaneous EEG-fMRI in unraveling spatiotemporal neural dynamics.

That said, I find the difference between EEG-informed vs. standalone fMRI analysis results worthy of discussing. For instance, in Supplementary Fig.10, EEG-informed fMRI analysis located a left dorsal striatum cluster which was absent in the standalone fMRI analysis. I'm assuming that the authors are showing these brain images in the Right/Left orientation convention in all supplementary figures (please specify the orientation in the future however), then this left-lateralized cluster is shown in the rightmost MNI brain on the first row in the "EEG-informed fMRI analysis" panel. Meanwhile, on the second row underneath the "Standalone fMRI version" panel, a huge cluster situated in the basal ganglia (most likely the caudate) is visible from the 4th to the 7th MNI brain, and also left-lateralized. Given that the caudate is also associated with learning under uncertainty (e.g., Chiu et al., 2017, JNeuro; Doi et al., 2020, eLife), how would this alter the authors' statement that clusters associated with implicit outcome value "would not have been identified when contrasting the binary behavioral bet responses in a standalone fMRI analysis"? (line 357)

In addition, for the EEG analysis results presented in Supplementary Fig.2 - the Az value for the decoder of bet vs. no-bet trials were barely above 0.5 (top-left subplot), whereas the Az value for the decoder of positive vs. negative feedback trials had a period where it went below 0.5 (-1 to 0s before feedback in

the top-right subplot). Can the authors elaborate why the classifier had a barely-above-chance performance for the bet/no-bet trials, and a worse than random guess performance for the positive/negative feedback trials?

Lastly, I suggest to include equations specifying the confidence model for clarification (could be placed as an inset in Fig.1 if word limit is a concern). I also suggest to specify what “no-bet” trials mean early in the main text, instead of only explaining it in the Methods (line 447-8).

Refs

Chiu, Y. C., Jiang, J., & Egnér, T. (2017). The caudate nucleus mediates learning of stimulus–control state associations. *Journal of Neuroscience*, 37(4), 1028-1038.

Takahiro Doi, Yunshu Fan, Joshua I Gold, Long Ding (2020) The caudate nucleus contributes causally to decisions that balance reward and uncertain visual information *eLife* 9:e56694.

Reviewer #2 (Remarks to the Author):

This paper used simultaneous EEG and fMRI to investigate neural correlates of confidence and how confidence facilitates learning in contrast to explicit response feedback. They found that decision confidence is similar to explicit feedback and that neural correlates for implicit and explicit feedback come from the striatum and go back into the external globus pallidus for integration. Moreover, the striatum has a spatial gradient for implicit and explicit feedback pathways.

The analysis of simultaneous EEG and fMRI was advanced and impressive. EEG was first analyzed to identify the most significant time intervals to dissociate conditions. Then, the authors analyzed fMRI during these critical time intervals. Thus, the results are in high temporal and spatial resolutions.

Unfortunately, the data does not support the interpretation. This is due to unclear concepts or constructs, making the experimental design and results very difficult to follow, leaving a gap in logic here and there.

First, it needs to be clarified what the target of the investigation is conceptually and operationally. According to the abstract, the authors investigated the neural correlates of metacognitive evaluation of

confidence. Also, the term "implicit feedback" appears often in the paper and is used interchangeably with confidence. I do not think they are the same. There are no operational definitions of them.

The EEG decoder that decodes whether participants would bet is also used to investigate confidence. However, I do not think betting is a good measure of confidence. This is because betting is associated with risk aversion. Yes, it may be OK to consider that the participants are confident when they bet. However, I do not think it is appropriate to say that they were not confident when they did not choose to bet. The paper argues that betting is a manifestation of being confident because they are more accurate when they bet. If this argument holds, the paper should use the participant's responses (correct or incorrect) as the measure of confidence. The paper could have used a rating to measure subjective confidence. How confident they were when they did not bet is unclear in this design.

Next, the paper addresses how humans learn from implicit and explicit value signals. However, it needs to be clarified what participants learn. I thought the paper targeted learning in the motion task, but this was different. Perseveration (which I take as the relation between the previous feedback and the next participant's response, though the definition was unclear) seems to be a measure of learning. In my mind, learning should be defined as improved performance after training or the process in which this happens. However, no analysis exists to test whether the degree of perseveration changes over blocks. On the other hand, there was learning in the motion task, as the coherence changed over time. Then, the next concern arises: improved perception may change confidence or at least interact with confidence. However, there is no effort to dissociate learning of motion direction from confidence.

The perseveration measure could have been clearer. I figure the perseveration is the probability that the participants repeat the same response, depending on the feedback. However, since the feedback and no-feedback trials were intermixed, it is unclear whether the previous trial had the same feedback condition as the current trial. Even if they could separate the two feedback conditions, there would be some influence from other feedback conditions. It was unclear why the authors intermixed these two feedback conditions or why they did not test the two feedback conditions independently.

Technically, the analysis of combined EEG and fMRI is impressive. However, it needs to be clarified what this impressive technique shows conceptually. The paper shows the EEG decoder that decodes whether the participants bet a few seconds later. However, this decoder was made individually; thus, the EEG channels used, and the timing of EEG selected may differ across participants. Then, the underlying neural mechanisms contributing to the decoder could be different. In addition, the motor readiness potential may have been contaminated in the decoder since participants were not required to move their fingers to respond when they chose not to bet. The same criticism applies to other results in other time windows where the same bet-vs-no-bet decoder was applied. It is possible that the decoder only decodes whether finger movements occur or not, not the confidence.

Following is an example of a gap in logic (Lines 155-156), making the sentence difficult to follow: "In the absence of explicit feedback, participants did use their confidence to infer the points they may have gained/lost." Even though the bet-vs-no-bet EEG decoder successfully decoded the EEG in the feedback-window in the no-feedback trials, no other data show (1) that they used their confidence in the feedback-window and (2) were inferring the possible points.

The EEG results take a lot of work to follow. Figure 2 has five subpanels, which are intricated and not independent. The borders between subpanels could be more explicit. The captions may not be accurate, as Fig.2E caption explains what the blue and yellow color-coded areas mean in the feedback-window. Still, this panel contains other colors used for the decision-window.

The following fMRI analyses are also difficult to follow in Figure 3. Fig. 3B caption shows correlations between GPe & dStr and GPe and vStr. However, what is shown in the horizontal axis in Fig. 3B needs to be clarified. According to the paper, the feedback-window showed the points participants would get. I understand that the possible points are shown in the horizontal axis. But then, what are the no-bet and bet categories in the horizontal axis? Betting was done in the bet-window, different from the feedback-window. Fig. 3D is supposed to show the resemblance between the brain activations in the decision- and feedback-windows, where the correlation coefficients demonstrate the resemblance. However, the correlation coefficient is extremely small, nearly zero, in each ROI. Thus, these data do not support the claim that "earlier BOLD related to confidence in the IFG continues to drive GPe BOLD responses following feedback."

Reviewer #3 (Remarks to the Author):

In this manuscript, Balsdon and colleagues show how implicit and explicit confidence signals are represented in the human brain, either following the decision or later, during a time-window in which feedback on performance was provided in half of the trials. During the task, participants responded to a perceptual task (decision window), betted on their performance, then got feedback in half of the trials (feedback window).

First, the authors use a computational model of reinforcement learning to show that the behavior was better reproduced under the assumption that confidence was used as implicit feedback during learning (compared to a model based only on explicit feedback). Second, the authors trained an EEG decoder to predict the outcome of the bet during the decision window. They found that during the feedback window, the output of the same decoder differed according to the feedback outcome when feedback was given and to the bet outcome in the absence of feedback. The later was considered to be a proxy to implicit outcome value. Conversely, they trained a decoder on the feedback outcome during the

feedback window, which they consider to be a proxy to explicit feedback. They found that during the decision window, the output of this decoder differed according to the bet outcome. Third, the authors describe the hemodynamic correlates of the output of these decoders and find that they differ for implicit and explicit feedback. Finally, based on an overlap between correlates of implicit and explicit feedback in the GPe, the authors performed connectivity analyses pointing to an additional involvement of the thalamus, the insula and the rostromedial prefrontal cortex. Together their results provide important empirical evidence on the way that confidence is used as an implicit feedback for learning.

This is a very strong and elegant paper. The study will give a new breath to the field of metaperception which has mainly focused on the monitoring aspect and less on control, or the way that meta-representations can be helpful for adaptive behavior. In terms of methods, combining computational modeling, EEG and fMRI is probably the best that can be achieved non-invasively in human cognitive science and the added benefit of EEG is very clear (+special congrats to the authors that had to prepare the EEG cap before the scanner...). Although the analyses are quite complex, it was a pleasure to read because it is clearly written and very nicely illustrated. To highlight just one example, I liked the way that Fig. 2 shows the EEG data and explains which signals relate to which BOLD activations with a number for each regressor that can be traced back to the main text. I have some points, mainly clarifications that I'd like the authors to consider/answer but I think this manuscript definitely deserves to be published here. Plus the data and code are open!

Signed: Michael Pereira

#1. About "integration".

I would like the authors to explicit what they mean by "integration" which seems to be a central claim to the paper (e.g. lines 10, 65, 344, 377). Do you mean that a learning signal is constructed by reading out either explicit or implicit feedback? Then the term integration would not seem appropriate.

I presume that the authors mean that additionally, in explicit feedback trials, both types of feedback are integrated. For this stronger version, it seems that the authors cannot solely rely on the fact that BOLD activity in the GPe was related to both implicit and explicit outcome value signals since these signals are measured in different conditions. I understand that the Granger connectivity analysis supports this strong version. However, although the result section does make a decent job in constructing this argument, I think it would be nice to explicitly tie these different findings together in the discussion. The sentence "Our analysis further suggests these implicit and explicit representations of outcome value are integrated in the external globus pallidus, which corresponded both to post-decision confidence in the decision-window and explicit outcome value in the feedback-window." wasn't completely clear to me.

Maybe the authors could make a link between the decoded EEG signals used as parametric regressors, interpreted as explicit or implicit outcome (line 205) and $E(V_t)$ in the computational model, in order to provide a computational interpretation of this integration? Is there any possibility of demonstrating integration at the behavioral level by comparing a model that uses both confidence and feedback in feedback trials, and a model that still uses confidence in no-feedback trials but updates only based on explicit feedback in feedback trials?

#2. Errors

Participants have higher d' when they bet, therefore, they err more when they don't.

Is there a way for the authors to confirm that information about the outcome of the bet is present in the EEG irrespective of error detection? For example, does the trained decoder predict the outcome of the bet above chance for correct responses only? This is not a serious issue since we don't really know to what extent error monitoring and metacognitive monitoring differ in terms of mechanisms but I think it is worth considering. For example, with confidence, it's nice to show that confidence levels can still be decoded when taking only correct trials.

#3. Positive BOLD correlations

I was surprised to see that Fig. 2D shows a "positive relation to the EEG bet-prediction" (line 225). I am assuming that positive bet prediction should be interpreted as positive confidence. But previous works (including ours) found that BOLD in pMFC, IFG, SPL and co was negatively correlated to both "behavioral" confidence or an EEG proxy to confidence (Rouault et al., 2018; Morales et al., 2020; Pereira et al., 2020, including the fMRI analysis in Gherman & Philiastides, 2018). Is it a typo or did I miss something? If the relationship is really that way (i.e. high bet prediction \rightarrow high activation of these regions), is there an explanation beyond the trivial ones (different tasks etc...)? Can we really say that it's "in line with previous literature" (line 353)? Wouldn't we expect a positive relationship more in the ventral striatum and vmPFC?

Relatedly –and I admit this is not a very original question– were there no other BOLD activations in the "other directions"?

#4. Methodology

While I'm at boring questions, are the bet and feedback predictions sufficiently decorrelated to be included as separate parametric regressors during the decision window? And is there a hierarchy in the way the regressors are treated (e.g. is the feedback predictor applied on the residuals after variance related to the bet predictor is regressed out)?

Also, the EEG decoder for explicit outcome is trained on the same window while the EEG decoder for implicit feedback is trained on a different window. Do you think this an issue for the interpretation of the BOLD correlate?

Minor:

line 50: is ref 28 really about the striatum and confidence.

lines 98 - 102: I'm afraid I did not get that sentence

Lines 202 - 205: You just focus on BOLD activity related to parametric regressors of EEG, right? At first I understood "BOLD related uniquely to the contribution of the EEG-prediction" as meaning that a region that would be activated both by parametric regressors and another task regressor would be discarded. If that were the case I would fail to get the rationale.

Fig. 2: Was there any baselining of the EEG signal? How do the authors explain the pre-feedback activity?

lines 439: 1-up 2-down, rather?

Methods: Would be nice to have a bit more info about the fitting procedure: how was the log likelihood computed, what optimization algorithm? (Unless I missed it).

Reviewer #4 (Remarks to the Author):

In this manuscript, Baldson and colleagues investigate learning from implicit and explicit value signals in perceptual decision-making, in a small cohort of healthy human subjects ($N = 23$). The perceptual is a random-dot motion task (binary choice), followed by a metacognitive evaluation (bet or not bet on the answer) and by a feedback or absence thereof. The authors use simultaneous EEG-fMRI: EEG signal is used to decode a variety of variables: expected outcome, decision to bet and (expected) feedback. EEG predictions of those variables is then regressed on whole-brain BOLD signal. The authors then attempt to

build a dynamic model of decision-making in the basal ganglia by leveraging a variety of functional connectivity analyses.

Overall, the manuscript is very dense: it constitutes a very brave endeavor, which led to the collection of a very rich and valuable simultaneous EEG-fMRI dataset. I also commend the authors for implementing an impressive collection of sophisticated methods, and for attempting to extract as much information about the decision-mechanism from their data as they possibly could

However these positive aspects often come at the cost of the clarity of the conclusion and the way they are exposed. I also have two major concerns about the analyses and their interpretations that would need to be addressed before I can recommend this manuscript for publications.

One of my main concern is with the EEG decoding of variables. Basically, the authors show that they can decode a set of variables from the EEG signal (e.g. bet vs no-bet at decision; lines 144-145), then show that this variable can be decoded at another time point (E.g. feedback; lines 151-156) and then ascribe a new computational meaning to it (participants infer the outcome; line 156).

The problem is, as the authors note elsewhere, all variables (decision to bet, expected reward, actual feedback, inferred feedback) are all highly correlated, and are also correlated with other latent variables (notably, the inferred probability of being correct should underpin all those estimations) as well as behavioral output (such as the decision reaction time, as noted line 146, which should basically correlate with everything).

Put simply, my take is that if the authors want to ascribe some specific EEG signals to some specific (latent) variables, they should estimate those variables from the behavior (choice, RT), stimulus (strength, direction, etc.) and model (supp mat), and properly test if and how EEG signal better correlate with some variables than others, at specific time-points.

Without those analyses, I am not convinced that the EEG decoding exercise can really inform us about the actual computations that are performed, given the potential correlation between all variables.

In a similar vein, the authors show that the EEG-informed analysis of fMRI is better than a model-free analysis of fMRI (Supplementary Figures 3-10) but I doubt that the EEG-informed analysis of fMRI signal can actually perform better than a classical model-based fMRI analysis, especially if the goal is to ascribe computational variables to some brain signals/regions.

My second major concern the last section of the result, regarding the building of the decision/learning neuro-computational model within the basal ganglia, using functional connectivity analyses (lines 264-336). Once again, this section looks very impressive, and the authors really put tremendous effort to combine a lot of sophisticated analysis tools (Pearson correlation functional connectivity, vector

autoregressive models of BOLD timeseries together with leave-one-out Granger causal tests, Psychophysiological interaction analysis).

But the strength of the interpretations and conclusions of these analyses remain unclear to me... because the hard truth is that all those analyses rely on the BOLD signal that is basically pictured in Figure 3C: noisy, lagged BOLD signal, from small samples ($N = 23$), in a task that is much more fast-paced than the hemodynamic signal.

In this respect, I find a bit worrying the BOLD time-courses, which all seem to peak -2 to +5 sec around feedback, when they should be elicited by the said feedback. Therefore, I sincerely think there is a significant risk in trying to extract too much out of this.

In my opinion, the paper would still be more than interesting enough with a more in-depth model-based analysis of the first section on the variables mapping – i.e. without this last “connectivity” section, which might have difficulty to stand the test of time and replications

Minor

As I said in my general evaluation, I found this manuscript quite hard to read, because sometimes too dense, too technical and maybe too ambitious in the breadth of the analyses exposed. Some statistical details span over more than 4-line parentheses, which completely break the flow of the demonstration. I suggest that the authors invest some time in streamlining their demonstration, to ease the reading.

I was sometime a bit shocked by how assertive some interpretations and conclusions were, notably about variable-mappings and mechanisms that are, in fine, not unambiguously demonstrated (e.g., from the abstract: “These two signals are then integrated into an aggregate representation in the external globus pallidus, which broadcasts updates to improve cortical decision processing via the thalamus and insular cortex, irrespective of the source of feedback”). I suggest to tone down, and make a better effort at acknowledging the potential limitations of the conclusions

We thank you for the opportunity to revise our manuscript (NCOMMS-23-42311) titled “Distinct basal ganglia contributions to learning from implicit and explicit value signals in perceptual decision-making” for *Nature Communications*. We were grateful to receive such insightful and constructive comments from the reviewers which have helped us make major improvements to the manuscript. To ensure we have fully addressed all comments, we have numbered each comment according to reviewer number and comment number, such that R2C1 corresponds to the first comment of Reviewer 2. We provide a track-changed document with all changes commented with this code. In responding to reviewers, we quote the text, referring to the page and line number of the clean (not tracked-changed) revised manuscript, should they wish to find the changes there.

We summarise the more major changes here:

- Clarified the definitions of implicit feedback (R2C1) and the integration of confidence with feedback (R3C1a) in the introduction, as well as clarifying the term ‘no-bet trial’ (R1C3a) and the participant instructions (R2C6) at the beginning of the results.
- We have separated out the analyses in the results further, and re-written large sections to be clear about their purpose (Reviewer 2) and make the manuscript less dense (Reviewer 4).
- We have added a new computational model to our model comparison analysis which helps support the claim that confidence is integrated for learning even when explicit feedback is provided (with thanks to Reviewer 3). In addition, we have followed the suggestion of Reviewer 3 to examine how the EEG-predicted outcome value signals predict behaviour in this model context. This helped to address the comment of Reviewer 4 (R4C1) to better validate the EEG-predictions with behaviour.
- We have clarified the discussion of the relationship between the EEG-informed and stand-alone fMRI analysis, in relation to comments from Reviewers 1, 2, and 4.
- We have weakened the weight on, and emphasised the exploratory nature of, the connectivity analysis, in response to Reviewer 4 (R4C3) and with thanks to the suggestions of Reviewer 3.
- We have weakened claims in the discussion, which we agree with Reviewer 4 could come across as too strong.
- We have clarified Supplementary Figure S1 and added a new Supplementary Figure S11.

Reviewer #1 (Remarks to the Author):

This manuscript by Balsdon and colleagues quantitatively examines how confidence is used for learning both under the presence and absence of explicit feedback. The authors first demonstrate through behavioral results that when asked to discriminate visual motion and before any explicit feedback was given, participants’ bet behaviors of whether their discrimination choice was correct or not are indicative of decision confidence. The authors continue to extract neural signatures pertaining to four distinct processes in bet- and feedback-prediction: post-decision confidence, implicit outcome value, explicit outcome value, and expected outcome. Using simultaneous EEG-fMRI, the authors show that the neural processes, especially the post-decision confidence and implicit outcome value related confidence, can be delineated with both high temporal and spatial resolution.

Overall I find this manuscript interesting and persuasively written. The paradigm was neatly designed and well suited to identify decision-making processes associated with implicit/explicit outcome. The study is also a great demonstration of the complementary power of simultaneous EEG-fMRI in unraveling spatiotemporal neural dynamics.

We thank the reviewer for their constructive comments, which have been very helpful in improving the manuscript. To ensure we have fully addressed all comments, we have numbered each comment according to reviewer number and comment number, such that R2C1 corresponds to the first comment of Reviewer 2. We provide a track-changed document with all changes commented with this code. Page and Line numbers of these changes refer to the clean revised manuscript (not track-changed), and use a similar code: P15L376 refers to page 15, line 376.

RIC1

That said, I find the difference between EEG-informed vs. standalone fMRI analysis results worthy of discussing. For instance, in Supplementary Fig.10, EEG-informed fMRI analysis located a left dorsal striatum cluster which was absent in the standalone fMRI analysis. I’m assuming that the authors are showing these brain images in the

Right/Left orientation convention in all supplementary figures (please specify the orientation in the future however), then this left-lateralized cluster is shown in the rightmost MNI brain on the first row in the “EEG-informed fMRI analysis” panel. Meanwhile, on the second row underneath the “Standalone fMRI version” panel, a huge cluster situated in the basal ganglia (most likely the caudate) is visible from the 4th to the 7th MNI brain, and also left-lateralized. Given that the caudate is also associated with learning under uncertainty (e.g., Chiu et al., 2017, *JNeuro*; Doi et al., 2020, *eLife*), how would this alter the authors’ statement that clusters associated with implicit outcome value “would not have been identified when contrasting the binary behavioral bet responses in a standalone fMRI analysis”? (line 357)

We thank the reviewer for highlighting this important point. We agree that the difference between the EEG-informed vs the stand-alone fMRI analysis deserves greater attention, especially the caudate cluster in the stand-alone analysis. We have clarified the discussion of the EEG-informed analysis in the results, also in response to Reviewer 4 (R4C2) and to help address a comment from Reviewer 2 (R2C5a). Specifically, we discuss the differences from the stand-alone analysis in a dedicated paragraph, highlighting where we also see less strong BOLD correlates and weakening the emphasis on the benefits of the EEG-informed analysis that could be interpreted as a claim that they are ‘better’. The paragraph now reads (P11L292):

“Of note, this EEG-informed analysis produces some key differences from the binary behavioural variables that would have been used in a stand-alone fMRI analysis (see **Supplementary Figures 3-10**). In particular, the binary bet vs no-bet comparison in the decision-window does not capture the relationship between post-decision confidence and the external globus pallidus. Similarly, the dorsal striatum cluster corresponding to the re-emergence of confidence as an implicit outcome value estimate in the feedback-window is absent for a comparison of binary bet vs no-bet trials in the feedback window. The stand-alone fMRI analysis does resolve a cluster in the caudate (which has previously been associated with learning under uncertainty; Chiu et al., 2017; Doi et al., 2020), however, this does not capture the gradient with explicit value in the striatum, which deserves further investigation. Had the endogenous trial-wise variability captured in the EEG-predictions resulted from unrelated noise, these variables would have been less powerful predictors in the fMRI GLM. The EEG-informed analysis also provides more specific results, for instance, the stand-alone feedback regressor results in a large cluster over occipital cortex that likely captures the difference in luminance of the feedback visual cues. This suggests that the EEG-predictions give a closer approximation of the graded subjective variables underlying behaviour, where this can reveal a richer picture of the BOLD correlates, especially in the basal ganglia.”

The quoted statement referred specifically to the dorsal striatum cluster that was not significant in the stand-alone analysis. However, we appreciate the reviewer’s point that we would have highlighted a caudate cluster from the stand-alone analysis had the experiment not included the simultaneous EEG measurements. We caution that this caudate cluster, though it does overlap with caudate, mainly covers white matter. In an effort to be transparent and show the broader extent of our z-maps, all the activations in the Supplementary Figures are raw, without cluster correction, white matter masking, and so forth. To avoid the potential ambiguity of the quoted sentence as the reviewer highlights, we have removed the sentence (the paragraph now begins on P15L406).

To further highlight some of the more nuanced comparisons between the EEG-informed and stand-alone analyses, we have added some comments to the Supplementary Figure legends to highlight important differences and similarities. For example, for Supplementary Figure 10 (implicit outcome value) we add:

“Here the EEG-informed analysis is much more specific, with three clusters surviving correction: right occipital, left middle frontal gyrus and left dorsal striatum. The left dorsal striatum cluster was not significant in the stand-alone analysis. The stand-alone analysis does show a cluster that overlaps with left caudate, but much of this cluster covers white matter.”

And for Supplementary Figure 9 (explicit outcome value):

“The difference in these two analyses is again suggestive of the more specific nature of the EEG-informed analysis. While frontal and subcortical regions typically associated with feedback and value emerge from both analyses, the occipital regions are much less prevalent in the EEG-informed analysis. The greater luminance for positive feedback (due to the + vs – sign on the screen) means that visual cortical activity does correspond to the stand-alone positive vs

negative feedback regressor, but is much less correlated with the graded EEG-informed regressor. That the frontal and subcortical regions remain strong is indicative of the fact that the graded EEG regressor does align with the endogenous variability in the graded neural representation of outcome value.”

(We have also added L/R labels to the figures, and thank the reviewer for pointing this out).

RIC2

In addition, for the EEG analysis results presented in Supplementary Fig.2 – the Az value for the decoder of bet vs. no-bet trials were barely above 0.5 (top-left subplot), whereas the Az value for the decoder of positive vs. negative feedback trials had a period where it went below 0.5 (-1 to 0s before feedback in the top-right subplot). Can the authors elaborate why the classifier had a barely-above-chance performance for the bet/no-bet trials, and a worse than random guess performance for the positive/negative feedback trials?

This is an important question, and we thank the reviewer for posing it. The reduced accuracy of the decoder in the bet-vs-no-bet discrimination is in part because EEG data are somewhat more noisy in the MRI scanner, but perhaps more critically due to the more subtle nature of the internal representations of confidence we are trying to exploit (some of which in subcortical structures). In contrast, explicit representations of outcome valence (as in our positive-vs-negative feedback discrimination) are much stronger due to the more proximal (to our EEG sensors) and distributed cortical nature of the underlying network. Ultimately, our decoder accuracy is in line with other work in the literature, for example, Desender et al., 2019 also show maximum confidence decoding accuracy less than .6 (Figure 5G); Krumpe et al., 2020 show decoding accuracy between .54 and .63 (Table 3); and our previous work showing a maximal average decoding accuracy for confidence around 0.57 for simultaneous EEG-fMRI (Figure 2A; Gherman & Philiastides, 2018). Similarly, the relative decoding accuracy for positive-vs-negative feedback discrimination, is consistent with previous work from our lab (and others), showing average decoding accuracy around 0.66 for simultaneous EEG-fMRI (Figure 1B; Fouragnan et al., 2015), which is virtually identical with what we report here.

We have now noted these previous study benchmarks in the figure legend (copied below).

On a more technical note, given this reduced decoder performance and to ensure that our results are statistically robust we normally use a permutations analysis to calculate a realistic interval around chance: we shuffle the labels and perform the same analysis, repeating 1000 times. One can then calculate 95% confidence on the maximum accuracy achievable by chance, and use this to calculate significance. We did this, but it resulted in a slightly more lenient threshold than the t-test. So, we went with the more conservative t-test and chose not to mention the permutations analysis for simplicity. But this question emphasises the importance of demonstrating the interval around chance. We have added this to the figures (shown below). You can see that feedback decoding performance does fall below the 95% confidence interval about 5 time points during the 1 second (1000 time points) prior to feedback. This is about what we would expect.

Supplementary Figure 2. Procedure for isolating subject-specific spatial filters related to representations of confidence and explicit feedback. For each subject, we ran a linear discriminant analysis (LDA) of bet vs no-bet trials in the decision-window and positive vs negative feedback in the feedback window. The analysis was run on data within a sliding window and assessed across time using the area under the ROC (Az) in a leave-one-out (loo) validation procedure. Notably, the performance of the decoder in bet vs no-bet trials is not as high as in positive vs negative feedback, but in both cases accuracy reaches levels well above chance and is in line with previous findings (Desender et al., 2019; Gherman & Philiastides, 2018; Fouragnan et al., 2015). We suggest the decoding accuracy of confidence is less than that of explicit feedback because the representation of explicit value is more distributed across cortical sources, and perhaps the implicit representation is less variable with respect to the external variable (for confidence there is more time in which the representation could evolve prior to the bet response). A permutations analysis was run to assess objective chance (grey shaded regions) by performing the analysis with shuffled labels 1000 times to determine the 95% confidence intervals. This resulted in a more liberal region of group-level significance than a one sided t-test, so we chose to use the t-test as the more conservative criterion. We assessed the group-level significant time window in each case (red shaded areas), and then took the best time-point for individual subjects from a 9-point moving average within this window (green line). The topographies show the resulting forward models (scalp projections of the spatial filter) at this time. The analysis in the Manuscript Figures 2A and 2B is performed by applying these spatial filters over time to generate the predicted value (bet-prediction, or feedback-prediction), for each trial, and then averaging across trials, taking either bet and no-bet trials separately, or explicit feedback trials separately.

Desender, K., Murphy, P., Boldt, A., Verguts, T., & Yeung, N. (2019). A postdecisional neural marker of confidence predicts information-seeking in decision-making. *Journal of Neuroscience*, 39(17), 3309-3319.

Krumpe, T., Gerjets, P., Rosenstiel, W., & Spüler, M. (2019). Decision confidence: EEG correlates of confidence in different phases of an old/new recognition task. *Brain-Computer Interfaces*, 6(4), 162-177.

Sadras, N., Sani, O. G., Ahmadipour, P., & Shanechi, M. M. (2023). Post-stimulus encoding of decision confidence in EEG: toward a brain-computer interface for decision making. *Journal of Neural Engineering*, 20(5), 056012.

R1C3a

Lastly, I suggest to include equations specifying the confidence model for clarification (could be placed as an inset in Fig.1 if word limit is a concern).

Thank you for this suggestion. In response to a suggestion from reviewer 3 (R3C1c) we have expanded the computational model comparison to include a third model that demonstrates behavioural evidence that confidence

is integrated with explicit feedback to inform the reward prediction error even when explicit feedback is provided. We have included the equations in this description (P5L115):

“We used a simple computational reinforcement learning model comparison to assess the use of confidence for learning. The models altered the internal response to the stimulus according to a learning rate (α) applied to the reward prediction error, that is, the difference between the explicit feedback value (r_t) and the expected value ($E[V_t]$). In this way, the mean response to a leftward stimulus, (μ_L) would be modified as: $\mu_{L_{t+1}} = \mu_{L_t} + \alpha(r_t - E[V_t])$. Three models were implemented to compare the use of confidence: 1) a model that does not use confidence (the expected value is updated according to a Rescorla-Wagner rule: $E[V_{t+1}] = \alpha_w(r_t - E[V_t])$); 2) a model that uses confidence only when explicit feedback is not provided (confidence, c_t , substitutes the reward prediction error, and is calculated as the cumulative probability of the perceptual evidence in terms of distance from the response criterion); 3) a model that also uses confidence on explicit feedback trials (the expected value is proportional to confidence, $E[V_t] = c_t + bet$). The model that used confidence on all trials (integrating confidence for the reward prediction error) provided the best description of behaviour (Figure 1C,F; $\sum BIC_3 - BIC_1 = -330.45$; $\sum BIC_3 - BIC_2 = -270.93$; exceedance probability > 0.99 ; see **Methods, Supplementary Figure 1**). This suggests confidence is integrated with explicit feedback to inform the reward prediction error and update processing for learning.”

R1C3b

I also suggest to specify what “no-bet” trials mean early in the main text, instead of only explaining it in the Methods (line 447-8).

We had not realised this was lacking, this will substantially improve clarity, thank you. We have added an explanation of the terminology at the beginning of the results section (P4L74):

“Each trial was composed of three time-windows (Figure 1A): The decision-window ... the bet-window (in which participants were given the opportunity to bet that their responses were correct, ‘bet trial’; or not, ‘no-bet trial’); and the feedback-window ...”

Refs

Chiu, Y. C., Jiang, J., & Egner, T. (2017). The caudate nucleus mediates learning of stimulus–control state associations. *Journal of Neuroscience*, 37(4), 1028-1038.

Takahiro Doi, Yunshu Fan, Joshua I Gold, Long Ding (2020) The caudate nucleus contributes causally to decisions that balance reward and uncertain visual information *eLife* 9:e56694.

Reviewer #2 (Remarks to the Author):

This paper used simultaneous EEG and fMRI to investigate neural correlates of confidence and how confidence facilitates learning in contrast to explicit response feedback. They found that decision confidence is similar to explicit feedback and that neural correlates for implicit and explicit feedback come from the striatum and go back into the external globus pallidus for integration. Moreover, the striatum has a spatial gradient for implicit and explicit feedback pathways.

The analysis of simultaneous EEG and fMRI was advanced and impressive. EEG was first analyzed to identify the most significant time intervals to dissociate conditions. Then, the authors analyzed fMRI during these critical time intervals. Thus, the results are in high temporal and spatial resolutions.

Unfortunately, the data does not support the interpretation. This is due to unclear concepts or constructs, making the experimental design and results very difficult to follow, leaving a gap in logic here and there.

We thank the reviewer for their careful critiques, which have helped us to improve the clarity of the manuscript. To ensure we have fully addressed all comments, we have numbered each comment according to reviewer number and

comment number, such that R2C1 corresponds to the first comment of Reviewer 2. We provide a track-changed document with all changes commented with this code. Page and Line numbers of these changes refer to the clean revised manuscript (not track-changed), and use a similar code: P15L376 refers to page 15, line 376.

R2C1

First, it needs to be clarified what the target of the investigation is conceptually and operationally. According to the abstract, the authors investigated the neural correlates of metacognitive evaluation of confidence. Also, the term "implicit feedback" appears often in the paper and is used interchangeably with confidence. I do not think they are the same. There are no operational definitions of them.

We agree with the reviewer that it is important this link between confidence and implicit feedback is clear, and thank them for highlighting the lack of clarity here. We do not want to suggest that confidence is the only source of implicit feedback, just an important one. Further, we do not want to suggest confidence is equivalent to implicit feedback. We think that confidence can be used as implicit feedback: if you feel confident that you were correct, you can use this to try to keep doing the task as you were doing, if you are not confident you were correct, you should try to change how you're doing the task to improve. We tried to introduce this concept of using confidence as implicit feedback in the introduction (P2L22):

“Confidence provides an estimate of the likelihood that a decision is correct [6,7], and as such could be used as an implicit proxy for explicit feedback.”

We have clarified this sentence:

“Confidence provides an estimate of the likelihood that a decision is correct [6,7], and so could be used as a proxy for explicit feedback, we refer to this proxy as ‘implicit feedback’.”

We also changed the sentence (P3L60):

“We use this design of intermixed trials with and without explicit feedback to directly compare the neural signatures of learning from explicit (outcome value) and implicit (confidence) feedback signals.”

To now read:

“We use this design of intermixed trials with and without explicit feedback to directly compare the neural signatures of learning from explicit (outcome value) and implicit feedback signals (such as confidence).”

This now more clearly highlights that we do not think confidence is the only source of implicit feedback, just an important one.

For perceptual decision-making, performance improvements from implicit and explicit feedback, have been computationally described by changes in how neurons (in this case those tuned to different directions of motion) are weighted for determining your decision (Law and Gold, 2009; Drugowitsch et al., 2019; and also the basis of how we implement learning in our model, Supplementary Figure 1). We have also now expanded the computational modelling section in response to a suggestion from Reviewer 3 (R3C1c), and made sure to add equations here, as suggested by reviewer 1 (R1C3a), so now the computational description of the use of confidence as implicit feedback should be more clear from the results section (P5L115):

“We used a simple computational reinforcement learning model comparison to assess the use of confidence for learning. The models altered the internal response to the stimulus according to a learning rate (α) applied to the reward prediction error, that is, the difference between the explicit feedback value (r_t) and the expected value ($E[V_t]$). In this way, the mean response to a leftward stimulus, (μ_L) would be modified as: $\mu_{L_{t+1}} = \mu_{L_t} + \alpha(r_t - E[V_t])$. Three models were implemented to compare the use of confidence: 1) a model that does not use confidence (the expected value is updated according to a Rescorla-Wagner rule: $E[V_{t+1}] = \alpha_w(r_t - E[V_t])$); 2) a model that uses confidence only when explicit feedback is not provided (confidence, c_t , substitutes the reward prediction error, and is calculated as the cumulative probability of the perceptual evidence in terms of distance from the response criterion); 3) a

model that also uses confidence on explicit feedback trials (the expected value is proportional to confidence, $E[V_t] = c_t + bet$). The model that used confidence on all trials (integrating confidence for the reward prediction error) provided the best description of behaviour (Figure 1C,F; $\sum BIC_3 - BIC_1 = -330.45$; $\sum BIC_3 - BIC_2 = -270.93$; exceedance probability > 0.99 ; see **Methods, Supplementary Figure 1**). This suggests confidence is integrated with explicit feedback to inform the reward prediction error and update processing for learning.”

R2C2

The EEG decoder that decodes whether participants would bet is also used to investigate confidence. However, I do not think betting is a good measure of confidence. This is because betting is associated with risk aversion. Yes, it may be OK to consider that the participants are confident when they bet. However, I do not think it is appropriate to say that they were not confident when they did not choose to bet. The paper argues that betting is a manifestation of being confident because they are more accurate when they bet. If this argument holds, the paper should use the participant's responses (correct or incorrect) as the measure of confidence. The paper could have used a rating to measure subjective confidence. How confident they were when they did not bet is unclear in this design.

We thank the reviewer for highlighting this important point, which we are sure other readers will sympathise with. There are several nuances to this comment which we address in turn.

- 1) “The EEG decoder that decodes whether participants would bet is also used to investigate confidence. However, I do not think betting is a good measure of confidence.”

As we outline more fully below (point 2), the use of betting has a long history of being used to study confidence. Here, we use bet vs no-bet responses to delineate trials in which the participant was more vs less confident in their decision, and use this to decode EEG representations of confidence in the earlier decision-window. We confirm that participants were indeed more confident when they bet using two stereotypical behavioural signatures of confidence (outlined more fully under point 3 below). Moreover, the topography of the EEG bet decoder is in line with previous studies using decoding analyses of confidence ratings (e.g. Desender et al., 2019), suggesting the decoding of bet responses is picking up on the same underlying confidence signals as would have resulted from using a rating procedure. In our previous work, we have also shown similar topography using both ratings and bets (German and Philiastides, 2015; 2018). This alignment of the topography is another indication that the decoder is picking up on representations of post-decision confidence. Importantly, this decoding analysis is performed long before (1 – 2 s prior to) the bet-window, where this temporal gap, and the temporal jitter, limits the ability of the decoder to pick up on representations of bet responses per se (outlined more fully in R2C5b). Rather, the analysis aligns with the proposed temporal order of using post-decision confidence to then inform whether to make a bet.

- 2) “However, I do not think betting is a good measure of confidence. This is because betting is associated with risk aversion... The paper could have used a rating to measure subjective confidence.”

Confidence is likely a graded internal representation, and the participant has to place a criterion for how confident they need to be to bet. This criterion is affected by risk aversion, which we did highlight in our original manuscript:

“Although the overall tendency to bet may partly depend on inter-individual differences in risk aversion...”

This means that two participants could feel equally confident overall, but one bets more often than the other. The reviewer suggests a confidence rating would have been a better behavioural measure of confidence. But confidence ratings suffer the same issue: the participant has to place (multiple) criteria to determine which range of graded internal confidence should correspond to each rating response. Participants’ tendency to give high confidence ratings is also biased, and this bias is due to a number of different factors. For example, a prevalent finding is that participants who score higher on measures of anxiety and depression report overall lower confidence (Rouault et al., 2018). A recent paper (Chen & Rahnev, 2023) has used data from a number of experiments on the confidence database, and shows that the most frequent confidence rating a participant uses is also pressed the fastest (meaning that the bias to choose a particular confidence rating is also a preference to use a particular response key).

As a side note, the betting paradigm was one of the earlier suggestions for getting rid of the bias in confidence ratings. Persaud and colleagues (2007) used a paradigm where they controlled how many bets the participant can place overall, thereby setting the criterion for the participant. This was shown to cause other problems (Clifford et al., 2008), which is why we did not attempt to fix participants’ criteria here.

- 3) “However, I do not think it is appropriate to say that they were not confident when they did not choose to bet... How confident they were when they did not bet is unclear in this design.”

We do not attempt to know ‘how confident’ a participant is when they bet or when they do not bet. We think the participant is more confident when they bet and less confident when they do not bet. This is the same for ratings, we assume that a participant is more confident when they give a rating of 2 compared to a rating of 1, but we do not know that a rating of 1 means they were ‘not confident’. A rating of 1 will mean different things for different participants.

- 4) “The paper argues that betting is a manifestation of being confident because they are more accurate when they bet. If this argument holds, the paper should use the participant's responses (correct or incorrect) as the measure of confidence.”

We use two checks to confirm that bets were made based on confidence, as opposed to randomly. Confidence shows two typical features: participants will be more likely to be correct when they are more confident, and participants will give faster responses when they are more confident. We show these two features in **Figure 1D and 1E**. We do not argue that betting is a manifestation of being confident because participants are more accurate when they bet. We use two stereotypical features of confidence to confirm participants used their confidence to make their bets. To make this clearer in the manuscript we have changed the paragraph in the manuscript, which used to read:

“Although the overall tendency to bet may partly depend on inter-individual differences in risk aversion, there were also signs that bet responses did reflect a metacognitive evaluation of decision confidence: participants were more likely to be correct when they bet (difference in d' = 0.98 ± 0.28 ; $t(22) = 7.35$, $p < 0.001$; Figure 1D) and they also showed faster reaction times (mean difference in median reaction time = $0.07 \text{ s} \pm 0.02$, $t(22) = 6.63$, $p < 0.001$; Figure 1E).”

And now reads (P4L94):

“We confirmed that participants use their confidence to inform their bets with two stereotypical signatures of confidence. First, participants were more likely to be correct when they bet (difference in d' = 0.98 ± 0.28 ; $t(22) = 7.35$, $p < 0.001$; Figure 1D). Second, participants showed faster reaction times when they bet (mean difference in median reaction time = $0.07 \text{ s} \pm 0.02$, $t(22) = 6.63$, $p < 0.001$; Figure 1E). Although the overall tendency to bet may be biased by other factors such as risk aversion, trials on which participants bet still reflect higher confidence, and so bet responses can be used to decode neural representations of post-decision confidence in the decision-window.”

Rouault, M., Seow, T., Gillan, C. M., & Fleming, S. M. (2018). Psychiatric symptom dimensions are associated with dissociable shifts in metacognition but not task performance. *Biological psychiatry*, 84(6), 443-451.

Chen, S., & Rahnev, D. (2023). Confidence response times: Challenging postdecisional models of confidence. *Journal of Vision*, 23(7), 11-11.

Persaud, N., McLeod, P., & Cowey, A. (2007). Post-decision wagering objectively measures awareness. *Nature neuroscience*, 10(2), 257-261.

Clifford, C. W., Arabzadeh, E., & Harris, J. A. (2008). Getting technical about awareness. *Trends in cognitive sciences*, 12(2), 54-58.

Desender, K., Murphy, P., Boldt, A., Verguts, T., & Yeung, N. (2019). A postdecisional neural marker of confidence predicts information-seeking in decision-making. *Journal of Neuroscience*, 39(17), 3309-3319.

R2C3

Next, the paper addresses how humans learn from implicit and explicit value signals. However, it needs to be clarified what participants learn. I thought the paper targeted learning in the motion task, but this was different. Perseveration (which I take as the relation between the previous feedback and the next participant's response, though the definition was unclear) seems to be a measure of learning. In my mind, learning should be defined as

improved performance after training or the process in which this happens. However, no analysis exists to test whether the degree of perseveration changes over blocks. On the other hand, there was learning in the motion task, as the coherence changed over time. Then, the next concern arises: improved perception may change confidence or at least interact with confidence. However, there is no effort to dissociate learning of motion direction from confidence.

We apologise for the lack of clarity about the perseveration analysis, which was echoed by reviewer 3 (R3C5b) below. This comment has been very helpful in clarifying the purpose of this perseveration analysis and for this we are grateful to the reviewer. We did not mean to imply that perseveration is a measure of learning. Rather, the pattern of perseveration shows us that participants are using the feedback to learn (as opposed to ignoring it). We clarified this by changing the paragraph that previously read:

“We found response perseveration (the interaction between response and stimulus repetition), depended on the sign of the previous explicit feedback, a typical signature of feedback-learning (Figure 1F, left; 2 (value, 1 vs 2) x 2 (sign, + vs -) repeated measures ANOVA, main effect of sign $F(1,22) = 60.04$, $p < 0.001$ after Bonferroni correction for three comparisons; non-significant main effect of value $F(1,22) = 2.23$, $p = 0.15$ uncorrected; and the interaction, $F(1,22) = 5.18$, $p = 0.033$, which would not survive Bonferroni correction).”

To (P4L101):

“We confirmed that participants were relying on the explicit feedback using an analysis of response perseveration, a typical signature of learning from feedback. Participants should not repeat the same response to the same stimulus after receiving negative feedback (and should, after receiving positive feedback). This was indeed the case, a 2 x 2 repeated measures ANOVA showed a significant main effect of feedback sign (positive vs negative) on the tendency to repeat a response given a stimulus repetition (Figure 1F, left; $F(1,22) = 60.04$, $p < 0.001$ after Bonferroni correction for three comparisons). This effect was somewhat moderated by feedback value ($|1|$ vs $|2|$); but the interaction would not survive correction for multiple comparisons; $F(1,22) = 5.18$, $p = 0.033$). This suggests that feedback did reinforce behaviour, that is, participants relied on explicit feedback to improve their performance (learn).”

We do not expect perseveration to change over blocks. By decreasing coherence, we maintained performance (Figure 1B and 1C), so participants should maintain similar learning (using feedback in the same way, such that perseveration should be the same). Similarly, By decreasing coherence we maintained performance so the perception of motion direction in the presented stimuli did not improve: we do not attempt to show that improved perception changed confidence because there was no improved perception (Figure 1C). This technique of increasing task difficulty to maintain performance (as opposed to keeping task difficulty the same and measuring increased performance) is standard in the perceptual learning literature (e.g. Doshier & Lu, 1999). This is our measure that participants did learn.

Doshier, B. A., & Lu, Z. L. (1999). Mechanisms of perceptual learning. *Vision research*, 39(19), 3197-3221.

R2C4

The perseveration measure could have been clearer. I figure the perseveration is the probability that the participants repeat the same response, depending on the feedback. However, since the feedback and no-feedback trials were intermixed, it is unclear whether the previous trial had the same feedback condition as the current trial. Even if they could separate the two feedback conditions, there would be some influence from other feedback conditions. It was unclear why the authors intermixed these two feedback conditions or why they did not test the two feedback conditions independently.

We thank the reviewer for this comment, which we hope to have partially addressed at the previous comment (R2C3). Critically, by intermixing feedback conditions, the participant did not know if they would receive feedback when they made their response. We measure perseveration as the effect of the previous feedback on the current response, when the current response cannot be affected by the future feedback, which is unknown. So, whether the current and previous trials have the same feedback condition does not affect the perseveration measure.

We intermixed the feedback conditions because of the possibility that the participant only uses confidence as implicit feedback when they know they will not ever be given explicit feedback. By intermixing explicit feedback

and no-feedback trials, the participant could rely only on feedback trials to learn, and not attempt to learn on trials where they do not get feedback. We tried to make this explicit on P4L90:

“We designed this experiment such that participants were frequently given explicit feedback (50% of randomly intermixed trials), and so did not have to rely on decision confidence as the sole source of information to improve their performance. Evidence for learning on no-feedback trials could therefore be considered robust evidence for the involvement of confidence in learning.”

We have also separated out the part of the paragraph that details the effect of betting on perseveration on no-feedback trials, to reinforce this motivation (P5L110):

“Given the design of intermixed explicit-feedback and no-feedback trials, participants could have relied solely on explicit feedback to improve their performance. However, we found the same pattern of response perseveration on no-feedback trials, depending on whether the participant bet on their previous response...”

R2C5a

Technically, the analysis of combined EEG and fMRI is impressive. However, it needs to be clarified what this impressive technique shows conceptually. The paper shows the EEG decoder that decodes whether the participants bet a few seconds later. However, this decoder was made individually; thus, the EEG channels used, and the timing of EEG selected may differ across participants. Then, the underlying neural mechanisms contributing to the decoder could be different.

We thank the reviewer for this interesting perspective, which has helped us reformulate the paragraph on the use of EEG-informed fMRI (also in response to reviewers 1 and 4; R1C1 and R4C2). The EEG decoding analysis helps us to 1) identify time-resolved neural activity on the scalp that is related to behavioural variables; and 2) use this activity to estimate a trial-wise prediction of the internal variable underlying behaviour. This trial-wise prediction is formed using the weighted summation of activity across all EEG channels (Equation 1, P20L599). This multivariate approach is more powerful, especially in the context of simultaneous EEG-fMRI, where some channels may incur more artefacts than others, but distributed representations can still be resolved by tuning weights across channels to best capture behaviour within each participant.

We do also allow the timing of the decoder to differ across participants (but within the group-level significant window). The EEG representation of post-decision confidence is sustained for a relatively long duration (as can be seen from Figure 2); we choose the time-point that has the best signal-to-noise ratio for an individual to get the best prediction for the fMRI analysis. In addition, this captures individual differences in the timing of confidence formation, which have been shown consistently in the literature (Pleskac and Busemeyer, 2010; Desender et al., 2019).

One of the more interesting aspects of this comment is what it implies about the purpose of the EEG-informed fMRI analysis. The EEG predictions give us an aggregate of neural activity from the scalp that best aligns this activity along an axis of ‘will not bet’ to ‘will bet’ or an axis of ‘received negative feedback’ to ‘received positive feedback’. We take this axis is an estimate of the internal variable used to make the bet response, or the internal variable corresponding to the value of explicit feedback. It is this variable we are interested in, not the EEG channel activity per se. And this gives our fMRI analysis more specificity.

We hope to have made this clearer in the following paragraph (P9L240):

“These EEG-predictions give a fine-grained estimation of the subjective variables used to implement behavioural responses, as well as capturing trial-by-trial variability in the neural activity underlying these internal variables. In this way they afford greater explanatory power in capturing meaningful differences between trials with otherwise the same behavioural outcomes. In addition, these predictions disentangle effects in close temporal proximity (within the decision- or feedback-windows), due to the temporal resolution of electrophysiological signals, which would otherwise be difficult to dissociate in sluggish BOLD responses. Here, we focus on clusters of voxels in which the BOLD related to the EEG-predictions, leveraging the trial-by-trial variability in the internal representations captured by the LDA analysis of the electrophysiological signals. We expect these EEG-predictions to be related to (respectively): 1)

post-decision confidence; 2) expected outcome value; 3) explicit outcome value; and 4) implicit outcome value.”

R2C5b

In addition, the motor readiness potential may have been contaminated in the decoder since participants were not required to move their fingers to respond when they chose not to bet. The same criticism applies to other results in other time windows where the same bet-vs-no-bet decoder was applied. It is possible that the decoder only decodes whether finger movements occur or not, not the confidence.

This is an important critique for which we thank the reviewer. We do not think the EEG decoder only decodes whether finger movements occur or not, nor do we think the decoder is contaminated by motor readiness potentials. There are several reasons for this:

1) Decoding was performed (and associated decoding weights derived) around the time of the decision (Supplementary Figure 2). Since the bet cue was not until 1-2 seconds later (Figure 1A), it is unlikely participants already started preparing their bet response when they had just entered their decision, and in addition, the unpredictable temporal jitter prevents motor preparation signals associated with bet responses from being temporally aligned with the timing of the decoder. We've now highlighted this more clearly in the results text (P7L154):

“...which was separated from the bet-response by 1-2 s.”

And on Figure 2C, where we now explicitly state the 1-2 s gap between windows.

2) There is no motor contribution evident in the EEG topography (insert of Figure 2A, this would look like a bright red patch over left-lateralised motor sensors, centred around C3/C5, but spreading out from there, and potentially a blue patch over right-lateralised motor sensors, example below). The topography we find is similar to previous topographies for decoding confidence (ratings; Gherman & Philiastides, 2018; Pereira et al., 2020; Desender et al., 2019).

Left: Topography of EEG bet-prediction (insert of Figure 2A). Right: Topography of response readiness from Galdo-Alvarez et al., 2016).

3) Participants did not make any movements during the feedback window, and yet we see that a reliable confidence signal re-emerges to discriminate whether participants had bet 2-3s earlier (2-3s is far too long for ongoing movement related activity in EEG). If the decoder were only picking up on movement related activity, it would not show relevant differences in the feedback-window.

Galdo-Alvarez, S., Bonilla, F. M., González-Villar, A. J., & Carrillo-De-la-Pena, M. T. (2016). Functional equivalence of imagined vs. real performance of an inhibitory task: An EEG/ERP study. *Frontiers in Human Neuroscience*, 10, 467.

R2C6

Following is an example of a gap in logic (Lines 155-156), making the sentence difficult to follow: "In the absence of explicit feedback, participants did use their confidence to infer the points they may have gained/lost." Even though the bet-vs-no-bet EEG decoder successfully decoded the EEG in the feedback-window in the no-feedback trials, no other data show (1) that they used their confidence in the feedback-window and (2) were inferring the possible points.

We thank the reviewer for again providing critical notes on clarity. We instructed participants to use their confidence to think about how many points they may have won/lost when the no-feedback ‘??’ was shown on the screen. This was not clear in the first paragraph of the results:

“On explicit-feedback trials, participants were shown the points gained/lost on that trial, but on no-feedback trials participants could only infer how many points were awarded, and they were cued to do so.”

We have changed this to (P4L78):

“On explicit-feedback trials, participants were shown the points gained/lost on that trial; on no-feedback trials participants were instructed to infer how many points were awarded, cued with two question marks.”

This was more clear in the legend of Figure 1, which we have chosen not to edit (P6L134):

“In the feedback-window, participants were cued (coloured fixation) about whether they would receive explicit feedback, or be shown two question marks indicating they should think about how many points they think they won/lost on that trial.”

And similarly in the Methods (P18L509):

“A magenta cue meant no explicit feedback would be provided, instead participants were shown a question mark cueing them to assess how many points they think they earned (where points were awarded in the same manner as explicit feedback trials).”

We have altered the sentence the reviewer quotes to be explicit about this instruction, and why the sentence is not making a great logical leap but actually just suggesting participants followed instructions (P7L176):

“...suggesting that in the absence of explicit feedback, participants did follow the instruction to use their confidence to infer the points they may have gained/lost.”

R2C7

The EEG results take a lot of work to follow. Figure 2 has five subpanels, which are intricate and not independent. The borders between subpanels could be more explicit. The captions may not be accurate, as Fig.2E caption explains what the blue and yellow color-coded areas mean in the feedback-window. Still, this panel contains other colors used for the decision-window.

We hope to have improved the Figure with thanks to this comment. Specifically, we have relabelled panels A and B so that they correspond better to the borders between the subpanels.

We have clarified the caption of Fig.2E, which used to read:

“Clusters of voxels with a significant positive relation to the EEG feedback-prediction (3, blue) and bet-prediction (4, yellow) in the feedback-window, with the clusters from the decision-window for comparison.”

And now reads (P11L272):

“Clusters of voxels with a significant positive relation to the EEG feedback-prediction (3, blue) and bet-prediction (4, yellow) in the feedback-window, with the clusters from the decision-window for comparison (1, purple and 2, green).”

R2C8

The following fMRI analyses are also difficult to follow in Figure 3. Fig. 3B caption shows correlations between GPe & dStr and GPe and vStr. However, what is shown in the horizontal axis in Fig. 3B needs to be clarified. According to the paper, the feedback-window showed the points participants would get. I understand that the possible points are shown in the horizontal axis. But then, what are the no-bet and bet categories in the horizontal axis? Betting was done in the bet-window, different from the feedback-window. Fig. 3D is supposed to show the resemblance between the brain activations in the decision- and feedback-windows, where the correlation coefficients demonstrate the resemblance. However, the correlation coefficient is extremely small, nearly zero, in each ROI. Thus, these data do not support the claim that "earlier BOLD related to confidence in the IFG continues to drive GPe BOLD responses following feedback."

We thank the reviewer for this comment, which highlights critical details we failed to clarify. The bet and no-bet classification is done for the whole trial: just as we make the separation in the decision-window based on bets in the bet-window (for example, Figure 2A, Figure 1D, Figure 1E), we also make the separation in the feedback window based on bets in the bet window (for example, Figure 2B, Figure 1F). We hope to have made this clearer in the axis label, and in the caption, which now states (P14L359):

“Correlation between GPe and dStr (left) and vStr (right) by feedback condition (or bet/no-bet trials on no-feedback trials)...”

We agree that the actual correlation shown in Figure 3D is small, but a substantial amount of time has elapsed between windows. What is of interest is that the IFG is, if anything, more correlated with future GPe than itself. We hope to have made this clearer on the Figure itself. We have also re-arranged the paragraph to make it clearer that the assertion about IFG driving GPe is about the Granger Causal analysis. We have weakened the sentence prior to the one the reviewer quotes to now read:

“GPe BOLD in the feedback window was significantly correlated with the earlier BOLD in the IFG, and this correlation was on average greater than (though not significantly different from) the earlier BOLD in the GPe itself, suggesting there could be a Granger causal connection.”

And then moved the quoted sentence to after the granger causal analysis (P13L349):

“Leave-one-out Granger causal tests showed significant evidence against excluding the lagged IFG BOLD in predicting GPe BOLD in 18/23 participants (median $X^2=38.3$, median $p = 2.62e^{-6}$, median lag = 12 s, including non-significant participants). That is, earlier BOLD related to confidence in the IFG continues to drive GPe BOLD responses following feedback.”

Reviewer #3 (Remarks to the Author):

In this manuscript, Balsdon and colleagues show how implicit and explicit confidence signals are represented in the human brain, either following the decision or later, during a time-window in which feedback on performance was provided in half of the trials. During the task, participants responded to a perceptual task (decision window), betted on their performance, then got feedback in half of the trials (feedback window).

First, the authors use a computational model of reinforcement learning to show that the behavior was better reproduced under the assumption that confidence was used as implicit feedback during learning (compared to a model based only on explicit feedback). Second, the authors trained an EEG decoder to predict the outcome of the bet during the decision window. They found that during the feedback window, the output of the same decoder differed according to the feedback outcome when feedback was given and to the bet outcome in the absence of feedback. The later was considered to be a proxy to implicit outcome value. Conversely, they trained a decoder on the feedback outcome during the feedback window, which they consider to be a proxy to explicit feedback. They found that during the decision window, the output of this decoder differed according to the bet outcome. Third, the authors describe the hemodynamic correlates of the output of these decoders and find that they differ for implicit and explicit feedback. Finally, based on an overlap between correlates of implicit and explicit feedback in the GPe, the authors performed connectivity analyses pointing to an additional involvement of the thalamus, the insula and the rostromedial prefrontal cortex. Together their results provide important empirical evidence on the way that confidence is used as an implicit feedback for learning.

This is a very strong and elegant paper. The study will give a new breath to the field of metaperception which has mainly focused on the monitoring aspect and less on control, or the way that meta-representations can be helpful for adaptive behavior. In terms of methods, combining computational modeling, EEG and fMRI is probably the best that can be achieved non-invasively in human cognitive science and the added benefit of EEG is very clear (+special congrats to the authors that had to prepare the EEG cap before the scanner...). Although the analyses are quite complex, it was a pleasure to read because it is clearly written and very nicely illustrated. To highlight just one example, I liked the way that Fig. 2 shows the EEG data and explains which signals relate to which BOLD activations with a number for each regressor that can be traced back to the main text. I have some points, mainly clarifications that I'd like the authors to consider/answer but I think this manuscript definitely deserves to be

published here. Plus the data and code are open!

Signed: Michael Pereira

We thank Dr Pereira for his insightful review, the suggestions, especially concerning the modelling, have been incredibly helpful in improving the manuscript. To ensure we have fully addressed all comments, we have numbered each comment according to reviewer number and comment number, such that R2C1 corresponds to the first comment of Reviewer 2. We provide a track-changed document with all changes commented with this code. Page and Line numbers of these changes refer to the clean revised manuscript (not track-changed), and use a similar code: P15L376 refers to page 15, line 376.

R3C1a

#1. About "integration".

I would like the authors to explicit what they mean by "integration" which seems to be a central claim to the paper (e.g. lines 10, 65, 344, 377). Do you mean that a learning signal is constructed by reading out either explicit or implicit feedback? Then the term integration would not seem appropriate.

This is a central claim of the paper and we thank Dr Pereira for highlighting the lack of clarity here. To ensure we properly address all aspects of this comment, we have separated it into three sub-comments and address each in turn.

We have added more specifics of the integrated signal in the introduction (P3L40):

“In this way, confidence could be valuable in providing more nuanced information of how to adjust decision-making processes, even in the presence of explicit feedback. For example, confidence could be linearly combined (integrated) with the predicted outcome (expected value) to form the reward prediction error from explicit feedback, as an aggregate reinforcement signal²¹. This might necessitate not only separate neural signatures for the different (confidence vs explicit value) signals but also the presence of a downstream process in which the two signals are integrated to form an aggregate reinforcement-like representation to jointly influence learning.”

R3C1b

I presume that the authors mean that additionally, in explicit feedback trials, both types of feedback are integrated. For this stronger version, it seems that the authors cannot solely rely on the fact that BOLD activity in the GPe was related to both implicit and explicit outcome value signals since these signals are measured in different conditions. I understand that the Granger connectivity analysis supports this strong version. However, although the result section does make a decent job in constructing this argument, I think it would be nice to explicitly tie these different findings together in the discussion. The sentence "Our analysis further suggests these implicit and explicit representations of outcome value are integrated in the external globus pallidus, which corresponded both to post-decision confidence in the decision-window and explicit outcome value in the feedback-window." wasn't completely clear to me.

This concern is related to the comments of reviewer 4 (R4C3 and R4C5) below and the comments of reviewer 2 (R2C8) above, and we thank Dr Pereira for explaining it so clearly. We have weakened the integration claim in the discussion, while separating out the analyses that support the claim more clearly (P16L431):

“Our analysis further suggests these implicit and explicit representations of outcome value could be integrated to form an aggregate representation in the external globus pallidus. BOLD in the GPe was related to EEG-predicted confidence in the decision-window and explicit outcome value in the feedback-window, while a Granger causal analysis suggested feedback-window BOLD was also influenced by earlier BOLD in the IFG related to confidence.”

R3C1c

Maybe the authors could make a link between the decoded EEG signals used as parametric regressors, interpreted as explicit or implicit outcome (line 205) and $E(V_t)$ in the computational model, in order to provide a computational interpretation of this integration? Is there any possibility of demonstrating integration at the behavioral level by comparing a model that uses both confidence and feedback in feedback trials, and a model that still uses confidence in no-feedback trials but updates only based on explicit feedback in feedback trials?

We thank Dr Pereira for this suggestion. The two models we originally compared were using confidence on all trials, or not at all. We added this third suggested model (using confidence only on no-feedback trials) and rephrased the paragraph in the results to more clearly articulate how this is evidence in favour of an integrated learning signal (P5L115; we have also added some equations in response to the suggestion from Reviewer 1 R1C3a):

“We used a simple computational reinforcement learning model comparison to assess the use of confidence for learning. The models altered the internal response to the stimulus according to a learning rate (α) applied to the reward prediction error, that is, the difference between the explicit feedback value (r_t) and the expected value ($E[V_t]$). In this way, the mean response to a leftward stimulus, (μ_L) would be modified as: $\mu_{L_{t+1}} = \mu_{L_t} + \alpha(r_t - E[V_t])$. Three models were implemented to compare the use of confidence: 1) a model that does not use confidence (the expected value is updated according to a Rescorla-Wagner rule: $E[V_{t+1}] = \alpha_w(r_t - E[V_t])$); 2) a model that uses confidence only when explicit feedback is not provided (confidence, c_t , substitutes the reward prediction error, and is calculated as the cumulative probability of the perceptual evidence in terms of distance from the response criterion); 3) a model that also uses confidence on explicit feedback trials (the expected value is proportional to confidence, $E[V_t] = c_t + bet$). The model that used confidence on all trials (integrating confidence for the reward prediction error) provided the best description of behaviour (Figure 1C,F; $\sum BIC_3 - BIC_1 = -330.45$; $\sum BIC_3 - BIC_2 = -270.93$; exceedance probability > 0.99 ; see **Methods, Supplementary Figure 1**). This suggests confidence is integrated with explicit feedback to inform the reward prediction error and update processing for learning.”

We additionally mention this to help motivate the exploratory analysis of GPe (P12L308):

“Our computational modelling analysis suggested learning is supported by both confidence and explicit feedback when explicit feedback is given, implying some integration of these signals at the neural level.”

We also added note to this result in the discussion (P15L394):

“At the behavioural level we observed no-feedback trials to be modulated by confidence in a similar manner to explicit feedback trials, and a computational modelling analysis suggested learning integrated confidence even on explicit feedback trials. This was supported by our analysis of the neural data.”

We also tried using the EEG-predicted explicit and implicit value signals as the reward prediction error in the model. This EEG-informed model ignores the actual feedback presented, as well as ignoring the bet responses in calculating confidence. We were surprised how well this worked. The model incorporating the EEG-predictions described behaviour about equally well as the behaviour-only model (which is informed by the actual feedback and the bet responses). The behaviour-only model was slightly better in terms of sum of the difference in BIC $\sum BIC_{beh} - BIC_{EEG} = -33.85$, but the exceedance probability of the EEG-informed model was slightly better (which takes account of variability across subjects) $xp = 0.76$, though the protected exceedance probability suggested there was limited difference ($pxp = 0.56$). Although the trial-wise learning updates were only weakly correlated within individuals (median $r = 0.278$), the sign of the updates was the same for the majority of trials (median = 0.66, range [0.32 – 0.998]), while the actual learning rate was small, meaning this can suffice to produce similar predicted behaviour (correlation in predicted performance across subjects $r = 0.82$, $p < 0.001$). Because of this, we don't think this actually provides much stronger evidence for the use of confidence beyond the additional model comparison above. But, this does help with a comment from Reviewer 4 below (R4C1). Specifically, the EEG-predictions from the feedback window provide a much better description of behaviour in terms of this model than the predictions from the decision-window ($\sum BIC_{EEG_{fb}} - BIC_{EEG_d} = -111.90$; protected exceedance probability = 0.981), showing that these variables do relate better to behaviour in the predicted way.

This Figure shows (left) participant proportion correct by the predicted proportion correct of the EEG-informed model (using the EEG-predictions from the feedback window); (middle) the same for the behaviour-only model; (right) the predicted performance of the behaviour-only model compared to the EEG-informed model. For brevity, and because (as Reviewer 4 states) the manuscript is already so dense, we choose to only report brief details of this analysis (P8L213):

“We then examined the relevance of the bet- and feedback- prediction for learning. While both the feedback-prediction and the bet-prediction showed behaviourally relevant signals in the decision- and feedback-windows, we examined whether the signals in the feedback-window were more relevant for learning. We used the feedback-window EEG-predictions (feedback-prediction on feedback trials and bet-prediction on no-feedback trials) as the reward prediction errors in the computational model, and found this resulted in no substantial difference in the fit to behaviour (compared to the behaviour-only model relying on the explicit feedback and bet responses; $\sum BIC_{beha} - BIC_{EEG} = -33.85$, protected exceedance probability in favour of the EEG-informed model = 0.56). But, the EEG-predictions from the feedback-window provided a better description of behaviour than those from the bet-window ($\sum BIC_{EEG_{fb}} - BIC_{EEG_d} = -111.90$; protected exceedance probability = 0.981). This indicates that the feedback-window predictions are more related to outcome value and its use for learning, than the earlier signals from these same spatial filters in the decision-window.”

R3C2

#2. Errors

Participants have higher d' when they bet, therefore, they err more when they don't.

Is there a way for the authors to confirm that information about the outcome of the bet is present in the EEG irrespective of error detection? For example, does the trained decoder predict the outcome of the bet above chance for correct responses only? This is not a serious issue since we don't really know to what extent error monitoring and metacognitive monitoring differ in terms of mechanisms but I think it is worth considering. For example, with confidence, it's nice to show that confidence levels can still be decoded when taking only correct trials.

We appreciate this suggestion, and have added this analysis on P7L168:

“The decoder is not driven by an error detection signal, as the decoder predicted bet decisions even on correct trials only (mean GLM β -weight = 1.53 ± 0.42 , $t(22) = 17.34$, $p < 0.001$).”

R3C3

#3. Positive BOLD correlations

I was surprised to see that Fig. 2D shows a "positive relation to the EEG bet-prediction" (line 225). I am assuming that positive bet prediction should be interpreted as positive confidence. But previous works (including ours) found that BOLD in pMFC, IFG, SPL and co was negatively correlated to both "behavioral" confidence or an EEG proxy to confidence (Rouault et al., 2018; Morales et al., 2020; Pereira et al., 2020, including the fMRI analysis in Gherman & Philiastides, 2018). Is it a typo or did I miss something? If the relationship is really that way (i.e. high bet prediction \rightarrow high activation of these regions), is there an explanation beyond the trivial ones (different tasks

etc...)? Can we really say that it's "in line with previous literature" (line 353)? Wouldn't we expect a positive relationship more in the ventral striatum and vmPFC?

Relatedly –and I admit this is not a very original question– were there no other BOLD activations in the "other directions"?

This is an interesting point, which deserves a more in-depth analysis. There is heterogeneity in the literature concerning the direction of the relationship in various brain regions, which is perhaps why the Vaccaro and Fleming (2018) review omits the direction factor, we have now included this reference. One key component is the timing of the regressor. Rouault et al., 2018, Rouault et al., 2023, Morales et al., 2018, and the negative associations of Gherman and Philiastides, 2018 all use the later time-window of the confidence report. If we look at the BOLD timeseries from our confidence ROI, the relationship becomes negative later. This is also true for the BOLD timeseries from Geurts et al., 2022, which show a positive relationship early, close to stimulus offset, and then a later negative relationship (though the positive relationship is only significant for the rIPFC). We have now included the bold time-series from the decision-window as Supplementary Figure 11.

Supplementary Figure S11. BOLD timeseries of ROIs locked to time from the decision for trials on which the participant bet (light blue) and did not bet (dark blue) separately. These ROIs show a positive relation to post-decision confidence, based on the EEG bet-prediction, used as a regressor in the time-window immediately following the perceptual decision. In the previous literature, these ROIs often show a negative relation to confidence, but the regressor is placed in the later time-window corresponding to the confidence report. The BOLD timeseries shows this is consistent with our results, where the relation to confidence later flips. Abbreviations: dStr, dorsal striatum; vStr, ventral striatum; GPe, external globus pallidus; rIPFC, rostralateral prefrontal cortex; IFG, inferior frontal gyrus; pMFC, posterior medial frontal cortex.

From Geurts et al., 2022, Figure 4E (top) and 4B (below). Top shows the effect of confidence (blue) and uncertainty (red) on dAI, dACC and rIPFC. Dark shading shows the stimulus, light shading shows the response window. The timeseries of the dAI below shows that this effect emerges very late with respect to the perceptual decision.

We present the full uncorrected BOLD clusters in Supplementary Figures S3-S10. Similar to our previous work (Gherman and Philiastides, 2018), it was rare to find BOLD negatively correlated with the positive EEG signals locked to their timing. We therefore speculate that the difference in the direction of the effects in mPFC between our results and those of Pereira et al., 2020 is due to the fast onset of error detection related EEG activity (error positivity, Pe), which emerges from the same channels:

B. Error detection

C. Decision confidence

Figure 1B and 1C from the review of Desender, Ridderinkhof, and Murphy, 2021, showing post-decision confidence signals quickly flipping to error positivity in the context of speeded decisions (1C) while undetected

errors show little difference from correct responses (1B; different studies). In addition, error detection signals show a much larger effect than the earlier effect of confidence (1C).

Pereira et al., 2020 used speeded decisions in which the participants frequently detected errors and rated their confidence from 'sure error' to 'sure correct'. In the current experiment we used a more liberal response deadline (up to 1s), where most errors would be associated with uncertainty, as opposed to 'sure error'. The EEG bet-prediction does flip around 400 ms following the response (Figure 2A), but the size of this flipped effect is much smaller than the positive effect around the time of decoding. In other words, the negative association with mPFC BOLD early in time in Pereira et al., 2020 may be due to the fast prominent error positivity emerging from the same underlying sources, which is less well pronounced in the current experiment.

So as not to add too much additional analysis and discussion to the main text, we choose to leave this to the Supplementary materials, where interested readers can examine this point more closely. We refer to Supplementary Figure 11 on P11L281, when first presenting the fMRI results.

R3C4

#4. Methodology

While I'm at boring questions, are the bet and feedback predictions sufficiently decorrelated to be included as separate parametric regressors during the decision window? And is there a hierarchy in the way the regressors are treated (e.g. is the feedback predictor applied on the residuals after variance related to the bet predictor is regressed out)?

Also, the EEG decoder for explicit outcome is trained on the same window while the EEG decoder for implicit feedback is trained on a different window. Do you think this an issue for the interpretation of the BOLD correlate?

We apologise this detail was omitted and thank Dr Pereira for pointing it out. We calculated the variance inflation factor of our fMRI GLM for each participant. We have now included details of this analysis in the Methods (P21L630):

“The subject-wise maximum variance inflation factor ranged between 1.67 and 4.25.”

We also added this to the results section (P9L235), along with reporting the maximum correlation between EEG-predictions, which is actually fairly small:

“An analysis of the variance inflation factor indicated correlations in these variables were not substantial enough to be problematic for the fMRI analysis (the maximum ranged between 1.67 and 4.25 across subjects). The most correlated EEG-predictors were the bet-prediction in the decision-window and the feedback-prediction in the decision-window, the median correlation across subjects was $r = 0.115$ (ranging from -0.039 to 0.53).”

Minor:

R3C5a

line 50: is ref 28 really about the striatum and confidence.

We added this reference because of the discussion of the caudate-FEF relationship, but a previous manuscript from the same authors (Ding and Gold, 2010) may be more appropriate. The manuscript does not refer to confidence per se, but shows some activity in caudate modulated by evidence strength irrespective of choice, which the authors suggest is important for 'evaluating the decision'. We think this is relevant for confidence signals in striatum, but we can remove the reference if the reviewer prefers.

R3C5b

lines 98 - 102: I'm afraid I did not get that sentence

We apologise, this was echoed by reviewer 2 (R2C3 and R2C4). We have re-written the paragraph and hope it is now clearer (P4L101):

“We confirmed that participants were relying on the explicit feedback using an analysis of response perseveration, a typical signature of incorporating feedback for learning. Participants

should not repeat the same response to the same stimulus type after receiving negative feedback (and should, after receiving positive feedback). This was indeed the case, a 2 x 2 repeated measures ANOVA showed a significant main effect of feedback sign (positive vs negative) on the tendency to repeat a response given a stimulus repetition (**Figure 1F, left**; $F(1,22) = 60.04$, $p < 0.001$ after Bonferroni correction for three comparisons). This effect was somewhat moderated by feedback value ($|1|$ vs $|2|$); but the interaction would not survive correction for multiple comparisons; $F(1,22) = 5.18$, $p = 0.033$, uncorrected). This suggests that feedback did reinforce behaviour, that is, participants relied on explicit feedback to improve their performance (learn).”

R3C5c

Lines 202 - 205: You just focus on BOLD activity related to parametric regressors of EEG, right? At first I understood "BOLD related uniquely to the contribution of the EEG-prediction" as meaning that a region that would be activated both by parametric regressors and another task regressor would be discarded. If that were the case I would fail to get the rationale.

We thank the reviewer for pointing this out. We have modified the sentence to remove this ambiguity. The sentence now reads (P9L246):

“Here, we focus on clusters of voxels in which the BOLD related to the EEG-predictions, leveraging the trial-by-trial variability in the internal representations captured by the LDA analysis of the electrophysiological signals.”

R3C5d

Fig. 2: Was there any baselining of the EEG signal? How do the authors explain the pre-feedback activity?

This is an interesting question. We did not baseline, because of the long duration between time-windows we assumed there would be little need, and baselining by trial risks adding noise from variability in the baseline period. We checked that subtracting a baseline averaged across trials (for each sensor) did not affect decoder performance. We think the pre-feedback activity could be a genuine anticipation effect, where there is some suppression (proportionate to expected value) of the signals contributing to the sensor activity that gives rise to the later feedback value representation. The sensor activity is a mix of underlying source activity, the EEG prediction aligns with the GPe BOLD prior to feedback (Figure 3C), though in the ventral striatum the pre-feedback BOLD looks more like an expected value signal, with greater BOLD for bet trials.

We have made a note of the lack of baseline in the methods (P18L526):

“No baseline was applied (due to the long duration between time-windows).”

We added a small note to the Figure 2 legend speculating about the anticipation effect (P11L263):

“We speculate the pre-feedback effect could be due to suppressive anticipation.”

R3C5e

lines 439: 1-up 2-down, rather?

We were using up and down to refer to the coherence (to decrease difficulty one must increase coherence). But it is true the name of the staircase procedure is better referred to as 1-up 2-down, we have made the change (P17L495).

R3C5f

Methods: Would be nice to have a bit more info about the fitting procedure: how was the log likelihood computed, what optimization algorithm? (Unless I missed it).

We apologise that this detail was omitted. This has now been added (P19L558):

“The log likelihoods were calculated from the cumulative gaussian probability density corresponding to the choice and confidence (demarcated by the criteria), and the model was fit to minimise the negative log likelihood of participants’ perceptual decisions and bet responses

using a constrained nonlinear interior point optimisation algorithm implemented with MATLABs fmincon function.”

Reviewer #4 (Remarks to the Author):

In this manuscript, Baldson and colleagues investigate learning from implicit and explicit value signals in perceptual decision-making, in a small cohort of healthy human subjects (N = 23). The perceptual is a random-dot motion task (binary choice), followed by a metacognitive evaluation (bet or not bet on the answer) and by a feedback or absence thereof. The authors use simultaneous EEG-fMRI: EEG signal is used to decode a variety of variables: expected outcome, decision to bet and (expected) feedback. EEG predictions of those variables is then regressed on whole-brain BOLD signal. The authors then attempt to build a dynamic model of decision-making in the basal ganglia by leveraging a variety of functional connectivity analyses.

Overall, the manuscript is very dense: it constitutes a very brave endeavor, which led to the collection of a very rich and valuable simultaneous EEG-fMRI dataset. I also commend the authors for implementing an impressive collection of sophisticated methods, and for attempting to extract as much information about the decision-mechanism from their data as they possibly could

However these positive aspects often come at the cost of the clarity of the conclusion and the way they are exposed. I also have two major concerns about the analyses and their interpretations that would need to be addressed before I can recommend this manuscript for publications.

We thank the reviewer for their valuable suggestions, which we hope to have fully integrated into the revised manuscript. To ensure we have fully addressed all comments, we have numbered each comment according to reviewer number and comment number, such that R2C1 corresponds to the first comment of Reviewer 2. We provide a track-changed document with all changes commented with this code. Page and Line numbers of these changes refer to the clean revised manuscript (not track-changed), and use a similar code: P15L376 refers to page 15, line 376.

R4C1

One of my main concern is with the EEG decoding of variables. Basically, the authors show that they can decode a set of variables from the EEG signal (e.g. bet vs no-bet at decision; lines 144-145), then show that this variable can be decoded at another time point (E.g. feedback; lines 151-156) and then ascribe a new computational meaning to it (participants infer the outcome; line 156).

The problem is, as the authors note elsewhere, all variables (decision to bet, expected reward, actual feedback, inferred feedback) are all highly correlated, and are also correlated with other latent variables (notably, the inferred probability of being correct should underpin all those estimations) as well as behavioral output (such as the decision reaction time, as noted line 146, which should basically correlate with everything).

Put simply, my take is that if the authors want to ascribe some specific EEG signals to some specific (latent) variables, they should estimate those variables from the behavior (choice, RT), stimulus (strength, direction, etc.) and model (supp mat), and properly test if and how EEG signal better correlate with some variables than others, at specific time-points.

Without those analyses, I am not convinced that the EEG decoding exercise can really inform us about the actual computations that are performed, given the potential correlation between all variables.

We really appreciate this comment. First, and in response to a comment from reviewer 2 (R2C6) we have clarified that participants were instructed to infer the outcome on no-feedback trials. The referenced statement previously on line 156 now reads (P7L176):

“...suggesting that in the absence of explicit feedback, participants did follow the instruction to use their confidence to infer the points they may have gained/lost.”

But the reviewer is correct that we do imply there is different computational meaning to the decoded variables at different times. We stated:

“Together these analyses give us four EEG-predictions to inform the GLM analysis of simultaneously acquired fMRI BOLD signal: the bet-prediction that best discriminates bet from

no-bet trials in the decision-window (representing post-decision confidence); the bet-prediction that re-emerges at the time of feedback, that discriminates bet from no-bet trials in the absence of explicit feedback (representing implicit outcome value); the feedback-prediction that best discriminates positive from negative feedback in the feedback-window (representing explicit outcome value); and the feedback-prediction in the decision-window, that shows the pattern of EEG activity relevant for discriminating the sign of explicit feedback is present even before explicit feedback is given (representing expected outcome).”

We meant these statements (eg “representing post-decision confidence”) more as a way to ascribe a conceptual tag to the EEG-predictions. We have changed the term “representing” to “related to”, now P9L225.

Rather than it being a problem that the EEG bet-prediction in the decision-window is correlated with choice-accuracy and response times, this is a feature that provides additional evidence that it is related to post-decision confidence. As the reviewer suggests, we show that the post-decision confidence EEG signal relates better to response times and choice accuracy than the re-emergence of this signal in the feedback-window (which we propose is related to implicit feedback). The correlation between the EEG bet-prediction in the decision-window and the re-emergence of these signals in the feedback window is actually rather low (median $r = 0.03$, range $[-0.07, 0.15]$). We highlight all of this more clearly in the following paragraph, which we have separated from the description of the bet decoding analysis (P7L162):

“As validation of the behavioural relevance of the EEG bet prediction, we show that the bet-prediction not only discriminated bet from no-bet trials (from the time of the response to 0.25 s after, mean $F(1,22) = 15.36$, cluster corrected $p < 0.001$; **Figure 2A, left**), but also showed a significant main effect of response accuracy (from 0.1 to 0.24 s after the response, mean $F(1,22) = 6.66$, cluster corrected $p = 0.002$). That the decoded bet-prediction in the decision-window is related to post-decision confidence is supported by the significant prediction of correct responses (mean GLM β -weight = 0.22 ± 0.16 , $t(22) = 6.62$, $p < 0.001$) and reaction times on correct trials (mean GLM β -weight = 0.15 ± 0.06 , $t(22) = 5.31$, $p < 0.001$). The decoder is not driven by an error detection signal, as the decoder predicted bet decisions even on correct trials only (mean GLM β -weight = 1.53 ± 0.42 , $t(22) = 17.34$, $p < 0.001$). In this way, the EEG bet-prediction could reflect a more graded representation of the underlying confidence used to arbitrate whether or not to bet on individual trials.”

We account for the correlation with response times in our fMRI analysis by modulating the stimulus onset regressor by response time, which we now indicate more clearly on P9L232:

“In addition, regressors on stimulus onset (modulated by response time), the bet cue, and the feedback cue were used to capture BOLD related to these externally driven events (Figure 2C, see Methods).”

Given how the GLM is implemented in FSL, any shared variance between RT and the EEG-informed regressor would be removed.

Finally, we present an additional analysis to relate the EEG-predictions in the feedback-window to the learning signals driving behaviour (and how they are more related than the EEG-predictions in the bet-window). Thanks to the suggestion of reviewer 3 (R3C1) we show that we are able to substitute the reward prediction error estimated in the computational model of behaviour (relying on the explicit feedback and bet responses) to that of the EEG-predicted explicit and implicit outcome value signals from the feedback-window without substantial decrement to the description of behaviour. In addition, we show that these feedback-window EEG-predictions relate better to the learning signals than the earlier signals in the decision-window (model comparison shows a substantial decrease in the model fit: $\sum BIC_{EEG_fb} - BIC_{EEG_d} = -111.90$; protected exceedance probability = 0.981). We outline this along with highlighting the relationship between the explicit EEG feedback-prediction and behavioural response perseveration in a separate paragraph (P8L213):

“We then examined the relevance of the bet- and feedback- prediction for learning. While both the feedback-prediction and the bet-prediction showed behaviourally relevant signals in the decision- and feedback-windows, we examined whether the signals in the feedback-window were more relevant for learning. We used the feedback-window EEG-predictions (feedback-prediction on feedback trials and bet-prediction on no-feedback trials) as the reward prediction

errors in the computational model, and found this resulted in no substantial difference in the fit to behaviour (compared to the behaviour-only model relying on the explicit feedback and bet responses; $\sum BIC_{\text{beha}} - BIC_{\text{EEG}} = -33.85$, protected exceedance probability in favour of the EEG-informed model = 0.56). But, the EEG-predictions from the feedback-window provided a better description of behaviour than those from the bet-window ($\sum BIC_{\text{EEG}_{fb}} - BIC_{\text{EEG}_d} = -111.90$; protected exceedance probability = 0.981). This indicates that the feedback-window predictions are more related to outcome value and its use for learning, than the earlier signals from these same spatial filters in the decision-window.”

We hope this provides more convincing evidence that the EEG-predictions are related to the computational variables as described, and related specifically as opposed to everything being related to each other (though we also tone down our assertions, in relation to R4C5 below). In addition, we clarify (also in relation to R3C4) that the correlation across EEG-predictions is not so strong (P9L237):

“The most correlated EEG-predictors were the bet-prediction in the decision-window and the feedback-prediction in the decision-window, the median correlation across subjects was $r = 0.115$ (ranging from -0.039 to 0.53).”

R4C2

In a similar vein, the authors show that the EEG-informed analysis of fMRI is better than a model-free analysis of fMRI (Supplementary Figures 3-10) but I doubt that the EEG-informed analysis of fMRI signal can actually perform better than a classical model-based fMRI analysis, especially if the goal is to ascribe computational variables to some brain signals/regions.

We thank the reviewer for highlighting this important point. We did not mean to imply such a value-laden statement. There are plenty of instances where stand-alone fMRI can perform sufficiently well without the addition of an EEG-informed analysis. The EEG-informed analysis is specifically beneficial in cases where one wishes to examine subjective variables, and variables that evolve dynamically over time. This is the case for confidence, which has been shown to emerge during decision-making and continue to evolve after. Subjective confidence is also thought to rely on a graded representation which is then used to make categorical behavioural responses. Previous work from our lab (and others using this technique) has examined more fully the relationship between EEG-informed and stand-alone fMRI analyses (Philiastides et al., *Ann Rev Neurosci*, 2021) and highlighted instances in which EEG-informed analyses can be more powerful in capturing BOLD associated with the endogenous neural variability, in particular, reflecting graded subjective representations.

Here the EEG gives a continuous estimate of the subjective confidence on which observers make their bet responses. We have modified the description in the results to more closely reflect this (P9L240):

“These EEG-predictions give a fine-grained estimation of the subjective variables used to implement behavioural responses, as well as capturing trial-by-trial variability in the neural activity underlying these internal variables. In this way they afford greater explanatory power in capturing meaningful differences between trials with otherwise the same behavioural outcomes. In addition, these predictions disentangle effects in close temporal proximity (within the decision- or feedback-windows), due to the temporal resolution of electrophysiological signals, which would otherwise be difficult to dissociate in sluggish BOLD responses. Here, we focus on clusters of voxels in which the BOLD related to the EEG-predictions, leveraging the trial-by-trial variability in the internal representations captured by the LDA analysis of the electrophysiological signals. We expect these EEG-predictions to be related to (respectively): 1) post-decision confidence; 2) expected outcome value; 3) explicit outcome value; and 4) implicit outcome value.”

And have also altered the paragraph discussing the difference between the EEG-informed and the stand-alone fMRI analysis to try to emphasise this is less about whether one is better, but rather how the EEG-informed analysis in this case can provide more specific results P11L292:

“Of note, this EEG-informed analysis produces some key differences from the binary behavioural variables that would have been used in a stand-alone fMRI analysis (see

Supplementary Figures 3-10). In particular, the binary bet vs no-bet comparison in the decision-window does not capture the relationship between post-decision confidence and the external globus pallidus. Similarly, the dorsal striatum cluster corresponding to the re-emergence of confidence as an implicit outcome value estimate in the feedback-window is absent for a comparison of binary bet vs no-bet trials in the feedback window. The stand-alone fMRI analysis does resolve a cluster in the caudate (which has previously been associated with learning under uncertainty^{49,50}), however, this does not capture the gradient with explicit value in the striatum, which deserves further investigation. Had the endogenous trial-wise variability captured in the EEG-predictions resulted from unrelated noise, these variables would have been less powerful predictors in the fMRI GLM. The EEG-informed analysis also provides more specific results, for instance, the stand-alone feedback regressor results in a large cluster over occipital cortex that likely captures the difference in luminance of the feedback visual cues. This suggests that the EEG-predictions give a closer approximation of the graded subjective variables underlying behaviour, where this can reveal a richer picture of the BOLD correlates, especially in the basal ganglia.”

R4C3

My second major concern the last section of the result, regarding the building of the decision/learning neuro-computational model within the basal ganglia, using functional connectivity analyses (lines 264-336). Once again, this section looks very impressive, and the authors really put tremendous effort to combine a lot of sophisticated analysis tools (Pearson correlation functional connectivity, vector autoregressive models of BOLD timeseries together with leave-one-out Granger causal tests, Psychophysiological interaction analysis).

But the strength of the interpretations and conclusions of these analyses remain unclear to me... because the hard truth is that all those analyses rely on the BOLD signal that is basically pictured in Figure 3C: noisy, lagged BOLD signal, from small samples (N = 23), in a task that is much more fast-paced than the haemodynamic signal.

In this respect, I find a bit worrying the BOLD time-courses, which all seem to peak -2 to +5 sec around feedback, when they should be elicited by the said feedback. Therefore, I sincerely think there is a significant risk in trying to extract too much out of this.

In my opinion, the paper would still be more than interesting enough with a more in-depth model-based analysis of the first section on the variables mapping – i.e. without this last “connectivity” section, which might have difficulty to stand the test of time and replications

We really appreciate this comment and entirely agree that the results need to be interpreted in the context of the small sample. Some of the earlier BOLD response is due to anticipation or confidence. For example, in the ventral striatum, it is the bet trials that show greater pre-feedback BOLD, but then the incorrect (feedback -2, red) trials BOLD declines, with a negative around 5 s post feedback, consistent with the haemodynamic response function. A similar picture is visible for GPe. We have now included the BOLD timeseries locked to the decision-window in Supplementary Figure 11. These timeseries are important for demonstrating, as the reviewer notes, that the signal is noisy and lagged, but also to appreciate the overall pattern of BOLD response in these different regions. We do aim to present this data as openly as possible. We would like to keep the connectivity section because it is interesting for future research. These findings also connect well with prior research. As we note in the discussion, the indirect pathway, and especially connectivity from GPe to thalamus and cortex has been shown to be important for modulating processing in a variety of contexts. As per the reviewer’s suggestion we now tone down some of the conclusions and emphasise instead the exploratory nature of this analysis. We hope made the appropriate modifications to make clear that we present this exploratory analysis as additional support for future examination of the role of GPe in learning and behaviour. We have stated this more clearly in the discussion (P16L442):

“These exploratory findings therefore build on mounting evidence supporting the cardinal role of GPe in modulating behaviour⁴⁹, where in addition, strong projections back from GPe to GABA interneurons in the striatum⁵⁹ put the GPe in the position to control information flow throughout the basal ganglia⁶⁴, which deserves further investigation in future work.”

In the results, we have highlighted the need for future investigation in relation to the integration of confidence and feedback in the GPe (P13L350):

“Taken together, these results suggest that GPe integrates post-decision cortical confidence and later subcortical outcome value signals, with no substantial difference in this integration depending on whether explicit feedback is provided. We encourage this integration of confidence and explicit feedback to be more closely assessed in future work.”

We made sure to highlight that this analysis was exploratory. At the beginning of the section we stated: “we investigated the post-hoc hypothesis...” we now emphasise this again in the third paragraph (P12L335):

“Next, we assessed the post-hoc hypothesis that the GPe BOLD in the feedback-window could be driven by earlier BOLD related to post-decision confidence...”

And the paragraph after (P14L372):

“Finally, we explored how an integrated feedback signal encoded in GPe could be used to implement learning, using a Psychophysiological interaction analysis...”

We hope by emphasising the exploratory nature of this analysis and encouraging future investigation we now make clear to the reader that they should be wary of these results.

Minor

R4C4

As I said in my general evaluation, I found this manuscript quite hard to read, because sometimes too dense, too technical and maybe too ambitious in the breadth of the analyses exposed. Some statistical details span over more than 4-line parentheses, which completely break the flow of the demonstration. I suggest that the authors invest some time in streamlining their demonstration, to ease the reading.

We thank the reviewer for expanding on this point. We have read through the results section to find places to ease reading. In particular, we now more clearly signal at the start of each paragraph the purpose of the analysis. For bet behaviour (P4L94):

“We confirmed that participants use their confidence to inform their bets with two stereotypical signatures of confidence. First, participants were more likely to be correct when they bet (difference in d' = 0.98 ± 0.28 ; $t(22) = 7.35$, $p < 0.001$; **Figure 1D**). Second, participants showed faster reaction times when they bet (mean difference in median reaction time = $0.07 \text{ s} \pm 0.02$, $t(22) = 6.63$, $p < 0.001$; **Figure 1E**). Although the overall tendency to bet may be biased by other factors such as risk aversion, bet responses still reflect higher confidence.”

With the help of the suggestions of reviewers 2 and 3 (R2C3, R3C5b), we have clarified the response perseveration analysis (where the statistics originally spanned a number of lines), and again more clearly state the purpose at the beginning of the paragraph. This now reads (P4L101):

“We confirmed that participants were relying on the explicit feedback using an analysis of response perseveration, a typical signature of incorporating feedback for learning. Participants should not repeat the same response to the same stimulus type after receiving negative feedback (and should, after receiving positive feedback). This was indeed the case, a 2×2 repeated measures ANOVA showed a significant main effect of feedback sign (positive vs negative) on the tendency to repeat a response given a stimulus repetition (**Figure 1F, left**; $F(1,22) = 60.04$, $p < 0.001$ after Bonferroni correction for three comparisons). This effect was somewhat moderated by feedback value ($|1|$ vs $|2|$; but the interaction would not survive correction for multiple comparisons; $F(1,22) = 5.18$, $p = 0.033$, uncorrected). This suggests that feedback did reinforce behaviour, that is, participants relied on explicit feedback to improve their performance (learn).”

We also separated this as an individual shorter paragraph, and expanded on the model comparison in a separate paragraph below.

We clarify the additional analyses of the EEG decoder predictions using a similar approach, stating the purpose of the analysis at the beginning of the paragraph. For the bet prediction (P7L162):

“As validation of the behavioural relevance of the EEG bet prediction, we show that the bet-prediction not only discriminated bet from no-bet trials (from the time of the response to 0.25 s after, mean $F(1,22) = 15.36$, cluster corrected $p < 0.001$; **Figure 2A, left**), but also showed a

significant main effect of response accuracy (from 0.1 to 0.24 s after the response, mean $F(1,22) = 6.66$, cluster corrected $p = 0.002$).”

We highlight the progression of the EEG analysis more clearly (P8L189):

“We next identified the EEG signatures of explicit feedback, using the same LDA analysis to discriminate positive (+1 or +2 points) from negative (-1 or -2 points) explicit feedback trials, training within the feedback-window.”

And the similarity of the analysis across the two decoders (P8L195):

“Similar to the bet-prediction, we applied this filter over time in the feedback window (**Figure 2B, bottom left**). Splitting trials by feedback, there was a main effect of feedback sign (positive vs negative feedback) from 0.35 to 0.53 s (2 (feedback sign) x 2 (feedback value) ANOVA, mean $F(1,22) = 9.62$, cluster corrected $p < 0.001$).”

We also separated out the ANOVA results for the EEG-feedback prediction so the statistics do not cross lines (P8L189):

“We next identified the EEG signatures of explicit feedback, using the same LDA analysis to discriminate positive (+1 or +2 points) from negative (-1 or -2 points) explicit feedback trials, training within the feedback-window. Spatial filters were first generated for each time-point within the feedback-window, and then a robust individual filter was selected for each participant as the average of five consecutive filters that best discriminated positive from negative feedback trials within the group-level significant time-window (see **Methods, Supplementary Figure 2**). Applying this filter over time in the feedback window, there was a main effect of feedback sign (positive vs negative feedback) from 0.35 to 0.53 s (2 (feedback sign) x 2 (feedback value) ANOVA, mean $F(1,22) = 9.62$, cluster corrected $p < 0.001$; **Figure 2B, bottom left**). Despite being trained only to discriminate feedback sign, the feedback-prediction showed an interaction between feedback sign and absolute (1 vs. 2) value (same ANOVA, from 0.43 to 0.52 s following feedback, mean $F(1,22) = 6.81$, cluster corrected $p = 0.006$).”

We once more highlight the relevance of the analysis at the beginning of the paragraph when discussing the new analyses for examining the relevance of the EEG signatures for learning (P8L213):

“We then examined the relevance of the bet- and feedback- prediction for learning.”

We better motivate the post-hoc analysis of the GPe (P12L308):

“Our computational modelling analysis suggested learning is supported by both confidence and explicit feedback when explicit feedback is given, implying some integration of these signals at the neural level. A cluster of voxels centred on the external globus pallidus showed significant relation to both the EEG representation of post-decision confidence and later, in the feedback-window, the EEG representation of explicit outcome value (**Figure 2E**).”

And signal the relevance of the comparison of GPe, dStr and vStr (P12L317):

“First, we examined the relationship between GPe BOLD and BOLD related to implicit and explicit feedback.”

R4C5

I was sometime a bit shocked by how assertive some interpretations and conclusions were, notably about variable-mappings and mechanisms that are, in fine, not unambiguously demonstrated (e.g., from the abstract: “These two signals are then integrated into an aggregate representation in the external globus pallidus, which broadcasts updates to improve cortical decision processing via the thalamus and insular cortex, irrespective of the source of feedback”). I suggest to tone down, and make a better effort at acknowledging the potential limitations of the conclusions

We are very grateful for this comment, we did not realise how assertive the claims appeared. We have been over the manuscript to ensure claims are not too assertive, signalling where we are interpreting the results, and speculating their relevance. Below, we list the changes, using Bold Font to highlight the exact differences.

We have softened the claims of the last sentence of the abstract (while keeping to the word limit; P2L9):

“These two signals **appear to** integrate into an aggregate representation in the external globus pallidus, which **could** broadcast updates to improve cortical decision processing via the thalamus and insular cortex, irrespective of the source of feedback.”

Previously: “These two signals **are then integrated** into an aggregate representation in the external globus pallidus, **which broadcasts** updates to improve cortical decision processing via the thalamus and insular cortex, irrespective of the source of feedback.”

And the last sentence of the introduction (P3L66):

“A psychophysiological interactions analysis **suggests** GPe broadcasts updates to improve cortical decision processing via the thalamus, irrespective of whether the source of feedback included explicit feedback or not.”

Previously: “Moreover, these two signals of implicit and explicit feedback appear to integrate into an aggregate representation in the external globus pallidus, **which broadcasts** updates to improve cortical decision processing via the thalamus, irrespective of whether the source of feedback included explicit feedback or not.”

In the first paragraph of the Discussion, we highlight that we are interpreting our analysis of the data (P15L392):

“a computational modelling analysis **suggested** learning integrated confidence even on explicit feedback trials. This was **supported by our analysis** of the neural data.”

Previously: “While at the behavioural level we observed no-feedback trials to be modulated by confidence in a similar manner to explicit feedback trials, at the neural level we found evidence that confidence was integrated to update decision processes even when explicit feedback was provided.”

And weaken the claims of the last sentence (P15L401):

“Stronger connectivity between the external globus pallidus and the thalamus, insular and frontal cortex **predicted** the interaction between response perseveration and implicit/explicit feedback sign, **supporting** the role of GPe in modulating learning via information flow in the basal ganglia.”

Previously: “Stronger connectivity between the external globus pallidus and the thalamus, insular and frontal cortex **predicted** the interaction between response perseveration and implicit/explicit feedback sign, **emphasising** the role of GPe in modulating learning via information flow in the basal ganglia.”

In the second paragraph (P15L402):

“Our results **point to** additional processes by which confidence **can be** used for learning.”

Previously: “Our results **highlight** the additional processes by which confidence **is** used for learning.”

We also weaken the claims about integration (P16L431):

“Our analysis further suggests these implicit and explicit representations of outcome value **could be** integrated to form an aggregate representation in the external globus pallidus.”

Previously: “Our analysis further suggests these implicit and explicit representations of outcome value **are** integrated to form an aggregate representation in the external globus pallidus”

And in response to comment 3 above (R4C3, P16L442):

“These **exploratory** findings therefore build on mounting evidence **supporting** the cardinal role of GPe in modulating behaviour, where in addition, strong projections back from GPe to GABA interneurons in the striatum put the GPe in the position to control information flow throughout the basal ganglia, **which deserves further investigation in future work.**”

Previously: “These findings therefore build on mounting **evidence for the** cardinal role of GPe in modulating behaviour, where in addition, strong projections back from GPe to GABA interneurons in the striatum put the GPe in the position to control information flow throughout the basal ganglia.”

And several changes in the final summary paragraph (P16L446):

“In summary, these results **are consistent with** the ubiquitous use of confidence as implicit feedback for improving future behaviour. **Our analysis suggests** the value of implicit feedback is encoded in a distinct manner to that of explicit feedback, where we show a dorsal-ventral spatial gradient within the striatum corresponding to these distinct motivational sources. This illustrates a distinction in striatal coding of value within a single task and learning context. The signals from dorsal and ventral striatum **appeared to be** combined in the external globus pallidus, facilitating updating of choice behaviour for learning via the thalamus in similar ways following both explicit-feedback and no-feedback trials. This highlights that even when external feedback is available, metacognitive estimates of confidence **could** provide us with **additional** nuanced information to update our internal processes for improving behaviour.”

Previously: “In summary, these results **highlight** the ubiquitous use of confidence as implicit feedback for improving future behaviour. **The value** of implicit feedback is encoded in a distinct manner to that of explicit feedback, where we show a dorsal-ventral spatial gradient within the striatum corresponding to these distinct motivational sources. This illustrates a distinction in striatal coding of value within a single task and learning context. The signals from dorsal and ventral striatum **were** combined in the external globus pallidus, **and facilitated** updating of choice behaviour for learning via the thalamus in similar ways following both explicit-feedback and no-feedback trials. This highlights that even when external feedback is available, metacognitive estimates of confidence provide us with nuanced information to update our internal processes for improving behaviour.

REVIEWER COMMENTS

Reviewer #1 (Remarks to the Author):

The authors have thoroughly addressed all the points I raised.

Reviewer #2 (Remarks to the Author):

I appreciate the authors' efforts to address my concerns, though one remains unresolved.

1. Figure 1B clearly shows that the coherence in the motion stimuli decreased to maintain performance with a standard staircase method. This aligns with what is commonly referred to as perceptual learning.

The study encompasses two distinct forms of learning: perceptual learning and learning to interpret feedback signals.

In the rebuttal, the authors stated, "By decreasing coherence we maintained performance so the perception of motion direction in the presented stimuli did not improve." This might suggest that the authors believe identical performance (response accuracy) implies the absence of perceptual learning. I am afraid that is not correct. Coherence in motion was reduced precisely because there was an improvement in the perception of motion direction.

I now understand that the intention behind increasing task difficulty was to maintain consistent feedback signals throughout the session. However, Figure 1B and other texts strongly suggest that perceptual learning did occur. This introduces the possibility of alternative interpretations. For example, it could be interpreted that the decoder classified something related to motion perception rather than betting.

Please clarify whether perceptual learning in motion occurred and, if so, explain why this does not pose a problem in the paper.

Minor concerns.

1. Please explicitly state in the text that finger movements or motor readiness potentials could be negligible, providing reasons for this assertion.
2. Figure 3. While the interpretation of the causal relation is acceptable, it is crucial to explicitly state that the effect sizes, represented by the correlation coefficients, are extremely small.

Reviewer #3 (Remarks to the Author):

The authors have provided a very thorough answer to all my comments. I find the "integration" aspect much clearer now and the authors have performed various additional analyses, sometimes beyond my expectations. I have no further comments.

Congratulations on an outstanding paper.

Reviewer #4 (Remarks to the Author):

In this revised manuscript, Baldson and colleagues have provided constructive answers to most points raised by all reviewers, leveraging a whole set of new analyses. Despite this net improvement, I remain very conflicted about this manuscript.

On the one hand, I can only acknowledge the quality, volume and sophistication of the provided analyses. On the other hand, I cannot get rid of the sentiment that the paper as a whole somewhat does not succeed to deliver the clear, unambiguous demonstration it ambitions to. After a genuine soul-searching exercise, I think I realized that I am not convinced by the premises of the manuscript about confidence-driven perceptual learning, and I think the chosen formalization (model) fails to clarify what it is and how it should/could work. This really limits my trust in the subsequent demonstration based on the complex analysis of neural signals (especially when analyses become too sophisticated and distal, and attempt to say too much from very noisy data, like the connectivity analysis).

Naturally, these sentiments should probably not be a reason to prevent this manuscript from being published, and scientific progress is made from debates between researchers who "agree to disagree".

Nonetheless, I'll detail my concerns and doubts below, such that the authors can evaluate if they can engage into any extra effort in the direction of clearing them – what would definitely help to convince the skeptics like me of the underlying value of the present work, starting from the very phenomenon that they are trying to capture.

Major concerns

I realized I am definitely not convinced by the proposed perceptual learning framework. Sure, performance improves over time (or rather, the staircase difficulty increases), but I do not understand (or agree about) what is learned, and whether it is indeed a consequence of feedback and/or confidence. These doubts are actually increased by the chosen computational framework: if I understand correctly, the reinforcement-learning inspired model proposed by the authors basically learn the value of correct response conditional on the evidence but this hardly makes sense: either the agent has access to the (noisy) perceptual evidence in which case the correct response is trivial (the response to be selected is congruent to the evidence) or the agent does not have access to the (noisy) perceptual evidence but then I do not understand what constitutes the stimulus/cue to be associated with a value: there is no way to categorize the cue/evidence ex-ante (μ_L or μ_R ?) before computing its “value”. In simpler terms, how can you update the value of leftward evidence (μ_L) if you haven't already categorized it as a left-stimulus – in which case the trivial correct answer is left? Also, what is really learned if perceptual sensitivity (d') does not improve (Figure 1C)? Can it really be “perceptual” learning if perception does not improve?

Regarding the model validation using the pattern of perseveration (Figure 1F), wouldn't the same pattern be observed “statistically” if the agent does not respond to feedback at all (no learning), but “simply” produces ~70% of correct responses (e.g. uniform between 60-80%), and only bet when they expect a certain accuracy (e.g. >70%) and we keep the task structure as is (pseudo-randomization of trials, etc.)? What is the unit of the y-axis? It also seems that the model fails to incorporate the magnitude dimension of the feedback, which does not surprise me, given its structure.

In addition to the conceptual problems outlined above, I also found the implementation and validation of the model difficult to read and lacking critical quality control analyses, such as parameter recovery and model identification analyses (Wilson & Collins, 2019). The inter-individual account given in Supp. Figures 1G-I is a very distal appreciation of the data, and capturing inter-individual variance does not necessarily mean that the important dimensions of within-individual variance are correctly captured by the model. As stated above, I'm not convinced that the figure provided in Figure 1F constitutes a decent support for the model in favor of alternative simpler explanations. The authors would need to find a unique signature of the model that can be found in the behavior and falsifies the alternative accounts (Palminteri et al., 2017)

Minor

If the authors really want to show that some signal (RT and/or EEG) is a signature of confidence, wouldn't it be convincing to show that it follows classical signatures of confidence signals, such as the X-pattern of confidence as a function of evidence for correct vs incorrect choices? See e.g. (Sanders et al., 2016).

Palminteri, S., Wyart, V., & Koechlin, E. (2017). The Importance of Falsification in Computational Cognitive Modeling. *Trends in Cognitive Sciences*, 21(6), 425–433.

<https://doi.org/10.1016/j.tics.2017.03.011>

Sanders, J. I., Hangya, B., & Kepecs, A. (2016). Signatures of a Statistical Computation in the Human Sense of Confidence. *Neuron*, 90(3), 499–506. <https://doi.org/10.1016/j.neuron.2016.03.025>

Wilson, R. C., & Collins, A. G. (2019). Ten simple rules for the computational modeling of behavioral data. *eLife*, 8, e49547. <https://doi.org/10.7554/eLife.49547>

Reviewer #1 (Remarks to the Author):

The authors have thoroughly addressed all the points I raised.

We thank the reviewer and are glad to have addressed their points.

Reviewer #3 (Remarks to the Author):

The authors have provided a very thorough answer to all my comments. I find the "integration" aspect much clearer now and the authors have performed various additional analyses, sometimes beyond my expectations. I have no further comments.

Congratulations on an outstanding paper.

We thank the reviewer for their support of this manuscript, and their previous suggestions which have been invaluable for improving the manuscript.

Reviewer #2 (Remarks to the Author):

I appreciate the authors' efforts to address my concerns, though one remains unresolved.

R2C1

1. Figure 1B clearly shows that the coherence in the motion stimuli decreased to maintain performance with a standard staircase method. This aligns with what is commonly referred to as perceptual learning.

The study encompasses two distinct forms of learning: perceptual learning and learning to interpret feedback signals.

In the rebuttal, the authors stated, "By decreasing coherence we maintained performance so the perception of motion direction in the presented stimuli did not improve." This might suggest that the authors believe identical performance (response accuracy) implies the absence of perceptual learning. I am afraid that is not correct. Coherence in motion was reduced precisely because there was an improvement in the perception of motion direction.

I now understand that the intention behind increasing task difficulty was to maintain consistent feedback signals throughout the session. However, Figure 1B and other texts strongly suggest that perceptual learning did occur. This introduces the possibility of alternative interpretations. For example, it could be interpreted that the decoder classified something related to motion perception rather than betting.

Please clarify whether perceptual learning in motion occurred and, if so, explain why this does not pose a problem in the paper.

We thank the reviewer for raising this point, which highlights an important potential ambiguity for us to address. We have done so by adding the necessary clarifications as the reviewer requested as well as by performing additional control analyses as outlined below.

We agree with the reviewer that some of the improvements in this experiment are likely due to perceptual learning (and this is motivates the description of behaviour in the computational model), however, some of the improved performance could also be due to other forms of learning, for example, learning attentional allocation. We therefore chose to use more general terms than 'perceptual learning' when referring to the learning in this experiment ('improve at the task' and 'improve their performance'). [Redacted] Normally, perceptual learning experiments take place over a longer time-period (several sessions over a week). We also did not perform any checks to see if this is truly perceptual learning, which is normally retinotopically specific, and does not generalise across different aspects of the same stimulus feature (for example, perceptual learning for horizontal directions of motion does not generalise to vertical directions of motion).

This is to say, we completely agree with the reviewer that Figures 1B and 1C are indicative of perceptual learning, or at least some form of learning that improves performance in this perceptual decision-making task. This is what we hoped to show in this experiment, that some learning occurs, and if so, whether and how confidence might be used to guide learning even when explicit feedback is frequently available. To make this clearer, we include on Figure 1C the expected d' if participants had not learnt:

We have added note of this in the results P4L91:

"As an additional check, we simulated observers performing the same trials as our human participants, but who do not learn, and so experience decreased sensitivity to the decreased stimulus coherence. This decreased sensitivity is plotted in **Figure 1C** (blue dashed line)."

Moreover, it is unlikely that participants need to learn how to interpret the feedback signals themselves, as the feedback – when provided – was always valid (i.e. fully deterministic). That is, when given explicit feedback the participant knew unambiguously that they had made a correct or incorrect response. The participants were given instructions about this before beginning the task. We have tried to make this more clear in the introduction P3L56:

“Here, we provide observers with intermittent valid explicit feedback during a perceptual decision-making task to test whether observers learn by exploiting their internal confidence estimates, even when explicit feedback is frequently available.”

At the beginning of the Results section P4L78:

“On explicit-feedback trials, participants were shown the true points gained/lost on that trial; on no-feedback trials participants were instructed to infer how many points were awarded, cued with two question marks. Explicit- and no-feedback trials were intermixed throughout the experiment. Participants were given instructions about the feedback prior to beginning the experiment.”

And in the Methods P18L532:

“In the feedback window participants were cued about what type of feedback they would receive. A cyan cue signalled they would receive explicit feedback (text showing +1 point for correct, -1 for incorrect, or +/- 2 in case the participant bet). A magenta cue meant no explicit feedback would be provided, instead participants were shown a question mark cueing them to assess how many points they thought they earned (where points were awarded in the same manner as explicit feedback trials). Participants were given instructions about this prior to the experiment, and all explicit feedback reflected the true points gained/lost on that trial.”

Participants did improve at the task: Their sensitivity (d') did not decrease, despite increased task difficulty (decreased stimulus coherence). This is what we meant by the quoted statement. The d' measure must always be considered relative to the presented stimulus, so no significant change in d' despite more difficult stimuli being presented suggests improved perception. We state this explicitly on P4L90:

“...suggesting participants learnt to improve at the task during the course of the experiment.”

And P5L117:

“...participants relied on explicit feedback to improve their performance (learn).”

We also ran additional analyses (see below) to rule out that the decoder classified something related to motion perception as opposed to confidence. First, we think it's unlikely that motion perception itself is driving post-decision confidence because the stimulus was presented for just 350 ms, and on average participants responded ~330 ms after stimulus offset. EEG components related to visual perception are very fast and not sustained for a long duration, they also tend to be concentrated at the back of the head, which doesn't match our late, central parietal component for decoding bets. More specifically, we see peak decoding of bets around 150 ms after the response, which is consistent with post-decision confidence as we have also shown in previous work (Philiastides et al., *JNeuro*, 2014; Gherman & Philiastides, 2015) (~480 ms after stimulus offset, though keep in mind the timing of the stimulus is jittered relative to the timing of decoding because of the variability in response time).

However, inspired by the reviewer's comment and to more fully/directly address this comment we performed a supplementary analysis, which is now presented in Supplementary Figure S3 (copied below). We used the same decoding analysis as presented in the manuscript, performed across three epochs: 1) EEG activity locked to stimulus onset; 2) EEG activity locked to the response; and 3) EEG activity locked to the cue to enter the bet response. First, we attempted to decode the stimulus direction (left vs right direction of coherent dots, red lines in Figure S3 A), we found decoding accuracy did not rise above chance in any of the

three epochs. Second, we tried to decode something to do with learning, by decoding trials coded as 1st vs 2nd half of the block (where performance improved from the 1st to the 2nd half, then between blocks coherence was decreased to keep sensitivity the same). We were also (not surprisingly) unable to get above chance decoding performance (green lines in Figure S3 A). This provides additional evidence that the decoding of bets in the decision window (locked to response time) is not affected by something to do with motion perception or learning: these signals aren't clear enough in the EEG to decode them directly.

We were able to show above chance decoding performance for the responded motion direction. This first emerged following stimulus onset, peaking around the average time of the response, and the topography is in line with a reliance on a motor signal (yellow lines in Figure S3 A). Decoding performance is increased in epochs aligned to the response, here the decoder could have also relied on the distinct visual information (a left or right arrow was presented on the screen corresponding to the responded direction). Note that the topography looks very different to that of the decoder trained to discriminate bet from no-bet trials.

Finally, and to more fully address comment R2C5b from the previous review (concerning the contamination of motor signals in the decoding of bets), as well as comment R2C2 from the present review, we applied the analysis to decode bets in epochs locked to the onset of the bet cue. In this epoch we can clearly decode whether participants bet. In fact, there are two cues which can easily be pulled out of the EEG signals; the signals related to the button press on bet trials (as indicated from the topography that relies strongly on contralateral motor electrodes), as well as the visual cue (the text on the screen was changed when the participant placed a bet).

The specificity of the bet-prediction used in the manuscript is most strongly demonstrated by the generalisation analysis (Figure S3 B). First, the spatial filter corresponding to training to discriminate the responded direction in the stimulus locked epochs (yellow) cannot be used to obtain predictions differentiating bet vs no-bet trials at any time (top row of Figure S3B). The same is true of the spatial filter corresponding to training to discriminate the responded direction in the stimulus locked epochs (orange). The bet-prediction (used in the manuscript; light blue) differentiates bet from no-bet trials in the response-locked epochs, but not in the epochs locked to the onset of the bet-cue. We show a double-dissociation between this and the spatial filter corresponding to training to discriminate bet from no-bet trials in the epochs locked to the bet cue (dark blue): This spatial filter gives a prediction that differentiates bet vs no-bet trials in the epochs locked to the bet cue, but not the earlier epochs locked to the time of the response. If finger movements or motor readiness potentials had contaminated the decoding of bet responses in the epochs locked to the decision, this cue would be more apparent in the epochs locked to the bet cue, and both filters would discriminate bet responses in both epochs.

Supplementary Figure S3. Decoding of variables around the decision. **A**) Decoding accuracy (area under the ROC (A_z) in a leave-one-out validation procedure) for epochs of EEG data locked to stimulus onset (left) the perceptual decision response (middle) and the cue to place a bet (right). Horizontal black line shows theoretical chance (0.5). We attempted to decode the presented stimulus direction (left vs right, red); the reported direction of the perceptual decision response (yellow), trials in the 1st vs 2nd half of each block (to target learning to improve sensitivity over the course of the block, green); and bet vs no-bet responses (blue). Horizontal coloured bars show significantly above chance decoding performance (corresponding colours) based on t -tests against theoretical chance. We did not observe above chance decoding for stimulus direction nor for trials in the 1st vs 2nd half of the block. We observed significant decoding performance for the responded direction in stimulus onset epochs and response epochs. There were windows of significant decoding of bet responses in all epochs. The distinct topographies (above) suggest these decoders were relying on distinct sources of EEG activity. **B**) Generalisation of the spatial filters over time. As for the bet- and feedback-predictions in the main manuscript, we selected spatial filters for each participant for each of the highlighted significant windows (corresponding to the topographies) and applied these filters over time within each of the epochs. Axis colours and arrows highlight which row corresponds to which decoding analysis: decoding the responded direction in the stimulus epochs (top), in the response epoch (second); decoding bets in the response epochs (third – this is the bet-prediction in the manuscript) and the bet-cue epochs (bottom). The y-axis corresponds to the prediction in terms of the decoded variable, but split by bet and no-bet trials. Decoding the responded direction shows no sign of discriminating bet from no-bet trials, indicating the EEG activity relevant for decoding the responded

direction could not be contaminating the bet-prediction. The decoder discriminating bets in the response-epochs does not generalise to the bet-epochs, in a double-dissociation with the decoder discriminating bets in the bet-epochs which does not generalise to the response-epochs. This is important because the bet-epochs contain the finger movements which provide clear EEG activity to discriminate bet from no-bet trials, yet has little to do with post-decision confidence. The double dissociation suggests EEG activity related to finger movements did not contaminate the post-decision bet-prediction.

Together, this analysis suggests that the EEG bet-prediction is not related to stimulus processing (we cannot decode stimulus direction), to learning (we cannot decode trials with improved performance), to the response direction (the topography is different and does not differentiate bet from no-bet trials), nor to the execution of the bet button press following the bet cue.

We thank the reviewer for this critical feedback, we believe this analysis provides convincing evidence that the EEG bet-prediction is relying on something to do with post-decision confidence (rather than other potentially confounding variables), and will be useful for other readers who, like the reviewer, have healthy scepticism of what information is being used in the EEG bet-prediction.

Minor concerns.

R2C2

1. Please explicitly state in the text that finger movements or motor readiness potentials could be negligible, providing reasons for this assertion.

We thank the reviewer for this suggestion. We have added a sentence to clarify the topography in the Results, and refer to the new analysis in Supplementary Figure S3, P7L182:

“The average topography of the discriminating components of this spatial filter (insert of **Figure 2A**) is in line with the literature on post-decision confidence^{40–43}. There is no evidence that the finger movements in the bet window (1 – 2 s later) had any effect on the LDA, nor potential confounds from the perceptual decision task (see **Supplementary Figure S3**).”

R2C3

2. Figure 3. While the interpretation of the causal relation is acceptable, it is crucial to explicitly state that the effect sizes, represented by the correlation coefficients, are extremely small.

We thank the reviewer for this suggestion, which is in line with previous comments from Reviewer 4. As suggested, we have added a note about the small effect size in the Results P13L377:

“However, we emphasise that this effect is small (Figure 3D) and encourage this integration of confidence and explicit feedback to be more closely assessed in future work.”

We also reiterate this in the discussion P16L461:

“...a Granger causal analysis suggested feedback-window BOLD was also influenced by earlier BOLD in the IFG related to confidence (although this was a small effect).”

Reviewer #4 (Remarks to the Author):

In this revised manuscript, Baldson and colleagues have provided constructive answers to most points raised by all reviewers, leveraging a whole set of new analyses. Despite this net improvement, I remain very conflicted about this manuscript.

On the one hand, I can only acknowledge the quality, volume and sophistication of the provided analyses. On the other hand, I cannot get rid of the sentiment that the paper as a whole somewhat does not succeed to deliver the clear, unambiguous demonstration it ambitions to. After a genuine soul-searching exercise, I think I realized that I am not convinced by the premises of the manuscript about confidence-driven perceptual learning, and I think the chosen formalization (model) fails to clarify what it is and how it should/could work. This really limits my trust in the subsequent demonstration based on the complex analysis of neural signals (especially when analyses become too sophisticated and distal, and attempt to say too much from very noisy data, like the connectivity analysis).

Naturally, these sentiments should probably not be a reason to prevent this manuscript from being published, and scientific progress is made from debates between researchers who “agree to disagree”. Nonetheless, I’ll detail my concerns and doubts below, such that the authors can evaluate if they can engage into any extra effort in the direction of clearing them – what would definitely help to convince the skeptics like me of the underlying value of the present work, starting from the very phenomenon that they are trying to capture.

We thank the reviewer for their careful attention with this manuscript and appreciate their healthy scepticism. We agree that progress requires proposals that may turn out to be disproven. The reviewer has highlighted that the model formalisation lacks clarity, we have responded to this by vastly simplifying the model and describing more clearly how the model is just a simple version of established models of perceptual learning and learning from confidence (Doshier and Lu, PNAS, 1998; Law and Gold, Nat Neuro, 2009). We have also performed additional analyses to fully address the reviewers’ other remaining comments as we outline below.

We want to emphasise that the modelling analysis is independent from the analysis of neural signals. The model was not designed to inform or make testable predictions for the EEG analysis. In fact, the opposite was true: following the analysis of our neural data that suggest a possible role of confidence in learning, we used the model to offer an initial insight into how confidence could be giving rise to the observed behavioural patterns/improvements. We use this model merely as additional evidence that the use of confidence for learning is a better description of behaviour than one in which confidence is ignored. We have also made an effort to highlight in the manuscript that this computational model comparison analysis is not designed to determine the nature of these learning signals nor how they are implemented. The model is simply a way of asking if it is more likely that participants do learn on no-feedback trials or not, and on feedback trials, if this learning is influenced by their confidence or just the feedback value. We leave the details of the computational description to other work (such as Drugowitsch et al., PNAS, 2019; Guggenmos et al., eLife, 2016), and ongoing/future work in our own labs.

Major concerns

R4C1

I realized I am definitely not convinced by the proposed perceptual learning framework. Sure, performance improves over time (or rather, the staircase difficulty increases), but I do not understand (or agree about) what is learned, and whether it is indeed a consequence of feedback and/or confidence. These doubts are actually increased by the chosen computational framework: if I understand correctly, the reinforcement-learning inspired model proposed by the authors basically learn the value of correct response conditional on the evidence but this hardly makes sense: either the agent has access to the (noisy) perceptual evidence in which case the correct response is trivial (the response to be selected is congruent to the evidence) or the agent does not have access to the (noisy) perceptual evidence but then I do not understand what constitutes the stimulus/cue to be associated with a value: there is no way to categorize the cue/evidence ex-ante (μ_L or μ_R ?) before computing its “value”. In simpler terms, how can you update the value of leftward evidence (μ_L) if you haven’t already categorized it as a left-stimulus – in which case the trivial correct answer is

left? Also, what is really learned if perceptual sensitivity (d') does not improve (Figure 1C)? Can it really be “perceptual” learning if perception does not improve?

We thank the reviewer for so clearly articulating this point, which has helped us understand how to simplify and clarify the model to be more accessible to a wider audience. In addressing this point, we first address each of the nuanced points in the comment and then describe how we’ve implemented changes in simplifying the model.

... either the agent has access to the (noisy) perceptual evidence in which case the correct response is trivial (the response to be selected is congruent to the evidence)...

The perceptual evidence is based on the neural response to the stimulus. The perception of motion direction relies on integrating the responses of neurons in V5/MT, where the neural firing statistics can be described by wide (~ 70 deg FWHM, Maunsell and Van Essen, J Neurophys, 1983) bell-shaped functions centred on different directions of motion and receptive fields 4-25 degrees of visual angle (Felleman & Kaas, J Neurophysiol, 1984). We presented observers with random dot motion stimuli of ~ 5 degrees of visual angle, where the majority of dots are moving in random directions. The combination of the large receptive fields and the wide tuning functions means large variability in the neural response to these stimuli. Even integrating over a large population of neurons, the pattern of neural firing in response to a leftward stimulus can look more similar to a pattern of neural firing in response to a rightward stimulus. The correct response is not trivial.

The observer has to infer whether a leftward or rightward stimulus caused the pattern of neural firing. The perceptual evidence can be summarised as the similarity of the pattern of neural firing to that resulting from rightward vs leftward stimuli. By central limit theorem, we can assume the variability in the patterns of firing result in gaussian distributed (noisy) perceptual evidence around the mean values of perceptual evidence for leftward and rightward stimuli. The optimal solution to the task is to set a criterion midway between the two distributions of perceptual evidence to select a rightward response if the perceptual evidence surpasses the criterion (this is the framework of Signal Detection Theory). The response to be selected is congruent to the evidence, but the variability (noise) of the evidence means the evidence is not always indicative of the correct response.

This simplified formalisation using the framework of Signal Detection Theory enables us to capture the neuronal properties of motion perception as well as describe the process of perceptual learning in line with the literature, as outlined further below.

...In simpler terms, how can you update the value of leftward evidence...

Perceptual learning occurs by fine-tuning the weights on neurons integrated across the population to improve the signal-to-noise ratio of the perceptual evidence (Doshier and Lu, PNAS, 1998). This adjustment of the neural weights for perceptual learning has already been demonstrated using a reinforcement framework (Law and Gold, Nat Neuro, 2009) and the optimal integration of confidence with explicit feedback has also been described using this reinforcement learning based weight updating (Drugowitsch et al., PNAS, 2019). In this framework, a negative prediction error weakens weights on neurons whose firing contributed to the perceptual evidence leading to the response, while a positive prediction error strengthens the weights on neurons whose firing contributed to the perceptual evidence leading to the response. In other words, rather than updating the value of leftward evidence, we are improving the correspondence between the external stimulus and the internal perceptual evidence, by learning which neurons to listen to.

In our original model we sought a simplification of this existing framework describing the reweighting of neurons. Instead of simulating neurons and adjusting the weights on these neurons, we implemented the downstream effect on the noisy perceptual evidence (in the Signal Detection Theory framework described above). In accordance with Doshier and Lu (1998) we implemented both an improvement in the mean signal strength, as well as a reduction in the variance of the noise. This meant 4 parameters for learning rates that describe how the mean and variance of the distributions of perceptual evidence for left and right motion directions are adjusted trial to trial based on the learning signal (reward prediction error). Essentially what this was doing was improving the signal-to-noise ratio. We’ve simplified the model so that the adjustment is

only made on the means, with the same learning rate for positive and negative learning signals (reward prediction errors; meaning 1 parameter for a single learning rate). This in turn simplifies our explanation of the model, so the reader can clearly see the basis in Signal Detection Theory with less distraction from the literature on weighted integration of MT/V5 neural firing.

The reasoning behind the adjustments of the mean perceptual evidence distributions in the newly simplified model is very simple. If one makes an incorrect response, and has a negative prediction error, the perceptual evidence was likely close to the criterion, weakening the weights on neurons whose firing led to perceptual evidence close to the criterion should result in evidence less close to the criterion, so both the mean perceptual evidence for left and right stimuli are moved away from the criterion. In contrast, if one makes a correct response (or more specifically, has a positive reward prediction error), the perceptual evidence was likely close to (or greater than) the mean for that stimulus, strengthening the weights on neurons whose firing led to perceptual evidence close to the mean of that stimulus should increase the mean evidence in response to that stimulus away from the criterion (leaving the response to the other stimulus the same, since these neurons are likely tuned to directions very far, ~ 180 degrees from each other). Every time an adjustment is made (irrespective of whether the reward prediction was positive or negative), the signal-to-noise ratio of the perceptual evidence is improved.

We hope to have improved Supplementary Figure S1 to outline this more fully. We copy the full Figure below, which the reviewer will see also contains model and parameter recovery analysis as suggested in R4C3 below.

Supplementary Figure 1. Computational modelling. *A)* We implement a simple Signal Detection Theory framework to describe perceptual decisions and bet responses in the experiment. According to the assumptions of Signal Detection Theory, the presentation of an external stimulus results in some internal perceptual evidence which is disrupted by Gaussian distributed noise. In this case, the leftward and rightward stimuli produce perceptual evidence of different means (μ_L , and μ_R), but enough noise that the distributions of evidence overlap, meaning that on some trials a leftward stimulus results in perceptual evidence that is more similar to that resulting from a rightward stimulus. To make a decision, the observer implements a criterion, which we assume is unbiased (0) for simplicity (thick vertical line). The observer responds 'right' if the evidence exceeds this criterion. Assuming the Gaussian distributed noise has unit variance (for simplicity), the distance between the means determines the observers' sensitivity in discriminating the stimuli. Bet responses are made based on

an additional criterion (b , dashed vertical lines), where the observer bets if the evidence exceeds this criterion. **B)** Previous work has demonstrated that perceptual learning increases sensitivity (equivalent to increasing the distance between the means), and this can be achieved by adjusting the weighted integration of neural activity from neurons tuned to different motion directions. In essence, a negative prediction error ($L_t < 0$, which is most likely occur from evidence close to the criterion, red bar) should prompt the observer to place less weight on neurons promoting this evidence (those tuned away from the left/right directions), and this would be equivalent to both means being shifted away from the criterion. A positive prediction error should prompt the observer to increase the weights on the evidence that led to that perceptual evidence (those tuned close to the direction of the response, blue and orange bars) and this would be equivalent to the mean for the chosen response shifting further from the criterion. **C)** Proportion correct as a function of the perceptual evidence given the distributions in A. For simplicity, we assumed confidence scaled with this ideal, and calculated trial-wise confidence from the expected value of the perceptual evidence given the distributions, the participant's perceptual decision, and bet response (which approximates the location of the evidence and its likelihood). **D)** Participant proportion correct vs the simulated proportion correct from the fitted parameters of the feedback only model (top), mixture model (middle) and the confidence model (bottom) for bet (light blue filled markers) and no-bet (dark blue open markers) separately. The model was fit to minimise the negative log-likelihood of the participants' perceptual decisions and bet responses and so gives a representation of the fit of the model to individual data. **E)** Sensitivity (d') by experimental block for participants (black, 95% within subject confidence intervals shaded) and the model predictions (dashed lines). **F)** Parameter recovery analysis: for each model, we simulated behaviour in our experiment (300 trials) and compared the simulated parameters with those fitted by our model fitting procedure. The simulated and fitted parameters showed strong linear correlation (average spearman's $\rho = 0.93$, minimum $\rho = 0.85$). The sum of squared error between the simulated and fitted parameters was on average just 15% of the squared error of the simulated parameters from their mean (maximum = 33%, for starting μ). **G)** Model recovery analysis: we simulated behaviour from each model and fit all three models, allowing us to compare the fit across 23 participants as in our experiment. The horizontal bars show the probability of each model appearing as the winning model. If the true model was the feedback only model (model 1) the chance finding the other models superior was $p < 0.001$; for the mixture model (model 2) the chance of finding the feedback model superior was $p = 0.004$ and the confidence model, $p = 0.036$; and for the confidence model (model 3) the chance of finding the feedback model superior was $p < 0.001$ and the mixture model, $p = 0.046$.

... Also, what is really learned if perceptual sensitivity (d') does not improve (Figure 1C)? Can it really be "perceptual" learning if perception does not improve?

We hope that now it is clearer why decreasing external stimulus coherence while maintaining the sensitivity of discrimination decisions is indicative of learning. Sensitivity should always be considered relative to the external stimulus. When we decrease coherence, there are more dots moving in random directions, meaning the neural response will again be less indicative of the true stimulus direction (Britten et al., Vis. Neurosci, 1993), meaning the perceptual evidence for the decision in our Signal Detection Theory framework is more variable (the signal-to-noise ratio is lower) and by definition, d' is worse (d' is the distance between the means in units of standard deviation). If d' is *not* worse, we would assume there has been some improvement such that the additional external noise is compensated for by an increase in the internal signal-to-noise ratio. Varying the stimulus with the aim of maintaining performance is standard practice in the perceptual learning literature (for example, Doshier and Lu, 1998; Law and Gold, 2009; Guggenmos et al., eLife, 2016).

To make this clearer in the manuscript, we have included the simulation of no-learning in Figure 1C. Also in response to R4C3 below, we simulated behaviour in the experiment for observers who start with the same sensitivity as our human participants, are presented with the same trials (with decreased coherence), but do not learn. The results clearly demonstrate the decreased sensitivity that would have occurred if our human participants did not learn.

To fully address this comment and the opening statement of the reviewer, we have simplified the model and have been careful to describe the simple assumptions of Signal Detection Theory. In the methods P20L576:

“To support the behavioural results indicating participants use confidence to learn, we used a simple computational model comparison. The models assume the basic framework of Signal Detection Theory⁸⁰ (**Supplementary Figure S1**). On each trial the observer has a sample of noisy perceptual evidence and must decide if this evidence resulted from the presentation of a leftward or rightward stimulus. We assume the noise affecting the perceptual evidence is Gaussian distributed with unit variance ($\sigma = 1$), added to the different mean perceptual evidence from leftward and rightward stimuli (μ_L, μ_R). To decide, the observer compares the perceptual evidence to a criterion, c , above which the observer responds that the perceptual evidence resulted from a rightward stimulus. Incorrect responses result from the overlap in the distributions, where the noise affecting the perceptual evidence can push the evidence to the wrong side of the criterion. Sensitivity, d' , is defined as:

$$d' = \frac{|\mu_R - \mu_L|}{\sigma} \quad (1)$$

Learning improves sensitivity. Previous work has addressed how adjusting the weighted integration of neural activity can improve the signal-to-noise ratio of the perceptual evidence^{2,3}, thus improving sensitivity. We implement this in our model by moving the means (μ_L, μ_R) further from the criterion. For simplicity, we fix the criterion at 0 and use a parameter, μ_0 , to define the starting values of the mean perceptual evidence, such that $\mu_L = -\mu_0, \mu_R = \mu_0$, and $d' = 2\mu_0$. A second free parameter, b , determines bet decisions by operating as a secondary criterion; the observer bets when the absolute value of the perceptual evidence exceeds this criterion. A change in the stimulus coherence ($coh_{old} \rightarrow coh_{new}$) has a systematic effect on the mean evidence:

$$\mu_{new} = \mu_{old} \cdot \frac{coh_{new}}{\sqrt{2\sigma^2} coh_{old}} \quad (2)$$

Without learning, d' would decrease with decreased coherence. Instead, learning shifts the means of the distributions of perceptual evidence further from the criterion (balancing the shift back toward the criterion with decreased coherence). We implement these shifts with learning in accordance with reinforcement learning frameworks (as in¹³): A third free parameter, the learning rate, α , moderates the influence of a trial-wise learning signal, L_t , to update the means on each trial:

$$|\mu_{t+1}| = |\mu_t| + \alpha L_t \quad (3)$$

If L_t is positive, the mean corresponding to the chosen stimulus is updated, otherwise both means are updated (moved away from the criterion, 0; see **Supplementary Figure S1**). We compared how this learning signal incorporated confidence across three models. The basis of the learning signal is the reward prediction error, the difference between the explicit value of the feedback on that trial, r_t , and the expected value on that trial, $E[V_t]$. The expected value started at 1 (expecting to be correct) and was updated according to a Rescorla-Wagner rule⁸¹ using the same learning rate as Equation 3. Model 1 did not incorporate confidence, the reward prediction error was computed on explicit feedback trials, and set to 0 on no-feedback trials. Model 2 used this same

reward prediction error on explicit feedback trials, but used confidence as the learning signal on no-feedback trials. Model 3 used confidence on all trials: on explicit feedback trials, the expected value was moderated by confidence; on no-feedback trials confidence was used as the learning signal as in Model 2. In all cases, if the participant bet on that trial, the expected value was increased by 1 (in line with the points earned).

Trial-wise confidence estimates were extracted based on the expected value of the perceptual evidence on each trial. Confidence was computed according to an ideal observer model⁸² as the probability of a correct response given the perceptual evidence (based on the cumulative density of the joint distribution of evidence).

All models had three free parameters: The initial mean perceptual evidence, μ_0 , the criterion for betting, b , and the learning rate, α . The models were fit to minimise the negative log likelihood of the participant's perceptual decisions and bet responses on each trial. The log likelihoods were calculated from the cumulative Gaussian probability density corresponding to the choice and confidence (demarcated by the criteria), and the model was fit using a constrained nonlinear interior point optimisation algorithm implemented with MATLABs `fmincon` function. Model and parameter recovery analyses are presented in **Supplementary Figure S1**.

We note that these simple models do not capture all behavioural patterns in the data, for example, all three models failed to produce as large an interaction between feedback value and feedback sign as in the data. But the purpose of this modelling exercise was merely to support the behavioural results suggesting participants were using their confidence to learn: The largest difference between the models is on which trials confidence influences learning. These models were not designed to maximally capture behaviour. Instead, we focus on the neural mechanisms of learning from confidence, leaving computational model development to future work.”

The results of this simpler model are very much in line with the previous results. In the results section we have also tried to highlight the purpose of the modelling, and that the distinction between the models is mainly due to which trials learning occurs on and whether confidence is used P5L124:

“We used a simple computational model comparison to support the behavioural evidence for learning from confidence. The models assume the basic framework of Signal Detection Theory³⁸, where responses are made by placing a criterion to discriminate the perceptual evidence from leftward vs rightward stimuli (two overlapping Gaussian distributions). Learning improves sensitivity to the stimuli², the distance between the means of the perceptual evidence, in units of standard deviation (though sensitivity is decreased for stimuli with decreased coherence). We model learning by shifting the mean perceptual evidence, μ , away from the response criterion (see **Methods** for details). The shift is implemented in accordance with reinforcement learning, where the size of the shift is proportional to the reward prediction error (the difference between the explicit feedback value, r_t , and the expected value, $E[V_t]$), moderated by a learning rate, α , such that: $\mu_{t+1} = \mu_t + \alpha(r_t - E[V_t])$. Three models were implemented to compare which trials participants use confidence to learn on: 1) a model that does not use confidence, learning only occurs on explicit feedback trials; 2) a model that learns in the same way on feedback trials, but additionally uses confidence to learn on no-feedback trials (confidence substitutes the reward prediction error); 3) a model that uses confidence to learn on all trials, where confidence moderates the expected value on explicit feedback trials (similar to¹³). The model that used confidence on all trials provided the best description of behaviour (**Figure 1C,F**; $\sum BIC_3 - BIC_1 = -82.21$; $\sum BIC_3 - BIC_2 = -32.11$; protected exceedance probability = 0.94; see **Methods, Supplementary Figure S1**). This suggests confidence is used for learning even on trials where explicit feedback is provided. Indeed, we found that simulated behaviour learning from explicit feedback alone did not show the difference in response perseveration following bet/no-bet responses (as in **Figure 1F**, $p = 0.046$). Note that this modelling exercise merely supports the use of confidence for learning, we do not seek to examine its exact implementation. Indeed, these models do not sufficiently capture all aspects of the behavioural data, such as the interaction between feedback value and sign, which may require more complex model formalisations.”

The model comparison is not for the purpose of understanding the details of the implementation of learning per se, which we now state explicitly, and highlight the model limitations, in relation to comment R4C2 below.

The results of this simplified model comparison analysis are very much the same as the more complex models before. This is also reflected in the analysis using the EEG predictions as the learning signals, where the results do not substantially change P9L240:

“We used the feedback-window EEG-predictions (feedback-prediction on feedback trials and bet-prediction on no-feedback trials) as the reward prediction errors in the computational model, and found this resulted in no substantial difference in the fit to behaviour (compared to the behaviour-only model relying on the explicit feedback and bet responses; $\sum BIC_{beha} - BIC_{EEG} = 1.004$, protected exceedance probability in favour of the EEG-informed model = 0.45). But, the EEG-predictions from the feedback-window provided a better description of behaviour than those from the bet-window ($\sum BIC_{EEG_{fb}} - BIC_{EEG_d} = -141.08$; protected exceedance probability > 0.99). This indicates that the feedback-window predictions are more related to outcome value and its use for learning, than the earlier signals from these same spatial filters in the decision-window.”

We hope that in simplifying the models and being more careful to explain the formalisation clearly, we have alleviated the concerns of the reviewer, and other readers who may share these concerns. We are very grateful to the reviewer for this opportunity to improve the manuscript.

R4C2

Regarding the model validation using the pattern of perseveration (Figure 1F), wouldn't the same pattern be observed “statistically” if the agent does not respond to feedback at all (no learning), but “simply” produces ~70% of correct responses (e.g. uniform between 60-80%), and only bet when they expect a certain accuracy (e.g. >70%) and we keep the task structure as is (pseudo-randomization of trials, etc.)? What is the unit of the y-axis? It also seems that the model fails to incorporate the magnitude dimension of the feedback, which does not surprise me, given its structure.

We thank the reviewer for highlighting the information lacking from this results and Figure. The perseveration analysis took the probability of repeating a response given a repeating/alternating stimulus given the previous feedback, and normalised this by the probability of repeating a response given a repeating/alternating stimulus irrespective of feedback. This detail was in the methods section, which we have clarified, P19L568:

“Response perseveration was calculated as the probability of repeating a response given a stimulus repeat or stimulus alternation in each feedback condition. To account for different base rates of repetition, this probability was normalised and divided by the overall normalised probability of repeating a response to a repeating/alternating stimulus of each participant. Statistics (2x2 repeated measures ANOVA for the explicit feedback conditions, t-test for bet/no-bet on no-feedback trials) were computed on the difference between stimulus repeat and stimulus alternate scores.”

We have added mention of this normalisation to the Results, P5L110:

“This was indeed the case, a 2 x 2 repeated measures ANOVA showed a significant main effect of feedback sign (positive vs negative) on the tendency to repeat a response given a stimulus repetition (normalised by the repetition tendency irrespective of feedback; **Figure 1F, left**; $F(1,22) = 60.04$, $p < 0.001$ after Bonferroni correction for three comparisons). This effect was somewhat moderated by feedback value ($|1|$ vs $|2|$; but the interaction would not survive correction for multiple comparisons; $F(1,22) = 5.18$, $p = 0.033$, uncorrected).”

And to the Figure Y-axis, which now reads: “Perseveration (z: normalised p(repeat response))”

We have also specified on the x-axis that this refers to what feedback occurred on the previous trial (or whether the participant bet on the previous trial, if there was no feedback). Because we normalise by the overall probability of repeating a response given a repeated or alternated stimulus, the baseline is 0. We checked this by repeating the analysis for each participant over 1000 permutations of the order of previous trial feedback labels. The results of this analysis are presented below in the circles overlaid on Figure 1F.

Of course, this removes the autocorrelation in the feedback, where correct feedback is more likely to follow any form of feedback because correct responses are overall more likely. Accounting for this autocorrelation is difficult, because some of the autocorrelation is what is predicted by learning from feedback. One thing we can do is randomise feedback only within correct vs incorrect previous responses. We performed 1000 permutations of the feedback labels within correct vs incorrect previous responses. We found the probability of obtaining as large a difference in perseveration due to feedback sign as observed in our data was $p = 0.085$; the probability of obtaining an interaction as large, $p = 0.025$; and a difference due to confidence, $p = 0.016$. We present the full distributions of permutations below, with our data in red markers.

This is not exactly what the reviewer requested, to examine the pattern “if the agent does not respond to feedback at all (no learning)”, since we have already shown that participants did learn. We simulated responses from our basic Signal Detection Theory framework with 0 learning rate. Simulated agents performed the same trials as our human participants, but did not learn to improve their performance. In this case, the probability of obtaining as large a difference in perseveration due to feedback sign as observed in our data was $p < 0.001$; the probability of obtaining an interaction as large, $p = 0.006$; and a difference due to confidence, $p = 0.004$.

We have added these statistics to the results section P5L113:

“As an additional check, we compared the differences in response perseveration with the behaviour of simulated observers who do not learn (as in Figure 1C dashed blue line). We found that the probability of obtaining as large a difference in perseveration due to feedback sign as observed in our data was $p < 0.001$; and the probability of obtaining an interaction as large, $p = 0.006$.”

Note that if the participant had not improved their performance, we would not have decreased coherence. A participant who does not learn, but is presented with decreased coherence, shows decreased sensitivity (Figure 1C). So, this does not address the other aspect of the reviewer’s comment, that these differences could be obtained statistically given accuracy. To address this, we again simulated agents with 0 learning rate, but this time simulated the full experiment, adjusting coherence as we would have during the real experiment. Without learning, coherence was roughly maintained (starting on average at 27% and ending on average at 35%, as opposed to 17% in the real experiment). In this case, the probability of obtaining as large a difference in perseveration due to feedback sign as observed in our data was $p < 0.001$; the probability of obtaining an interaction as large, $p = 0.008$; and a difference due to confidence, $p = 0.006$.

We agree that the interaction between feedback sign and value appears smaller for the model than for the participants. The model using confidence for learning gives appreciably similar expected values for the difference due to feedback sign $p(\text{model} > \text{data}) = 0.506$, and the difference given previous bets $p(\text{model} > \text{data}) = 0.164$, but the interaction is statistically smaller $p(\text{model} > \text{data}) = 0.016$.

This is true of all our models. Whether we model learning only from explicit feedback, using confidence on all trials, or a mix between explicit feedback and confidence, we do not obtain an interaction as large as in our data:

We make sure to mention this explicitly in the Results P5L143:

“Note that this modelling exercise merely supports the use of confidence for learning, we do not seek to examine its exact implementation. Indeed, these models do not sufficiently capture all aspects of the behavioural data, such as the interaction between feedback value and sign, which may require more complex model formalisations.”

And in the Methods P21L622:

“We note that these simple models do not capture all behavioural patterns in the data, for example, all three models failed to produce as large an interaction between feedback value and feedback sign as in the data. But the purpose of this modelling exercise was merely to support the behavioural results suggesting participants were using their confidence to learn: The largest difference between the models is on which trials confidence influences learning. These models were not designed to maximally capture behaviour. Instead, we focus on the neural mechanisms of learning from confidence, leaving computational model development to future work.”

We would like to emphasise that none of our other analyses rely on this model. We include it because our neural data suggests confidence is being used to learn, and possibly integrating confidence with explicit feedback. The model comparison suggests this is a good description of behaviour too.

R4C3

In addition to the conceptual problems outlined above, I also found the implementation and validation of the model difficult to read and lacking critical quality control analyses, such as parameter recovery and model identification analyses (Wilson & Collins, 2019). The inter-individual account given in Supp. Figures 1G-I is a very distal appreciation of the data, and capturing inter-individual variance does not necessarily mean that the important dimensions of within-individual variance are correctly captured by the model. As stated above, I'm not convinced that the figure provided in Figure 1F constitutes a decent support for the model in favor of alternative simpler explanations. The authors would need to find a unique signature of the model that can be found in the behavior and falsifies the alternative accounts (Palminteri et al., 2017)

We are very grateful that the reviewer has reminded us to perform this model and parameter recovery analysis, which we should have included earlier. We have now included this in Supplementary Figure S1. As is clear from the figure, parameter recovery in all models was quite good. The simulated and fitted parameters showed strong linear correlation (average spearman's rho = 0.93, minimum rho = 0.85). The sum of squared error between the simulated and fitted parameters was on average just 15% of the squared error of the simulated parameters from their mean (maximum = 33%, for starting mu; though this may be a limitation in the data, 50 trials is not much for estimating d'). The model recovery was also good: 1000 samples of 23 participants from the set of simulated and fitted models, we found that if the true model was the feedback only model (model 1) the chance finding the other models superior was $p < 0.001$; for the mixture model (model 2) the chance of finding the feedback model superior was $p = 0.004$ and the confidence model, $p = 0.036$; and for the confidence model (model 3) the chance of finding the feedback model superior was $p < 0.001$ and the mixture model, $p = 0.046$. This supports the results presented in the manuscript suggesting that it is unlikely that we would have found support for the confidence model in the case that participants were using feedback only or a mixture.

The inter-individual account, plotting participant vs simulated proportion correct was meant to provide a visualisation of the model fit. Seeing as we are fitting to minimise the negative log-likelihood of both perceptual decisions and bet responses, we have additionally split this between bet and no-bet response (Supplementary Figure 1D). We agree that this does not give a great appreciation of how well the models capture important aspects of the data. For this reason, we have also included the model predictions of sensitivity (d') over the blocks of trials. The models are similar, but the confidence model is clearly a better description of how participant sensitivity evolves over the course of the experiment. Of note, the feedback only model underestimates learning (sensitivity decreases). We suspect this is because learning can only be implemented on half the trials, so the model would need to shift the means twice as much on these trials. These larger shifts on half the trials are clearly not indicative of behaviour, as the fit suggests the better predictor of participants' responses is learning less rather than learning the right amount but on intermittent trials.

One of the core motivators of this experimental design is to examine learning on trials without feedback, in a context where participants could rely solely on the frequent explicit feedback to learn. We express this at the beginning of the results section P4L94:

“We designed this experiment such that participants were frequently given explicit feedback (50% of randomly intermixed trials), and so did not have to rely on decision confidence as the sole source of information to improve their performance. Evidence for learning on no-feedback trials could therefore be considered robust evidence for the involvement of confidence in learning.”

In behaviour, we see indications of learning on no-feedback trials from the perseveration analysis. As can be seen from the Figure above, the feedback only model struggles to match this difference in perseveration due to previous trial bet responses ($p = 0.046$), where as the models with confidence do not show statistically significant differences. We have now included this detail in the results, P5L141:

“Indeed, we found that simulated behaviour learning from explicit feedback alone did not show the difference in response perseveration following bet/no-bet responses (as in **Figure 1F**, $p = 0.046$).”

We think this could be considered a falsification of the feedback only model.

Minor

R4C4

If the authors really want to show that some signal (RT and/or EEG) is a signature of confidence, wouldn't it be convincing to show that it follows classical signatures of confidence signals, such as the X-pattern of confidence as a function of evidence for correct vs incorrect choices? See e.g. (Sanders et al., 2016).

We thank the reviewer for this suggestion. We present data concerning two typical signatures of confidence: greater performance for high confidence and faster reaction times for correct high confidence trials. As we explain below, we don't have the data to plot the suggested X-pattern because of the nature of our experimental design. Sanders et al., 2016 present this X-pattern of confidence as a function of 'discriminability': “*We found that the mean statistical confidence for a given level of discriminability increases for correct and decreases for incorrect choices*”. In many experiments, one can assume that different levels of external stimulus 'difficulty' (such as coherence) correspond to different levels of discriminability (referring to the internal representation of the stimulus, which Sanders and colleagues refer to as d_{hat} , Figure 1A of their manuscript). Here we do present participants with 6 levels of stimulus coherence (coherence is adjusted each block). But we cannot assume that these levels of stimulus coherence correspond to different levels of discriminability, in fact, we show that they don't, the adjustment is made to try to match the external stimulus difficulty with the improved discrimination ability (maintaining sensitivity/discriminability across blocks). In other words, we don't have the data to plot this X-pattern.

Palminteri, S., Wyart, V., & Koechlin, E. (2017). The Importance of Falsification in Computational Cognitive Modeling. *Trends in Cognitive Sciences*, 21(6), 425–433. <https://doi.org/10.1016/j.tics.2017.03.011>

Sanders, J. I., Hangya, B., & Kepecs, A. (2016). Signatures of a Statistical Computation in the Human Sense of Confidence. *Neuron*, 90(3), 499–506. <https://doi.org/10.1016/j.neuron.2016.03.025>

Wilson, R. C., & Collins, A. G. (2019). Ten simple rules for the computational modeling of behavioral data. *eLife*, 8, e49547. <https://doi.org/10.7554/eLife.49547>

REVIEWERS' COMMENTS

Reviewer #2 (Remarks to the Author):

I asked to clarify whether perceptual learning in motion occurred, and if so, explain why this does not pose a problem in the paper.

The supplementary data, added in response to my concern, which I appreciate, was enough to show that the occurrence of perceptual learning does not matter in the paper.

However, the authors avoid clarifying whether perceptual learning in motion did occur. Improvement at a task would be a type of perceptual learning, according to researchers (for instance, see Watanabe et al., *Annu Rev Psychol*, 2015; Doshier et al., *Annu Rev Vis Sci* 2017). I do not think avoiding the term perceptual learning does not mean that perceptual learning did not happen.

Rather, the paper would be much more robust if the authors clarified that perceptual learning occurred and then argued that this was not a problem in their interpretation.

Reviewer #2 (Remarks on code availability):

I am not capable of reading codes.

Reviewer #4 (Remarks to the Author):

Once again, I commend the authors for engaging very constructively and very thoroughly on the issue that I raised. I appreciate the attempt to clarify and simplify the model, and the additional analyses regarding model validation/falsifications.

While some conceptual disagreements may persist, I think the manuscript deserves to be published, so that it can feed the scientific debate on confidence-driven perceptual learning.

Congratulations to the authors for this very thorough piece of work.

Reviewer #2 (Remarks to the Author):

I asked to clarify whether perceptual learning in motion occurred, and if so, explain why this does not pose a problem in the paper.

The supplementary data, added in response to my concern, which I appreciate, was enough to show that the occurrence of perceptual learning does not matter in the paper.

However, the authors avoid clarifying whether perceptual learning in motion did occur. Improvement at a task would be a type of perceptual learning, according to researchers (for instance, see Watanabe et al., *Annu Rev Psychol*, 2015; Doshier et al., *Annu Rev Vis Sci* 2017). I do not think avoiding the term perceptual learning does not mean that perceptual learning did not happen.

Rather, the paper would be much more robust if the authors clarified that perceptual learning occurred and then argued that this was not a problem in their interpretation.

Author response:

We entirely agree with the reviewer and have changed the following statement to highlight more explicitly the role of perceptual learning (the added phrase is highlighted in bold italic):

“There was an overall increase in task difficulty (Figure 1b, mean difference in coherence between first and last block = $-10.43\% \pm 4.67$, 95% within subject confidence interval, $t(22) = 4.64$, $p < 0.001$) with no substantial change in sensitivity (Figure 1c, mean difference in $d' = -0.37 \pm 0.45$ $t(22) = 1.67$, $p = 0.109$, with Bayes factor, calculated based on the savage-dickey ratio with a unit-information prior, $BF_{10} = 0.75$, representing insubstantial evidence in favour of the null hypothesis for no difference in sensitivity). ***This is a typical signature of perceptual learning***, suggesting participants learnt to improve at the task during the course of the experiment.”

We also highlight more explicitly in the main text that the computational model is based on previous perceptual learning models, referencing Doshier and Lu (1998) and Law and Gold (2009) (as we do in the methods). We copy the sentence here, highlighting the added phrase in bold italic.

“The models, ***based on previous perceptual learning models***^{2,3}, assume the basic framework of Signal Detection Theory³⁸, where responses are made by placing a criterion to discriminate the perceptual evidence from leftward vs rightward stimuli (two overlapping Gaussian distributions). Learning improves sensitivity to the stimuli², the distance between the means of the perceptual evidence, in units of standard deviation (though sensitivity is decreased for stimuli with decreased coherence).”

We thank the reviewer for their comments on the manuscript, which have been very useful for improving its quality and comprehensibility.

Reviewer #2 (Remarks on code availability):

I am not capable of reading codes.

Reviewer #4 (Remarks to the Author):

Once again, I commend the authors for engaging very constructively and very thoroughly on the issue that I raised. I appreciate the attempt to clarify and simplify the model, and the additional analyses regarding model validation/falsifications.

While some conceptual disagreements may persist, I think the manuscript deserves to be published, so that it can feed the scientific debate on confidence-driven perceptual learning. Congratulations to the authors for this very thorough piece of work.

Author response:

We thank the reviewer for all their comments and for their support of our manuscript. We are grateful for the opportunity to improve the manuscript, especially for those readers who may approach the manuscript from different conceptual stances.